# RNA is a key component of extracellular DNA networks in *Pseudomonas aeruginosa* biofilms

Sudarsan Mugunthan[1], Lan Li Wong[1], Fernaldo Richtia Winnerdy[2], Stephen Summers[1,3], Muhammad Hafiz Bin Ismail[4], Yong Hwee Foo [1,5], Tavleen Kaur Jaggi[6], Oliver W. Meldrum[6], Pei Yee Tiew[7], Sanjay H. Chotirmall [6,8], Scott A. Rice [1,9,14], Anh Tuân Phan [10], Staffan Kjelleberg [1,11,12] ✉ & Thomas Seviour [1,13] ✉

The extracellular matrix of bacterial biofilms consists of diverse components including polysaccharides, proteins and DNA. Extracellular RNA (eRNA) can also be present, contributing to the structural integrity of biofilms. However, technical difficulties related to the low stability of RNA make it difficult to understand the precise roles of eRNA in biofilms. Here, we show that eRNA associates with extracellular DNA (eDNA) to form matrix fibres in *Pseudomonas aeruginosa* biofilms, and the eRNA is enriched in certain bacterial RNA transcripts. Degradation of eRNA associated with eDNA led to a loss of eDNA fibres and biofilm viscoelasticity. Compared with planktonic and biofilm cells, the biofilm matrix was enriched in specific mRNA transcripts, including *lasB* (encoding elastase). The mRNA transcripts colocalised with eDNA fibres in the biofilm matrix, as shown by single molecule inexpensive FISH microscopy (smiFISH). The *lasB* mRNA was also observed in eDNA fibres in a clinical sputum sample positive for *P. aeruginosa*. Thus, our results indicate that the interaction of specific mRNAs with eDNA facilitates the formation of viscoelastic networks in the matrix of *Pseudomonas aeruginosa* biofilms.

Extracellular DNA (eDNA) was first identified as a key component of the *Pseudomonas aeruginosa* biofilm matrix in the early 2000s and was found to play a vital role in initial biofilm attachment[1]. It was subsequently observed in other environmental and clinical biofilms, initially with the environmental river isolate F8 that produces stable filamentous network structures of eDNA[2], followed by a range of pathogenic (*Staphylococcus aureus*, *Staphylococcus epidermidis*, *Escherichia coli*, *Burkholderia pseudomallei* and *Haemophilus influenzae*) and

[1]Singapore Centre for Environmental Life Sciences Engineering, Nanyang Technological University, Singapore 637551, Singapore. [2]School of Medicine, University of California, San Francisco, CA 94158, USA. [3]St John's Island National Marine Laboratory c/o Tropical Marine Science Institute, National University of Singapore, 119227, Singapore. [4]Environmental Health Institute, National Environmental Agency, Singapore 138667, Singapore. [5]Institute for Digital Molecular Analytics and Science (IDMxS), Nanyang Technological University, Singapore 636921, Singapore. [6]Lee Kong Chian School of Medicine, Nanyang Technological University, Singapore 636921, Singapore. [7]Department of Respiratory and Critical Care Medicine, Singapore General Hospital, Singapore, Singapore. [8]Department of Respiratory and Critical Care Medicine, Tan Tock Seng Hospital, Singapore, Singapore. [9]The iThree Institute, University of Technology Sydney, Sydney 2007, Australia. [10]School of Physical & Mathematical Sciences, Nanyang Technological University, Singapore 637371, Singapore. [11]School of Biological Sciences, Nanyang Technological University, Singapore 637551, Singapore. [12]School of Biological, Earth and Environmental Sciences, University of New South Wales, Sydney 2052, Australia. [13]Centre for Water Technology (WATEC), Department of Biological and Chemical Engineering, Aarhus University, Aarhus 8000, Denmark. [14]Present address: CSIRO, Agriculture and Food, Westmead and Microbiomes for One Systems Health, Canberra, Australia. ✉e-mail: laskjelleberg@ntu.edu.sg; twseviour@bce.au.dk

environmental (*Microbacterium sp.* and *Serratia sp.*) bacterial biofilms, demonstrating that eDNA is a key component of many biofilms[3–5]. While the localisation[6] and importance of eDNA in the extracellular biofilm matrix of *P. aeruginosa* have been described, the mechanisms for its export by *P. aeruginosa* into the extracellular environment have not been resolved. An early model proposed that quorum sensing mediates eDNA release, as *lasI*, *rhlI*, *pqsA*, *pqsL*, and *fliM-pilA* deficient mutants produced less eDNA[6]. It was subsequently shown that a cryptic prophage endolysin activated explosive cell lysis, resulting in the release of eDNA from the cytosol into the extracellular biofilm matrix environment[7,8]. Despite these observations, there is still no eDNA knockout mutant for *P. aeruginosa*[9], in contrast to *S. epidermidis* where eDNA release is regulated by the autolysin AtlE. Δ*atlE* mutants cannot produce matrix eDNA and their biofilms are less well-developed[10].

Biofilms are typically viscoelastic, and this is generally attributed to the extracellular polymeric substances (EPS) secreted by the bacterial cells[11]. This viscoelastic behaviour is completely removed for *P. aeruginosa* biofilms in the early stages of growth by DNase I treatment, resulting in biofilm dissolution[1,9]. Furthermore, EndA, a DNA-specific endonuclease, degrades eDNA in *P. aeruginosa* biofilms[12], and leads to biofilm dispersal. These findings are consistent with our recent report that eDNA is primarily responsible for the viscoelasticity of *P. aeruginosa* biofilms[9], where the loss of eDNA fibre structures was coincident with biofilm dissolution. It remains unclear, however, how eDNA builds viscoelastic extracellular biofilm structures and which supramolecular interactions are involved. Hence, understanding the higher-order structure of DNA or its interaction with other exopolymers upon the transition to eDNA is key to resolving how *P. aeruginosa* uses eDNA to build extracellular viscoelastic structures.

There has thus been a recent shift in eDNA research towards describing factors that allow eDNA to form network structures and how these associates with other EPS molecules. The association of eDNA with DNA binding proteins (DNABII), such as histone-like proteins (HU) and integration host factor (IHF) proteins, was implicated in the stability of the biofilm matrix of uropathogenic *E. coli* (UPEC) strain UT189[13]. More recently, Holliday junction recombination intermediates were suggested to enable eDNA to act as a structural component in the biofilm matrix by forming cruciform structures[14]. The positively charged Pel exopolysaccharide cross-links with eDNA through ionic interactions in the biofilm stalk of *P. aeruginosa*[15]. We additionally demonstrated that eDNA from *P. aeruginosa* biofilms is characterised by its ability to form G-quadruplex DNA structures and that these contribute to the viscoelastic behaviour and networks of extracellular matrix structures[9]. The rare DNA Z-form has also been suggested as a key structural component of the bacterial biofilm matrix of UPEC, *Klebsiella pneumoniae* and non-typeable *Haemophilus influenzae* (NTHI)[16]. Despite these key supramolecular insights in the biofilm matrix, it is not known how or why the eDNA forms Holliday Junctions or G-quadruplex structures, and whether certain genes or DNA sequences are important.

Extracellular RNA (eRNA) is present in high concentrations in eDNA-containing biofilm matrices and may also contribute to eDNA viscoelastic networks[17]. Ribonuclease treatment inhibited eDNA-containing *H. influenzae* (NTHI) biofilm formation[18]. A structural function for eRNA was also described in *S. aureus* biofilms where it was found to associate with eDNA and stabilise the polysaccharide–rich matrix[17]. Nonetheless, the lack of robust eRNA extraction protocols and the molecule's low stability[19] has made it difficult to understand the precise structural role of eRNA and assess whether it could possibly also be associated with eDNA in biofilm matrices.

To improve our understanding of the molecular mechanisms of G-quadruplex formation in biofilms, we sought to resolve the nucleic acid composition of *P. aeruginosa* biofilm matrix eDNA. Such information is required to understand how the eDNA network is assembled, which could help to describe *P. aeruginosa* biofilm matrix formation, the viscoelastic phenotype of *P. aeruginosa* biofilms, and establish a basis for new biofilm control strategies.

## Results and discussion

### eRNA is a fundamental network component of *P. aeruginosa* eDNA biofilm matrix fibres

*P. aeruginosa* biofilm EPS is typically dissolved by raising the pH to >12[20]. Using this method, our model biofilm, as well as the extracellular nucleic acid (NA) gel recovered from the biofilm (i.e., NA gel isolate)[9], were dissolved. Alkali treatment has also been shown to result in the dissolution of *Pseudomonas protegens* and *Pseudomonas putida* biofilms[9]. Total correlation spectroscopy (TOCSY) and $^1$H-$^{13}$C heteronuclear single quantum coherence (HSQC) NMR can identify proton NMR correlations within individual ribose sugars and their proton-carbon single bond correlations, respectively. Using this approach, at pH 12 NA peaks were shown to dominate the solution $^1$H-$^{13}$C HSQC spectra for the alkalinised NA gel isolate. The NA spectrum was resolved as two clusters of sugar proton peaks (C1′-H1′), according to differing correlations to neighbouring carbons (C2′-H1′) (Fig. 1a). The first cluster (rectangles), with C2′ chemical shift values of ∼ 40 ppm, represents deoxyribose (i.e., DNA) and the second cluster (ovals) with C2′ shifts of ∼ 70 ppm, is associated with ribose sugar conformations (i.e., RNA). The broadened form of the deoxyribonucleotide peaks is consistent with high molecular weight (MW) molecules, indicating that the chain structure of eDNA is preserved following alkalinisation. In contrast, the sharpness of the ribonucleotide peaks suggests that the RNA (i.e., eRNA) content in the NA gel isolate has been degraded into small nucleotides.

The full $^1$H-$^{13}$C HSQC spectrum indicated that neither proteins nor polysaccharides were present in the NA gel isolate (Supplementary Fig. 1)[21]. eDNA fibre networked structures were visible in the polysaccharide mutants and the proteinase K digested PAO1 wildtype biofilms (Supplementary Fig. 2a–c). The viscoelastic NA gel isolate from *P. aeruginosa* contained eDNA fibres (Supplementary Fig. 2d) and disappeared upon alkalinisation (Supplementary Fig. 2e). Viscoelastic NA gels were isolated from the Δ*pelA* and Δ*pslF* polysaccharide mutants and the proteinase K treated PAO1 biofilm (Supplementary Fig. 2f), as for the untreated PAO1[9]. These observations further suggest that polysaccharides and proteins are not essential for the fundamental eDNA network structure in *P. aeruginosa* biofilms.

The $^{31}$P NMR spectrum of the NA gel isolate at elevated pH revealed the presence of a mixture of monoesterified (3.4 to 4.1 ppm) and diesterified phosphates (-0.8 to -1.2 ppm), consistent with the coexistence of monoribonucleotides and DNA respectively (Fig. 1b (i)). The $^{31}$P NMR spectrum of the biofilm and NA gel isolate displayed no monoribonucleotide phosphate peaks pre-alkalinisation (Supplementary Fig. 3a). Furthermore, the monoribonucleotide $^1$H peaks could only be resolved in the 1-D $^1$H NMR spectrum of the *P. aeruginosa* biofilm after alkalinisation (Supplementary Fig. 3b). The monoribonucleotides were assigned by $^{31}$P-$^1$H HETCOR (Fig. 1b(ii)), HSQC-TOCSY (Supplementary Fig. 3c), correlation spectroscopy (COSY) (Supplementary Fig. 3d) and heteronuclear multiple bond coherence (HMBC) spectroscopy (Supplementary Figs. 4a, b) to 2′- and 3′-(A, U, G, and C)-monophosphates. We also deduced that the eRNA was purine rich (i.e., 57 mol% A + G) and that the G + C mol% content of 46-50% (Supplementary Table 1) differed from that of the *P. aeruginosa* genome (i.e., 67 mol%)[22]. This is consistent with our previous finding that the eDNA contains non-canonical base pair interactions[9]. The same peaks were also observed in the crude biofilm $^{31}$P spectrum (Fig. 1b(i)).

The monoribonucleotides appearing after alkalinisation of the NA gel isolate were thus derived from chain RNA structures that exist at biological pH, indicating that the eRNA has been transesterified. In contrast, the eDNA remained intact after alkalinisation. The tendency of RNA, but not DNA, to undergo transesterification upon mild

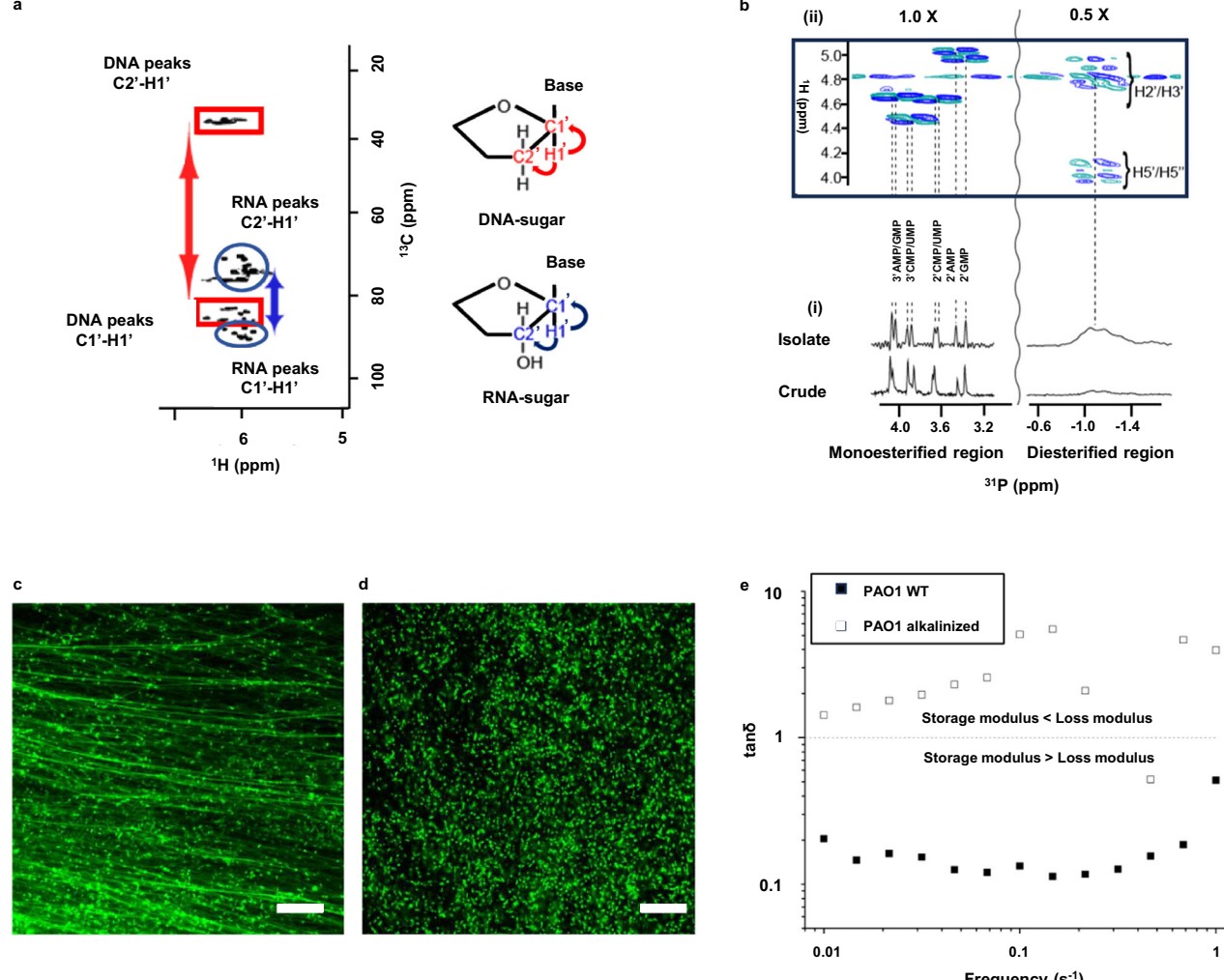

**Fig. 1 | Hydrolysis of RNA and not DNA results in loss of eDNA fibres and *P. aeruginosa* biofilm dissolution.** Nuclear magnetic resonance (NMR) of extracted nucleic acid (NA) gel isolate and confocal laser scanning microscopy of *Pseudomonas aeruginosa* biofilms. **a** NMR $^1$H-$^{13}$C heteronuclear single quantum (HSQC)-total correlation spectroscopy (TOCSY) spectrum of extracellular nucleic acid (NA) gel isolate dissolved in 0.1 M NaOD, (10 mg ml$^{-1}$) at 25 °C showing the C1'-H1' cross peaks of RNA (blue ovals) and DNA (red rectangles) and their correlations to the neighbouring carbon C2'-H1'. Molecule structures (right) illustrate correlations. **b** 1-D $^{31}$P NMR of NA isolate gel and crude *P. aeruginosa* biofilm with proton decoupling showing the presence of monoesterified (i.e., monoribonucleotides) and diesterified (i.e., DNA) phosphate peaks (i), and 2-D $^1$H-$^{31}$P heteronuclear correlation (HETCOR) spectrum of extracellular NA showing the $^{31}$P-$^1$H cross-peaks of monoribonucleotides and DNA (ii). Coupling of monoesterified phosphates to H2' or H3' of eight monoesterified monoribonucleotides (from left to right: 3' AMP/3' GMP, 3' CMP/3' UMP, 2' CMP/2' UMP, 2' AMP, 2' GMP); and diesterified phosphate to DNA H3' and H5'/H5" protons, are denoted by the dashed lines. There is a discontinuity (vertical wavy line) in the $^{31}$P axis due to the different thresholds required to illustrate the $^{31}$P-$^1$H correlations in the monoesterified and diesterified regions. All samples were prepared in 0.1 M NaOD, (10 mg ml$^{-1}$) at 25 °C. **c** Confocal micrographs of five-day *P. aeruginosa biofilms* at pH 7 and **d** following alkaline transesterification (37 °C for 16 h; *n* = 3), stained with TOTO-1 showing eDNA fibres (green). Scale bars represent 10 μm. **e** Rheogram of five-day *Pseudomonas aeruginosa* PAO1 wildtype biofilm and alkalinised wildtype biofilm (0.3 M KOH at 37 °C for 16 h; *n* = 3) in frequency sweep at 25 °C, 0.026 mm gap, 0.3 amplitude. A tan δ > 1 represents fluid behaviour, while tan δ < 1 indicates gel behaviour. For **e**, the biological triplicates are averaged for both conditions and plotted against frequency. Relevant source data for Fig. 1e is provided as a source data file.

alkalinisation is well understood[23] and appears to also occur for biofilm eRNA and eDNA.

Transesterification of eRNA in *P. aeruginosa* biofilms additionally resulted in complete loss of eDNA fibres (Fig. 1c, d), with the 3-D image of the entire biofilm (13 μm thick) showing that eDNA fibres disappeared throughout the biofilm only following RNA transesterification (Supplementary Fig. 5a, b). Alkalinisation increased tan δ in the rheology frequency sweep for the biofilm from 0.2 to greater than 1 (Fig. 1e), indicating that the loss modulus subsequently exceeded the storage modulus, and that removal of eDNA fibres is coincident with loss of viscoelasticity. The same response upon alkalinisation was also observed for five-day *P. protogens* and *P. putida* biofilms (Supplementary Fig. 6a, b). Furthermore, the biofilm displayed viscoelastic properties from days 1-5 of growth (Supplementary Fig. 6c), with the number of dead cells increasing from 8 to 68% from days 2 to 5, suggesting that the viscoelastic behaviour was independent of the number of dead cells (Supplementary Fig. 7a–e). This loss of fibres and biofilm viscoelasticity therefore occurs coincident with eRNA transesterification and despite preservation of DNA chain structures following alkalinisation (Fig. 1a–e).

### *P. aeruginosa* biofilm matrix eDNA is complexed to specific highly expressed RNA transcripts

The NA gel isolate consisted of nucleic acids 2000-10000 base pairs (bp) in length (Supplementary Fig. 8a(i), lane 2). Heating to 55 °C was shown in our earlier studies to disrupt the network eDNA structure[9]

and here slightly reduced NA size to 1500-5000 bp (Supplementary Fig. 8a(i), lane 3). RNase A treatment also reduced NA size to a similar degree (Supplementary Fig. 8a(i), lane 4). DNase I treatment, on the other hand, reduced the NAs to 400-1000 bp (Supplementary Fig. 8a(i), lane 5) and subsequent RNase A treatment further reduced them to 100 bp (Supplementary Fig. 8a(i), lane 6). Thus, the extracellular nucleic acid gel isolate from *P. aeruginosa biofilms* was additionally sensitised to RNase A treatment after DNase I treatment, which is consistent with eRNA being complexed to eDNA in the reconstituted extracellular NA gel extract.

To identify whether the viscoelastic behaviour of nucleic acids in *P. aeruginosa* biofilms could be explained by a distinct RNA expression profile, RNA expression levels were compared between the matrix from a five-day *P. aeruginosa* biofilm with planktonic *P. aeruginosa*.

Assessing established biofilm cells with 16 h planktonic cells (i.e., cells sampled prior to their onset of EPS production) provided the means to ascertain the level of RNA produced in the different phases, that is, biofilm versus planktonic, without the interference of aggregate formation. Expression levels in the matrix were considered stable, based on their consistent mechanical stability and viscoelasticity (Supplementary Fig. 6c). Evaluating eRNA as described above revealed several mRNA transcripts that were highly enriched in both planktonic cells and the biofilm matrix cells, as indicated by their high base mean average (Fig. 2a, Supplementary Fig. 8b, Supplementary Table 2b). Notably, these include *ssrA*, a transfer messenger RNA involved in the trans-translation process for protein stability under stress, which is often detected in high abundance in biofilm transcriptomics studies[24,25], *crcZ*, a regulatory mRNA involved in carbon catabolite

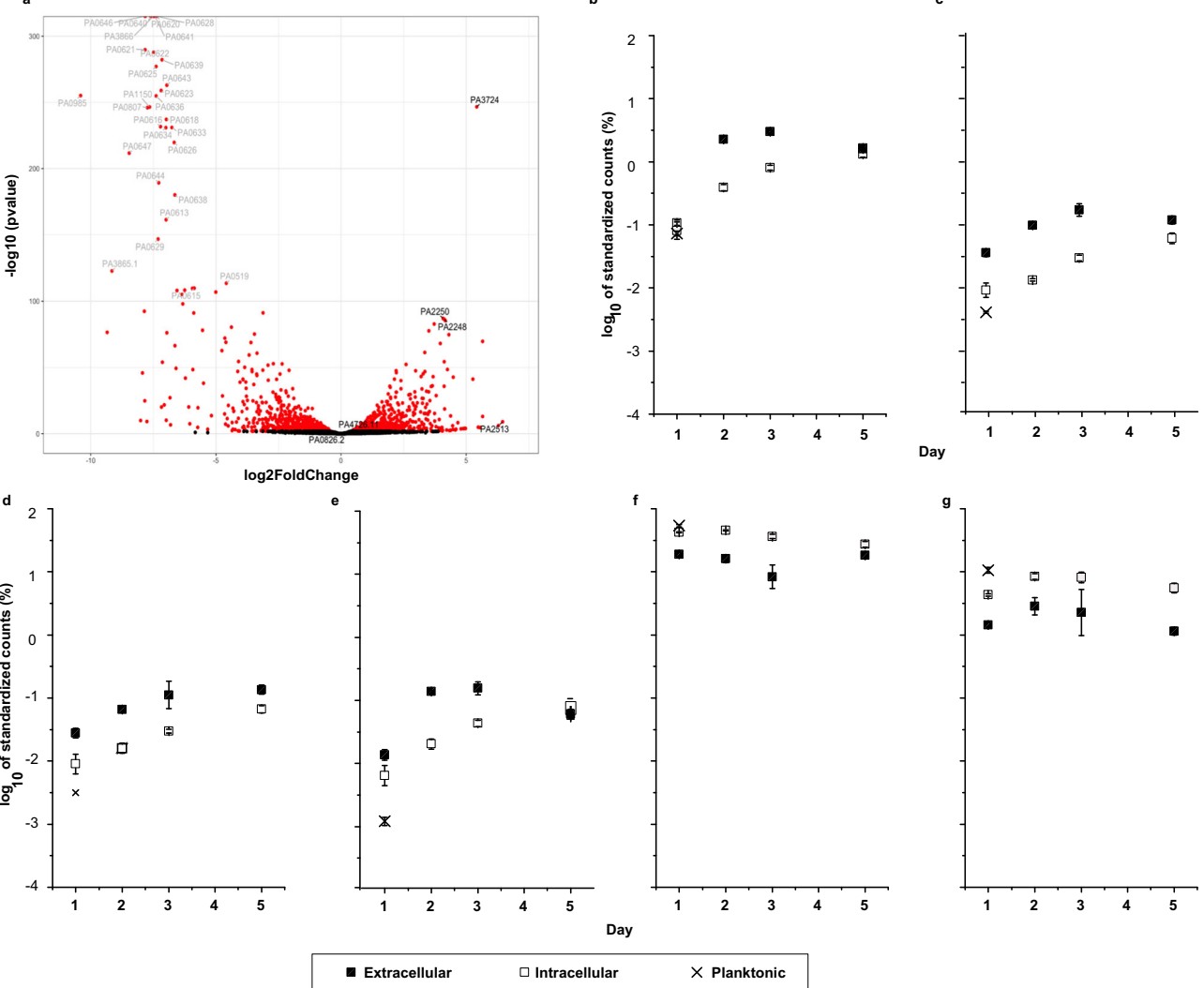

**Fig. 2 | The abundance of specific mRNA transcripts increases in the extracellular matrix of *P. aeruginosa* biofilms relative to planktonic cells.** Extracellular mRNA transcripts identified through RNA analysis of five-day *P. aeruginosa* biofilms. **a** Volcano plot of total RNA sequenced from the biofilm matrix of five-day *P. aeruginosa* biofilms and planktonic *P. aeruginosa* cells (16 h). mRNA transcripts that are highly abundant in planktonic cells compared to the biofilm matrix are highlighted in grey in the left part of the image with a negative log 2-fold change. mRNA transcripts with a positive log 2-fold change and higher negative log10 p-value such as PA3724 (*lasB*), PA2250 (*lpdV*), PA2248 (*bkDA-2*) and PA2513 (*antB*), are more abundant in the extracellular biofilm matrix and are highlighted in bold in the right-hand part of the image. A higher negative log 10 *p*-value indicates a higher probability of the presence of

that particular mRNA transcript in either planktonic cells or in the biofilm matrix of *P. aeruginosa*. **b**–**g** Standardised gene counts (%) of highly abundant mRNA transcripts (**b**) *lasB*, (**c**) *bkDA-2*, (**d**) *lpDV*, (**e**) *antB*, (**f**) *ssrA* and (**g**) *crcZ* in extracellular biofilm matrix and intracellular biofilm cells across different days of biofilm growth (days 1-5). All extractions and sequencing were performed in biological triplicates. The standard deviation bars indicated in Fig. 2b–g are generated based on the mean values calculated from biological triplicates. The negative binomial model was fitted to estimate size factors and dispersion. Significance was determined using DeSeq2 to identify differentially expressed genes which incorporates two two-tailed Wald test with a $P < 0.01$. Relevant source data for Fig. 2a–g have been deposited to public repository and can be found in the data availability section.

repression in *P. aeruginosa*[26], *rnpB*, an RNA component of RNaseP that regulates tRNA maturation[27], and three probable bacteriophage proteins (PA0622, PA0620, PA0641) (Fig. 2a and Supplementary Table 2b).

Highly expressed mRNA transcripts with a positive log 2-fold change in the biofilm matrix of *P. aeruginosa* relative to the planktonic cell samples were also observed (Fig. 2a; bold text). The most highly expressed transcripts in the biofilm matrix relative to planktonic cells, with a minimum base mean average of 30, are presented in Supplementary Table 2a. Elastase, encoded by *lasB* (PA3724), promotes *P. aeruginosa* biofilm formation through rhamnolipid production[28] and is a major virulence factor in *P. aeruginosa* biofilm infections, where it degrades host elastin for nutrients[29]. *lasB* mRNA was the most highly enriched mRNA (Fig. 2a; highest base mean and second highest log 2-fold change) in the biofilm matrix on day 5. *IpdV* (lipidoamide dehydrogenase) and *bkDA-2* (2-oxoisovalerate dehydrogenase beta subunit) mRNAs were also highly abundant in the biofilm matrix. These belong to the same operon and are involved in cellular amino acid metabolic processes and secondary metabolite production[30]. Finally, *antB* mRNA, which encodes anthranilate dioxygenase for anthranilate degradation and mature biofilm formation in *P. aeruginosa*[30], was also abundant in the biofilm matrix.

Conversely, several mRNAs with a high base mean average encoding hypothetical proteins and a set of pyocin mRNAs (a bacteriocin released by *P. aeruginosa*), such as soluble S2, S4 and pore forming S5 pyocins, were more highly expressed in planktonic cells, as indicated by a log 2-fold change <1 (Supplementary Table 2b).

To further investigate whether eRNA might be predicted from gene counts in the biofilm matrix, RNA from the biofilm matrix and its embedded cells (i.e biofilm cells) were sequenced on days 1, 2, 3 and 5 of biofilm growth (Fig. 2b–g). For putative extracellular mRNA *lasB*, *bkDA-2*, *IpDV* and *antB*, gene counts standardised against total reads per sample were higher in the biofilm matrix than in biofilm cells and planktonic cells throughout the biofilm growth assay. These extracellular mRNA are therefore enriched in the biofilm matrix compared to its abundance in both planktonic and biofilm cells. *ssrA* and *crcZ* transcripts, on the other hand, were higher in the biofilm cells than in the biofilm matrix, yet present in similar abundances within the biofilm cells and planktonic cells (Fig. 2f, g). The difference between eRNA levels in the biofilm matrix and its cells became increasingly less pronounced from days 2 to 5. This correlates with increased cell death and lysis with time (Supplementary Fig. 7) and is likely a consequence of increased iRNA solubility in the matrix. Nonetheless, while expression levels of putative iRNA and eRNA in the matrix varied from days 2 to 5, the relative proportions of iRNA and eRNA remained constant. Greater amounts of eRNA were detected in the biofilm matrix compared to those in association with planktonic cells, despite the age of the biofilm. Further, the iRNA was consistently lower in proportion in the biofilm matrix than the planktonic cells. Thus, this approach for assessing iRNA and eRNA associated with microbial cells is valid.

## mRNA transcripts colocalise with the eDNA network in the extracellular matrix of *P. aeruginosa* biofilms

Single molecule inexpensive fluorescent in situ hybridisation (smiFISH) confocal microscopy was performed for highly abundant biofilm matrix RNA (i.e., relative to planktonic cells) using gene specific primary probes pre-hybridised to a FLAP sequence labelled with two fluorophores. Primary probes were designed to target specific regions of these extracellular mRNAs (Supplementary Table 2a). Without DNase I pre-treatment, the most highly enriched mRNA, *lasB*, was not detected in the biofilm matrix when probed using 13 *lasB*, RNA-specific oligoribonucleotide primary probes (Fig. 3a). As shown in Supplementary Fig. 8a, nucleic acids are sensitised to RNase A digestion only after DNase I treatment. Hence, smiFISH was repeated after mild pre-treatment with 0.02, 0.05 and 0.1 mg ml⁻¹ DNase I. This revealed a

DNase I concentration-dependent binding of the *lasB* RNA probe (red channel) to eDNA fibres (green channel). No red fluorescent *lasB* signal was observed in the eDNA fibre after washing when only fluorescently labelled secondary probes were used (i.e., without the primary probe as a negative control), even following mild DNase I pre-treatment to 0.1 mg ml⁻¹ DNase I (Supplementary Fig. 9a). Furthermore, smiFISH performed on *lasB* knockout mutants with the *lasB*-specific oligo probes showed no fluorescent signal in the biofilm extracellular matrix (Supplementary Fig. 9b, c). These results suggest that the observed *lasB* signal resulted from the primary probe binding to *lasB* mRNA colocalising with eDNA fibres in the *P. aeruginosa* biofilm.

At the lowest DNase I concentration (i.e., 0.02 mg ml⁻¹; Fig. 3b), a weak, dispersed *lasB* RNA signal was observed on the eDNA fibres (seen in merge channel). Following pre-treatment with 0.05 mg DNase I ml⁻¹ (Fig. 3c), the signal from the *lasB*-RNA specific oligoribonucleotide on the eDNA fibres increased. While the signal resulting from the *lasB* oligoribonucleotide-specific on the eDNA fibres was also high at 0.1 mg ml⁻¹ DNase I (Fig. 3d), the eDNA fibre signal was fainter, more fragmented and less networked, possibly indicating that eDNA hydrolysis occurred at the higher DNase I concentration. Nonetheless, the eDNA fibre structures were preserved even after prolonged incubation with 0.1 mg ml⁻¹ DNase I (16 h) (Supplementary Fig. 10(ii)). The 3-D transect of the 0.1 mg ml⁻¹ DNase I treated biofilm (Fig. 3e) shows that the *lasB* binding to the eDNA fibres is distributed evenly throughout the biofilm (Mander's coefficient 38 ± 4.5%, *n* = 4 images). The increasing intensity of the *lasB* RNA-specific oligoribonucleotide probe upon DNase I treatment, where *lasB* intensity increases five-fold upon DNase I pre-treatment (Fig. 3f), suggests that higher concentrations of DNase I enzyme (0.1 mg ml⁻¹) improve the accessibility of the *lasB* mRNA probe to eRNA.

Colocalisation analysis of *lasB* mRNA with eDNA fibre under DNase I concentration was performed and the colocalised regions presented in yellow (Fig. 3a–d, colocalised region channel). Mander's coefficients for pre-treatment with 0, 0.01, 0.05 and 0.1 mg ml⁻¹ DNase1 were 0%, 8.6 ± 2%, 29 ± 7% and 49 ± 1%, respectively (Fig. 3g), further indicating the DNase I concentration-dependent binding of the *lasB* oligoribonucleotide probes to the eDNA fibres.

While there are many methods for visualising eDNA in biofilms, this is the first description of a method enabling the distribution of particular nucleic acid sequences to be determined. This has led to the observation that eRNA transcripts can colocalise with eDNA fibres in the biofilm matrix. However, despite the observation that the eRNA co-localises with eDNA fibres, and that eRNA transesterification upon alkalinisation occurs coincident with biofilm and matrix dissolution, only an 18% reduction in eDNA fibre fluorescence intensity was observed in five-day *P. aeruginosa* biofilms following 0.3 mg ml⁻¹ RNase A treatment, which was not enough to break down the eDNA network (Supplementary Fig. 10(iv)).

smiFISH was also applied to the remaining top 10 most upregulated mRNAs in the biofilm matrix (Supplementary Table 2a). Colocalisation of mRNA with eDNA fibres after pretreatment with mild DNase I was also observed and quantified for oligoribonucleotide probes specific for *IpdV* (Fig. 3h, top left image), *antB* (Fig. 3h, top middle image) and *bkDA-2* (Fig. 3h, top right image)[31,32]. The colocalisation analysis for each of the three mRNA with eDNA fibre is visualised in yellow (Fig. 3h, bottom images).

Of the abovementioned mRNA, *bkDA-2* was the most highly enriched in five-day *P. aeruginosa* biofilms followed by *antB* and *IpDV* (Fig. 3i). No signal was observed from scrambled probes of extracellular mRNA such as *lasB, IpDV, bkDA-2* and *antB* in eDNA fibres of *P. aeruginosa* biofilms (Supplementary Fig. 11).

The same analysis was then performed using *P. aeruginosa*-specific conventional 16 S and 23 S ribosomal RNA probes on biofilm pretreated with 0.1 mg ml⁻¹ DNase I, at both 37 °C (Supplementary Fig. 12a(i) and 12b(i)) and 46 °C (Supplementary Fig. 12a(ii) and 12b(ii)).

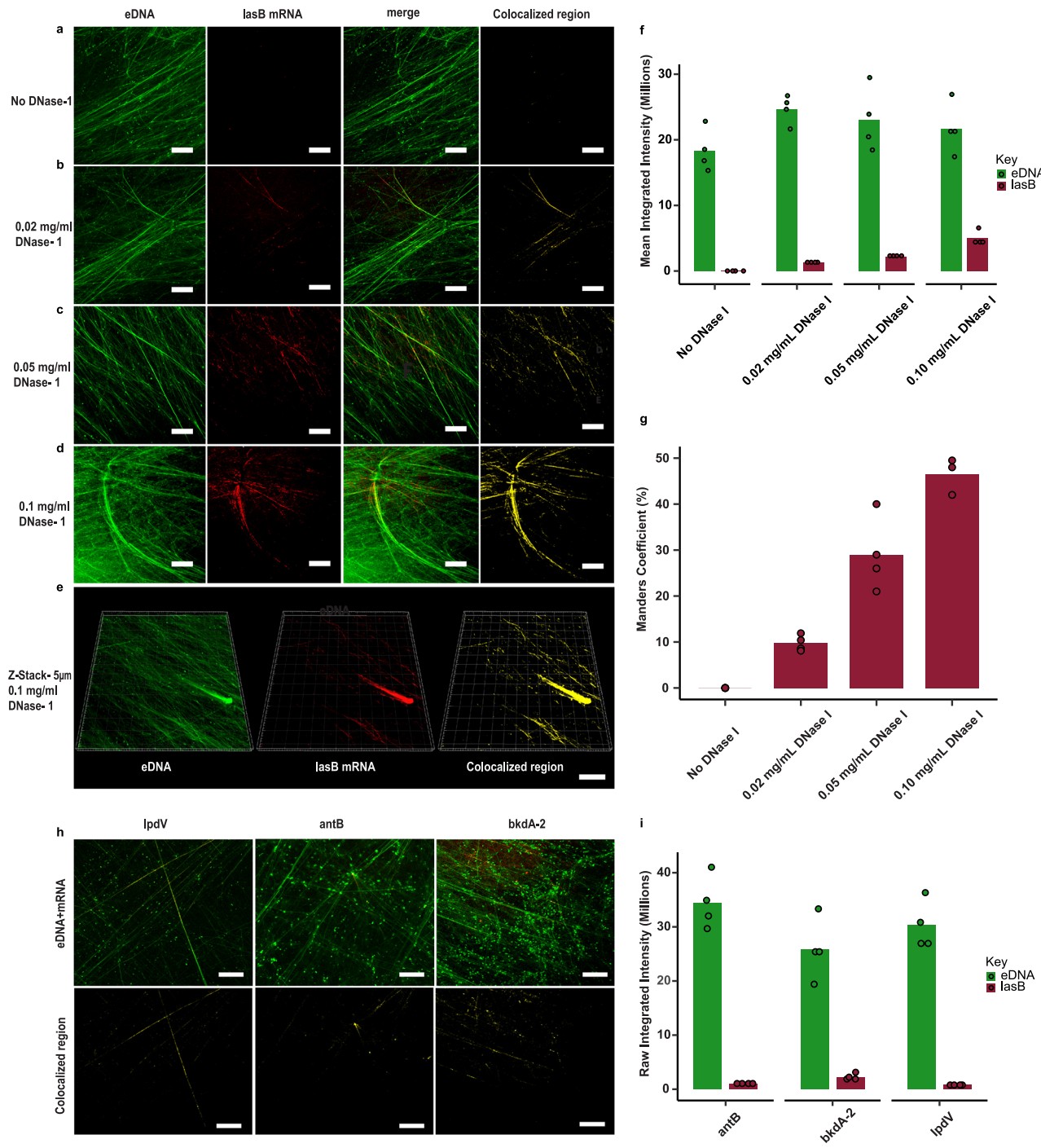

Colocalisation analysis of 16 S rRNA and 23 S rRNA to cells showed Mander's coefficients of about 66% and 76% for 16 S rRNA at 37 °C and 46 °C, respectively, and 62% and 60.4% for 23 S rRNA at 37 °C and 46 °C, respectively (Supplementary Fig. 12a (i),(ii) and 12b (i),(ii)). Additionally, both 16 S and 23 S rRNA were targeted by smiFISH using multiple probes to increase the sensitivity of detection (Supplementary Fig. 12a(iii) and 12b(iii)). The lower hybridisation temperature compared to conventional FISH precludes staining of genomic DNA by 16 S and 23 S probes following smiFISH[33]. The absence of both 16 S and 23 S ribosomal RNA signals in the eDNA fibres, along with the negative log 2-fold change of 23 S rRNA seen in Supplementary Fig. 12a(iii) and 12b (iii)), supports our observation from total RNA sequencing that mRNA, but not ribosomal RNA, is dominant in the extracellular biofilm

matrix of the *P. aeruginosa*. The 16 S rRNA and 23 S rRNA probes used here are included in Supplementary Table 3.

### Dominant eRNA transcripts complexed with eDNA fibres change over time and are independent of total RNA expression

The amount of *lasB* mRNA, associated with eDNA fibres was determined for each day of biofilm growth. No *lasB* mRNA was detected in the fibres for the early stages of biofilm growth i.e., days 1 (Supplementary Fig. 13) or 2 (Fig. 4a, b). *lasB* RNA only became visible in the biofilm after the matrix was established on day 3 (Fig. 4a, b) and its expression was maintained across days 4 and 5 (Fig. 4a, b). There was a 5% increase in *lasB* mRNA detection levels on day 3 compared to days 1 and 2 and its level increased until day 4 before reducing

**Fig. 3 | mRNA is detected in eDNA fibres upon mild DNase I pretreatment.**
Extracellular *lasB* mRNA visualisation using the smiFISH method. smiFISH confocal micrographs of five-day *P. aeruginosa* biofilms showing *lasB* mRNA smiFISH probes (red), eDNA specific TOTO-1 (green), merged image i.e (red + green) and colocalised region (yellow): **a** without DNase I treatment showing no affinity of *lasB* probe for *lasB* mRNA in the eDNA fibre, **b** treated with 0.02 mg ml⁻¹ DNase I indicating weak dispersed *lasB* RNA signal on the eDNA fibre, and **c** treated with 0.05 mg ml⁻¹ and **d** 0.1 mg ml⁻¹ of DNase I respectively showing increased *lasB* RNA signal intensity (yellow; c and d colocalised region panel). Scale bars represent 10 μm. **e** Three-dimensional (3-D) confocal micrograph of five-day *P. aeruginosa* biofilm (Z-stack = 5 μm) showing overall spatial distribution of *lasB* mRNA (red) and their colocalisation (yellow) with TOTO-1 stained eDNA fibre (green) (*n* = 4). Scale bars represent 10 μm. **f** Fluorescence intensity quantification (*n* = 4) of eDNA (green bars) and *lasB* mRNA (red bars) expression in five-day *P. aeruginosa* biofilms treated with 0, 0.02, 0.05 and 0.1 mg ml⁻¹ DNase I. **g** Colocalisation analysis (*n* = 4) showing

Mander's coefficients for eDNA and *lasB* mRNA colocalisation in five-day *P. aeruginosa* PAO1 wildtype biofilms at 0, 0.02, 0.05 and 0.1 mg ml⁻¹ DNase I. **h** Confocal micrograph of DNase I pre-treated (0.1 mg ml⁻¹) five-day *P. aeruginosa* biofilms performed using smiFISH probes specific *for IpdV* (27 primary probes), *antB* (13 primary probes), and *bkDA-2* (8 primary probes) mRNA (red), which colocalise with TOTO-1 (green) stained eDNA fibres (reddish yellow streaks in top panel of merged images). Mander's coefficients for *Ipdv, antB,* and *bkDA-2* of 16 ± 5%, 11 ± 3%, and 18 ± 6%, respectively (*n* = 5 images each), were determined based on the overlap of the mRNA signal with TOTO−1 stained (green) eDNA fibres (colocalised region panel at bottom; yellow). Scale bars represent 10 μm. **i** smiFISH confocal micrograph fluorescence intensity quantification (*n* = 5) of eDNA (green bars) and highly enriched extracellular mRNA *bkDA-2, IpDV* and *antB* (red bars) in five-day *P. aeruginosa* biofilms. The standard deviation bars indicated in Fig. 3f, g, i are generated based on the mean values calculated from biological triplicates. Relevant source data for Fig. 3f, g and i are provided as a source data file.

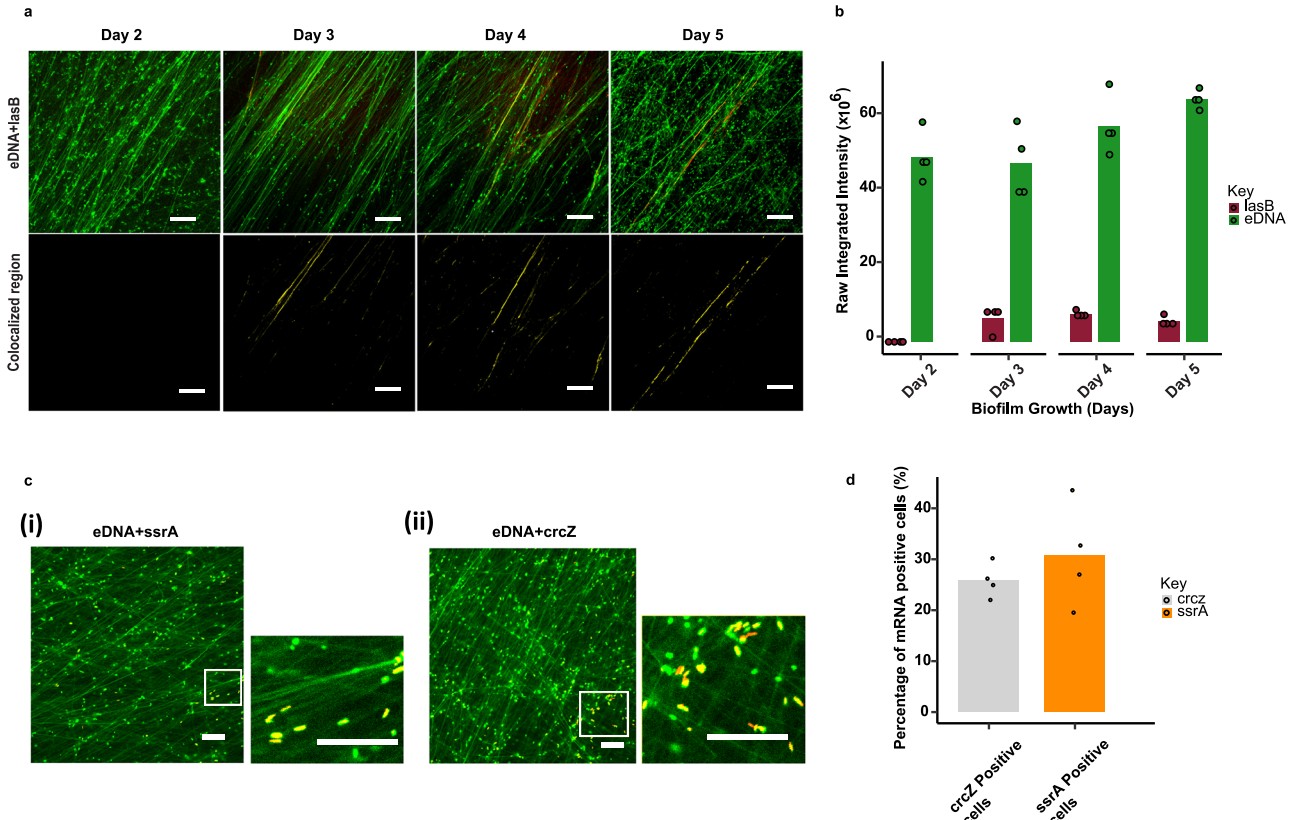

**Fig. 4 | *lasB* not required for eDNA fibres, and abundant iRNA are not seen in eDNA fibres. a** smiFISH confocal micrograph of DNase I pre-treated (0.1 mg ml⁻¹) *P. aeruginosa* biofilms stained with *lasB* mRNA smiFISH probes (red) after two, three, four, and five days of growth. eDNA fibre was visualised with TOTO−1 (green) and colocalised region (yellow). Mander's coefficients for days 1, 2, 3, 4, and 5 biofilms were zero, zero, 14 ± 2%, 22 ± 4%, and 24 ± 8% respectively (*n* = 5 images each) based on the overlap of the mRNA signal with TOTO−1 stained (green) eDNA fibres (colocalised region panels; yellow). **b** smiFISH confocal micrograph fluorescence intensity quantification (*n* = 4) of extracellular DNA and *lasB* mRNA expression across different days (days 2 to 5). Green bars represent eDNA and red bars represent *lasB* mRNA. **c** Confocal micrograph of DNase I (0.1 mg ml⁻¹) pre-treated

five-day *P. aeruginosa* biofilms (*n* = 4) stained with 10 and six smiFISH primary probes specific for transfer messenger RNA (i) (tmRNA) *ssrA* and (ii) *crcZ* respectively showing 25% of cells with *crcZ* and 30% with *ssrA*. Cells staining positive for *ssrA* and *crcZ* RNA are seen in yellow (Fig. 4c(i),(ii) ROI, respectively). ROI indicates zoom-in of region of interest shown in small white square boxes. **d** smiFISH confocal micrograph fluorescence intensity quantification (*n* = 4) of *crcZ* (25 %) and *ssrA* (30%) positive cells in five-day *P. aeruginosa* biofilms using maxima function in ImageJ Fiji software. Scale bars represent 10 μm. The standard deviation bars indicated in 4b and d are generated based on the mean values calculated from biological triplicates. Relevant source data for Fig. 4b, d are provided as a source data file.

slightly on day 5 (Fig. 4b). This reduction in lasB from days 4 to 5 was also observed in the mRNA count for the biofilm (Fig. 2b). Further, eDNA levels consistently increased across these days. This corroborates the increase in *lasB* mRNA gene counts detected in the *P. aeruginosa* biofilm matrix from days 1 to 3 (Fig. 2b). *lasB* mRNA binding is presented here, as it displayed the highest log 2-fold increase in the

biofilm matrix relative to planktonic cells, coupled with high total expression (Fig. 2a). However, despite being the most abundant eRNA in the five-day biofilm, the *lasB* knockout mutants still produced a biofilm under the same growth conditions and the mRNA transcript profile of the *lasB* knockout biofilm was different to the wildtype biofilm (Supplementary Fig. 14a).

Thus, *lasB* is not deemed essential for *P. aeruginosa* biofilm formation and there is likely a redundancy with regards to the mRNA that *P. aeruginosa* employs in eDNA fibre assembly. Accordingly, *lasB* does not completely cover the eDNA fibres (Fig. 3a–e) and there is a possibility that other mRNAs contribute to a greater extent in eDNA fibre formation than *lasB* during earlier stages of *P. aeruginosa* biofilm growth.

RNA seq data identified transcripts with a high total abundance in both planktonic and biofilm matrix samples (Supplementary Table 2b), most notably transfer messenger RNA (*ssrA*), which was the most highly expressed mRNA across both samples. smiFISH confocal microscopy performed on five-day biofilms using 10 and six oligo primary probes targeting *ssrA* and *crcZ* mRNA respectively, showed about 30% and 25% of *ssrA* and *crcZ* mRNAs positive cells within the biofilms (yellow; Fig. 4c (i) ROI and 4c (ii) ROI, respectively, and 4d). However, neither *ssrA* nor *crcZ* mRNA associated with the eDNA fibres, which is consistent with their low abundance in the biofilm matrix relative to the biofilm cells. This further suggests some contamination of iRNA in the biofilm matrix. Nonetheless, the observations by smiFISH confocal microscopy of highly expressed biofilm matrix RNA along the eDNA fibres, and highly expressed biofilm cell RNA inside cells, validate the methods described for identifying eRNA. The dispersal of iRNA throughout the biofilm likely contributes to the inability to detect it in the biofilm matrix, which contributes to lower fluorescence signal, in contrast to the eRNA, which is concentrated in the eDNA fibres. Furthermore, the absence of the transcripts of *ssrA* and *crcZ* in the biofilm matrix, indicates that the specific mRNA transcripts appearing in the matrix do not simply reflect overall cellular RNA expression. The release of RNA might therefore be a regulated, rather than random, process.

### *lasB* mRNA colocalises with eDNA fibres in airway specimens colonised with *P. aeruginosa*

In a clinical bronchiectasis airway sputum specimen with a 96% abundance of *P. aeruginosa relative* to total microbial reads, colocalisation of *lasB* mRNA (Fig. 5a, merged image of eDNA and *lasB*) was clearly observed along eDNA fibres following pre-treatment with 0.1 mg ml$^{-1}$ DNase I, as per the in vitro *P. aeruginosa* biofilm model. While some *lasB* mRNA could be detected without DNase I treatment, fluorescence intensity increased more than ten-fold following pre-treatment with DNase I in sputum samples (Fig. 5b), with a final Mander's coefficient of colocalisation of 56.72% (Fig. 5a, colocalised region panel; yellow). eDNA fibres were observed in two of the three clinical sputum samples (HP0005, HP0007) that were positive for *P. aeruginosa*, and *lasB* mRNA was seen only when eDNA fibres were present in the sample. Three clinical sputum samples that were positive for *P. aeruginosa* were fully degraded at a higher DNase I concentration (0.4 mg ml$^{-1}$ DNase I, Fig. 5c), demonstrating that eDNA fibres also modify sputum viscoelasticity in infected hosts. The *lasB*-specific probe did not bind the DNase I digested (0.4 mg ml$^{-1}$) sputum sample (Fig. 5d). No *lasB* fluorescence was observed by smiFISH confocal microscopy performed on a human clinical sputum sample without *P. aeruginosa* infection (Fig. 5e). Additionally, no eDNA fibres were observed in the non-infected sample (Nano0015) (Supplementary Fig. 5e), and *lasB* can only be observed when eDNA fibres were present in the sputum sample (Fig. 5a).

The *lasB* scrambled probes did not give a fluorescence signal in the *P. aeruginosa* -positive sputum sample (Fig. 5f). An in silico binding affinity study revealed a far weaker specificity and binding affinity for the designed *lasB* probe to the human elastase gene (Supplementary Table 4)[34]. This indicates that the designed *lasB* probed was not binding to human elastase mRNA.

The identification of an mRNA colocalising with eDNA in a human clinical sputum sample highlights the clinical relevance of eRNA in biofilms. Our findings can hence be extrapolated to clinical settings where biofilms possess a decreased susceptibility to antimicrobials due to the protective environment of the matrix, creating a major health risk[35].

### Extracellular RNA is key to viscoelasticity of *Pseudomonas* biofilms

There was a reduction in DNA length of approximately 8000 bp following alkaline transesterification of the NA gel isolate (Supplementary Fig. 14b), indicating some loss of mass or secondary structure. To assess directly whether eRNA contributes to *P. aeruginosa* biofilm viscoelasticity, 0.3 mg ml$^{-1}$ of RNase A digestion of the *P. aeruginosa* biofilm was coupled with mild pre-DNase I treatment (0.1 mg ml$^{-1}$), as per the smiFISH microscopy protocol. This increased the tan δ for the biofilm, relative to the untreated biofilm, in the rheology frequency sweep and reduced eDNA fibre fluorescence intensity by 47% (Fig. 6a–e;, Supplementary Fig. 10), although tan δ remained below 1. This is consistent with the observation that the viscoelastic network was preserved. However, when RNase A was replaced with 0.3 mg ml$^{-1}$ of RNase H, targeting the RNA strand of RNA:DNA hybrids, the model *P. aeruginosa* biofilm dissolved and lost its viscoelasticity, as indicated by the increase in tan δ to greater than 1 and 73% reduction in eDNA fibre fluorescence intensity (Fig. 6d, e). Without mild pre-DNase I treatment and 0.3 mg ml$^{-1}$ of RNase A and RNase H, there was a slight increase in tan δ, and 18, and 51% reductions in eDNA fibre fluorescence intensity, respectively. This demonstrates that the viscoelastic networks were preserved (Fig. 6d, e). Mild DNase I treatment (0.1 mg ml$^{-1}$) with RNase H buffer alone additionally did not dissolve the biofilm and resulted in less than 4% loss of eDNA fibre fluorescence intensity (Fig. 6d, e).

The same effect of RNase H digestion with mild DNase I pre-treatment was observed for the NA gel isolate, as well as *P. putida* and *P. protegens* biofilms (Supplementary Figs. 8a(ii) and 14c, respectively). These observations further indicate that eRNA is an integral part of eDNA fibre formation in the biofilm matrix by complexing to eDNA as extracellular RNA:DNA hybrids, and hence an important contributor to *Pseudomonas* biofilm viscoelastic networks. This likely explains why the eRNA was shielded from the oligoribonucleotide smiFISH probes.

### Possible mechanisms of eDNA-eRNA network assembly

This study demonstrates that biofilm dissolution by alkalinisation occurs coincident with eRNA, but not eDNA hydrolysis, and that eRNA contributes to biofilm matrix formation. eDNA, nonetheless, remains a key structural component of *P. aeruginosa* biofilm matrix, as indicated by its sensitivity to DNase I treatment. Furthermore, the concentration of DNA in the viscoelastic nucleic extract was approximately twice that of RNA (Supplementary Table 5). The lower abundance of RNA as well as its high structural versatility suggest that eDNA is the basic building block of the viscoelastic network, and that eRNA facilitates the assembly of eDNA into crosslinked networks. RNA can form non-canonical base pairings[36] and G-quadruplex structures like those we observed in *P. aeruginosa* biofilms[9]. Such structures are known to enable strand exchange between duplexed DNA. One possible explanation, therefore, is that eRNA bound to eDNA could facilitate junctions forming between eDNA strands, enabling them to assemble into networks. One such junction could be the Holliday Junction[14]. Understanding the higher-order structure of key matrix biopolymers such as extracellular nucleic acids has thus provided important insights into *P. aeruginosa* biofilm and matrix formation.

Our study revealed that *lasB* mRNA aligns along the DNA fibres and not the network junctions. The fibres observed (Fig. 3) are 0.3- 0.5 μm thick. Double-stranded DNA is 24 Å, or 0.0024 μm thick. We submit that these nucleic acid fibres are therefore the result of multiple nucleic acid strands bundling or intertwining into a supramolecular assembly. One such example where DNA achieves this is G-wires. These

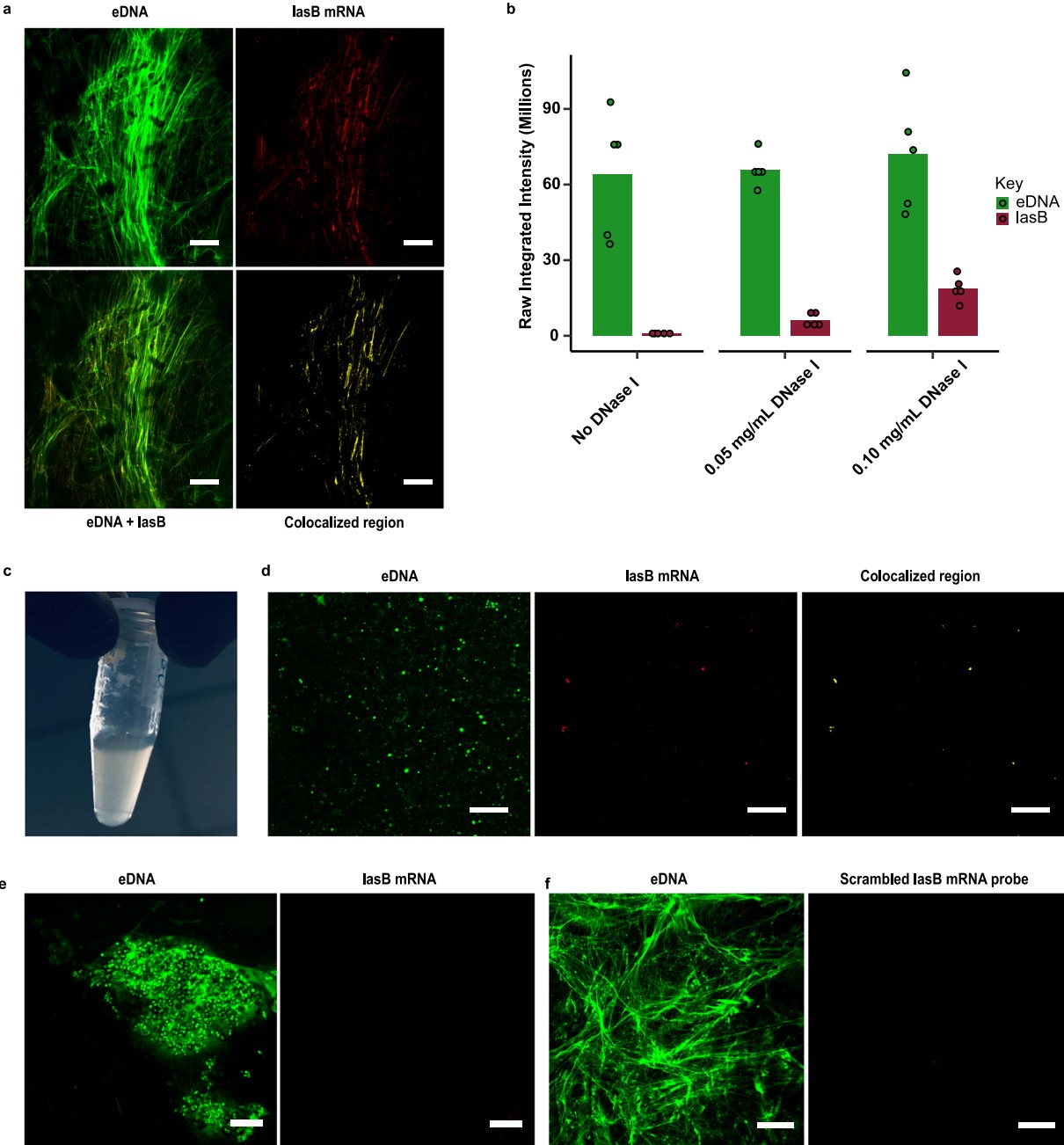

**Fig. 5 | *lasB* mRNA is also present in eDNA fibres of human clinical sputum sample.** smiFISH performed on clinical sputum samples highly enriched with *P. aeruginosa*. **a** Confocal micrograph of 0.1 mg ml⁻¹ DNaseI pretreated human clinical sputum sample HP0005 highly enriched in *P. aeruginosa* showing eDNA fibres (TOTO−1 stain; green), *lasB* mRNA (*lasB* mRNA specific smiFISH probe; red streaks), merged image of *lasB* mRNA and eDNA fibres and colocalised region of eDNA fibres with *lasB* mRNA (yellow). A Mander's coefficient of 56.72 % (*n* = 8 images) was determined for the clinical sample based on the overlap of the *lasB* mRNA signal with TOTO−1 stained (green) eDNA fibres (colocalised region panel; yellow). Scale bars represent 10 μm. **b** Fluorescence intensity quantification (*n* = 4) of extracellular DNA (green bars) and *lasB* mRNA (red bars) expression in human clinical sputum sample positive for *P. aeruginosa* biofilms treated with 0, 0.05, and 0.1 mg ml⁻¹ DNase I. **c** Clinical sputum sample treated with 0.4 mg ml⁻¹ of DNase-I at 37 °C for 1 h showing complete dispersal of sputum and increased solution turbidity **d** smiFISH

confocal micrograph of 0.4 mg ml⁻¹ DNase-I treated human clinical sample at 37 °C for 1 h (*n* = 3), showing eDNA staining with TOTO−1 dye (green) after DNase treatment and *lasB* signal as visualised with *lasB*-specific oligoribonucleotide smFISH probes (red), and colocalisation of eDNA and *lasB* (yellow). **e** smiFISH confocal micrograph (*n* = 4) of clinical sputum sample negative for *P. aeruginosa* after 0.1 mg ml⁻¹ DNase1 pre-treatment showing dead cells (green) and no *lasB* oligoribonucleotide smFISH probes signal (red). **f** smiFISH confocal micrograph (*n* = 4) of clinical sputum sample positive for *P. aeruginosa* after 0.1 mg ml⁻¹ DNase1 pre-treatment following staining with scrambled *lasB*-specific oligoribonucleotide smFISH probe (red), and eDNA specific TOTO−1 stain (green) showing eDNA fibres. The scale bars represent 10 μm. The standard deviation bars indicated in 5b are generated based on the mean values calculated from biological triplicates. Relevant source data for Fig. 5b are provided as a source data file.

---

are stable and long supramolecular structures that form as a result of stacking of G-quadruplex structures[37]. A possible explanation for how these supramolecular structures form, considering the relative abundances and versatilities of DNA and RNA, is that the crosslinking of

eDNA strands by eRNA enables eDNA to form similar types of supramolecular eDNA wires. It is also possible that different types of eRNA perform multiple roles in the extracellular millieu, as is the case for RNA within the cell.

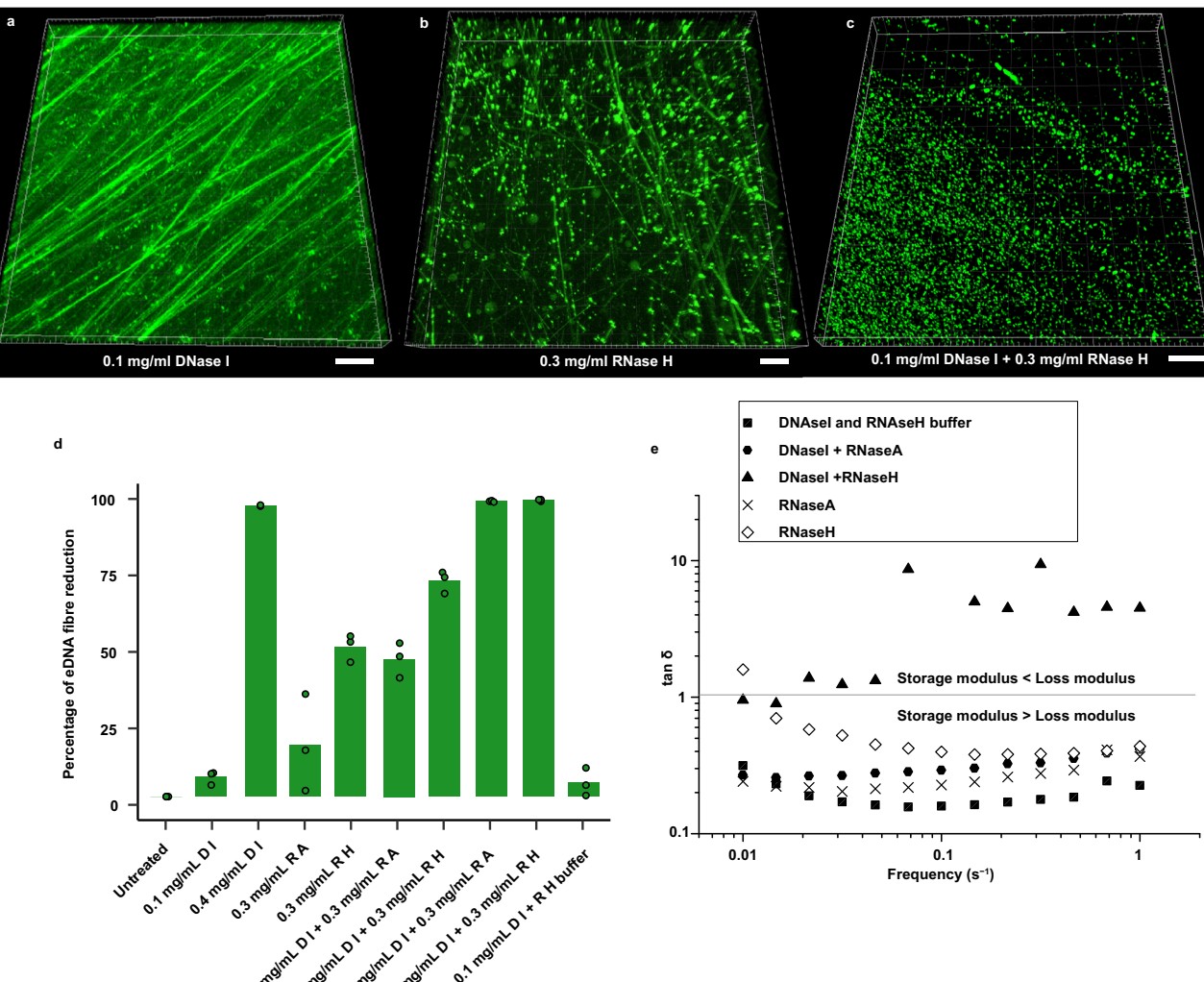

**Fig. 6 | Enzymatic digestion of eRNA leads to loss of eDNA fibres and loss of biofilm viscoelasticity.** Three-dimensional (3-D) confocal micrographs of five-day eDNA specific TOTO-1-stained *P. aeruginosa* PAO1 wildtype biofilm (green) **a** with 0.1 mg ml⁻¹ DNase I pre-treatment and **b** 0.3 mg ml⁻¹ RNase H treatment, **c** 0.1 mg ml⁻¹ DNase I pre-treatment followed by subsequent 0.3 mg ml⁻¹ RNase H digestion. The thickness of biofilm is 8 µm. Scale bars represent 10µm. **d** smiFISH confocal micrograph fluorescence intensity quantification ($n = 3$) of extracellular DNA fibre reduction (green bars) across different enzymatic treatments such as DNase I (D I), RNase A (R A) and RNase H (R H). **e** Rheogram of five-day *Pseudomonas aeruginosa* PAO1 wildtype biofilm ($n = 3$) pre-treated with 0.1 mg ml⁻¹ DNaseI followed by either RNase A or RNase H treatment in frequency sweep at 25 °C, 0.026 mm gap, 0.3 amplitude showing that DNase I pretreatment followed by 0.3 mg ml⁻¹ of both RNase A or RNase H digestion respectively reduces (i.e., increases tan δ) or removes (tan δ > 1) biofilm elasticity. The standard deviation bars indicated in 6d are generated based on the mean values calculated from biological triplicates. For e, the biological triplicates are averaged for each condition and plotted against frequency. Relevant source data for Fig. 6d, e are provided as a source data file.

## Conclusions

eRNA is a key viscoelastic component of *Pseudomonas* biofilms. Biofilm alkalinisation, resulting in eRNA transesterification, therefore degrades a foundation structural material. This could account for *Pseudomonas* biofilm dissolution by alkanisation. mRNA transcripts appear in eDNA fibres, suggesting they have an extracellular function in addition to their role in transcription and translation. The ability to bind to eDNA enables eRNA to act as a key structural biopolymer of *P. aeruginosa* biofilm matrix, which may explain how eDNA forms viscoelastic networks. Specific mRNA sequences, including *lasB*, were enriched in the biofilm matrix of *P. aeruginosa*, and smiFISH enabled these to be detected in eDNA fibres of *P. aeruginosa* biofilms. This finding suggests that the appearance of eRNA in the biofilm matrix is not random, nor solely based on expression levels. While *lasB* appears not to be essential for *P. aeruginosa* biofilms, it is nonetheless a suitable biomarker for observing eRNA in viscoelastic eDNA networks of clinical and model *P. aeruginosa* biofilms. Future studies will focus on

understanding how RNA is exported to the extracellular matrix. This will identify mechanisms of eDNA fibre assembly and biofilm matrix formation.

## Methods

### Bacterial strains and growth conditions

*P. aeruginosa* PAO1 wildtype, *lasB* knockout mutants PW7302, PW7303, polysaccharide mutants Δ*pslF* and Δ*pelA*, as well as *P. putida* and *P. protogens* strains were grown on Luria-Bertani (LB) agar plates (5 g l⁻¹ NaCl, 5 g l⁻¹ yeast extract, 10 g l⁻¹ tryptone, 15 g l⁻¹ agar) or *Pseudomonas* isolation agar (PIA) (20 g l⁻¹ peptone, 10 g l⁻¹ K₂SO₄, 1.4 g l⁻¹ MgCl₂, Irgasan™ 25 mg l⁻¹, agar 13.6 g l⁻¹) at 37 °C for 16 h. For 1D ³¹P, ¹⁵N labelled NH₄Cl-supplemented M9 media was used for biofilm growth. Briefly, M9 minimum media consisted of 9.552 g l⁻¹ Na₂HPO₄.2H₂O, 4.41 g l⁻¹ KH₂PO₄, 1.71 g l⁻¹ NaCl, 1 g l⁻¹ NH₄Cl, 0.24 g l⁻¹ MgSO₄, 0.011 g l⁻¹ CaCl₂, 2 g l⁻¹ casamino acid, and 0.4 g l⁻¹ glucose[9]. Overnight pre-cultures of *P. aeruginosa* strains were grown at 37 °C, 4 x *g* in either

10 ml of LB broth (5 g l⁻¹ NaCl, 5 g l⁻¹ yeast extract, 10 g l⁻¹ tryptone) or 10 ml of M9 minimum media (MPBiomedicals).

## Biofilm growth

For biofilm cultures, 10 ml of respective overnight pre-cultures were incubated overnight, diluted 50 times with fresh LB broth in 2 l Erlenmeyer flasks and incubated for 5 d at 37 °C under static conditions. All biofilm growth experiments were performed with three biological replicates. For extracellular nucleic acid gel isolation, NMR analysis and RNA sequencing, viscoelastic biofilms, shown to contain eDNA as key networking agents[9], were concentrated by centrifugation at 10,621 x *g* for 15 min in 50 ml Falcon tubes. Biofilm pellets were collected and lyophilised. For microscopy and rheology assays, the biofilms were collected directly from the Erlenmeyer flasks by gently aspirating the viscoelastic biofilm edge using a cut-off 1 ml pipette tip to minimise disturbance caused to the biofilms and specifically to preserve the native structure of biofilm matrix.

## Biofilm and nucleic acid gel alkalinisation for NMR, rheology and agarose gel electrophoresis

Solution state NMR experiments were performed on purified NA gel isolate alkalinised with 0.1 M NaoD at 37 °C for 16 h. Sample concentration was 10 mg ml⁻¹ unless otherwise specified. For live-dead microscopy experiments, biofilms were alkalinised at RNA transesterified conditions of 0.3 M KOH at 37 °C for 16 h and the pellet was collected after centrifugation at 10621 x *g* for 5 min and further processed according to live-dead staining procedure mentioned in the staining section below. For rheology measurements on alkalinised biofilms, the abovementioned alkalinisation condition was followed, and the suspension was directly pipetted to the HAAKE MARS rheometer (Thermofisher Scientific, catalogue number: 3790600,) for analysis. For agarose gel electrophoresis, 1% agarose gel solution was prepared from Viviantis LE grade agarose using 1x TAE buffer (40 mM Tris, 20 mM acetate and 1 mM EDTA, pH 8.6). Alkalinised nucleic acid gel isolate (5 mg ml⁻¹) was run horizontally for 45 min at 85 V. After electrophoresis, the gel was stained for 0.5 h with ethidium bromide and visualised under UV light[38].

## Enzymatic digestions

Twenty mg (wet weight) of biofilms directly harvested from the Erlenmeyer Flasks, clinical sputum sample, or 10 mg (wet weight) of nucleic acid gel isolate, were resuspended in 1 ml of either (i) RNase buffer (50 mM Tris HCl, 10 mM EDTA, pH8) with 0.3 mg RNase A from bovine pancreas (Sigma Aldrich R 6513), (ii) DNase I buffer (100 mM Tris (pH 7.5), 25 mM MgCl₂ and CaCl₂) with 0.4 mg DNase I from bovine pancreas (Sigma Aldrich 9003-98-9), or (iii) RNase H buffer (20 mM HEPES-KOH Buffer (pH 8.0), 50 mM KCl, 4 mM MgCl2, 1 mM DTT, 50 µg ml⁻¹ BSA and 1 Unit Ribonuclease H (R6501, Sigma Aldrich). All digestions were performed at 37 °C for 0.5 h[9]. For pre-treatment with DNase I, treated biofilm samples were incubated with either 0, 0.02, 0.05, 0.10 or 0.4 mg ml⁻¹ DNase I at 37 °C for 30 min. All NA gel isolate and enzymatic digestion experiments were performed in three biological replicates.

For rheological assay, digested samples were collected and deposited by pipetting onto a HAAKE MARS 60 Rheometer (Thermofisher Scientific, catalogue number: 3790600,) with parallel plates for viscoelasticity measurements. For microscopy, digested samples were washed in situ once with double-distilled water. For tape station analysis (Agilent 2200, catalogue numbers: screen tape 5067-5576 and sample buffer 5067-5577), digested samples were centrifuged in situ for 5 min at 3824 x *g*, the supernatant collected and then analysed as per standard Agilent RNA ScreenTape System Quickguide.

## Extracellular nucleic acid gel isolation

Briefly, freeze-dried biofilms were solubilised using ionic liquid: 40% (v/v) 1-ethyl-3-methylimidazolium acetate (EMIM Ac): 60% (v/v)

dimethyl acetamide (DMAc) at 55 °C for 16 h. This was followed by centrifugation at 10621 x *g* for 10 min to remove any undissolved material. The supernatant was subjected to perchloric acid precipitation (70%) on ice for 15-30 min. The sample was centrifuged at 10621 x *g* for 10 min to recover the pellet. The pellet was dialysed against distilled water for two days and purified using gel permeation chromatography column (Agilent PLgel 10 µm, 10⁵ Å, Serial number-0006167034-4) with ionic liquid as an eluent[9]. Purified eDNA formed a gel and was used for further analysis by NMR.

## Planktonic and biofilm cellular RNA extraction for sequencing

For planktonic RNA, overnight pre-cultures of *P. aeruginosa* and *lasB* mutant PW7302 and PW7303 were collected by centrifugation (10621 x *g* for 10 min at 22 °C). For biofilm cell RNA, five-day *P. aeruginosa* PAO1 wildtype biofilms were lyophilised. Cellular pellets and freeze-dried biofilm were washed with ice-cold 1x PBS and resuspended in 1x TE buffer followed by probe sonication of four cycles, 25 s pulse on per cycle and 15 s pulse off, with an amplitude of 40%. The sample was centrifuged at 3824 x *g* for 10 min and the supernatant was collected and concentrated using 0.2 µm syringe filter (Acrodisc® Syringe Filters with Supor® Membrane, Sterile - 0.2 µm, 25 mm) to remove any traces of cells. The RNA was then extracted using the trizol method[39]. The RNA sample was solubilised in TE buffer, treated with turbo DNase kit (Invitrogen, catalogue number: AM2238) and further purified using a Zymo RNA Clean & Concentrator-5 kit (Zymo Research, catalogue number: R1016). After clean-up, RNA concentrations and purity were measured using the Qubit high sensitivity RNA and DNA fluorometry kit (Thermofisher Scientific, catalogue numbers: Q32852, Q33230, respectively) and NanoDrop 2000 Spectrophotometer respectively.

## Biofilm matrix RNA extraction for sequencing

Biofilm matrix extraction was performed on five-day *P. aeruginosa* PAO1 wildtype and *lasB* mutant (PW7302 and PW7303) biofilms. The perchloric acid precipitate intermediate from the nucleic acid gel isolation was resolubilised in ionic liquid 40% (v/v) 1-ethyl-3-methylimidazolium acetate (EMIM Ac): 60% (v/v) dimethyl acetamide (DMAc) at 55 °C for 1 h. The sample was centrifuged at 3824 x *g* for 10 min at 4 °C. This supernatant was collected, and pure isopropanol was added (3:4 volume ratio of perchloric acid extract) and kept on ice for 2 h. After 2 h, the sample was centrifuged at 3824 x *g* for 10 min at 4 °C and supernatant removed. The resulting pellet was dialysed against double distilled water (SnakeSkin™ Dialysis Tubing, 3.5 K MWCO, 22 mm) and the retentate was lyophilised (FreeZone Plus 4.5 Liter Cascade Benchtop Freeze Dry System). The lyophilised sample was solubilised in TE buffer pH 8. Solubilised sample was treated with turbo DNase kit (Invitrogen, catalogue number: AM2238). This was repeated for another cycle until the concentration of DNA was less than 5 % of the RNA concentration in the sample. RNA was further purified using Zymo RNA Clean & Concentrator-5 kit (Zymo Research, catalogue number: R1016). After cleaning up, the RNA concentration and purity were measured using Qubit high sensitivity RNA and DNA fluorometry kit (Thermofisher Scientific, Catalog no: Q32852, Q33230 respectively) and NanoDrop 2000 Spectrophotometer respectively. Biofilms were grown under model static conditions in biological triplicate, and sampled on days 1, 2, 3 and 5 for extraction of iRNA and eRNA. Hence, six samples (three for iRNA and three for eRNA extraction) were collected each day.

## Nuclear magnetic resonance (NMR)

Solution state NMR experiments were performed on non-alkalinised and alkalinised nucleic acid gel isolated and *P. aeruginosa* biofilm using an 800 MHz Bruker Avance III spectrometer at 25ºC. NMR buffer contained 10 mM KH₂PO₄ and K₂HPO₄ (pH 7) and 10% (v/v) D₂O or 100% (v/v) D₂O. 1D NMR experiments included ¹H and ³¹P direct detection, while 2D NMR analysis included ¹³C-HSQC, ¹³C-HSQC-

TOCSY, and $^1$H-$^{31}$P HETCOR, COSY. All spectral analyses were performed using Bruker Topspin and SPARKY software. Qualitative and quantitative analyses of the purified nucleic acid isolate from biofilms were undertaken from one-dimensional NMR analyses across three biological replicates, while the detailed structural analysis was achieved using two-dimensional NMR on a single replicate.

## Library preparation (RiboZero _modified method)

The RNA quality and RIN value of biofilm matrix, planktonic and biofilm cellular RNA extracted from *P. aeruginosa* were confirmed using by tape station. Library preparation was performed using Illumina's TruSeq® Stranded Total RNA Library Prep, following the manufacturer's protocol with 50 ng of the total RNA as input. During the library construction, RNA enrichment was not conducted. The RNA underwent fragmentation (5 min at 94 °C) and reverse transcription to obtain double-stranded DNA inserts with fragment size of 120–250 bp (a median size of 180 bp). The fragmented dsDNA then underwent end repair, adapter ligation, and PCR amplification to generate the final sequencing-ready library. The quality of the library was checked using Agilent D1000ScreenTape. A single peak in the expected region of ~300 bp indicated that the library was suitable for sequencing. The different libraries were pooled, and quality control performed using Agilent high sensitivity DNA kit and KAPA quantification.

## RNA sequencing and analyses

The sequencing was performed using the Illumina® HiSeq 4000 (Paired-End) Cluster kit on the cBot 2 System, generating paired end 150 bp reads. All raw reads were quality checked and adapter trimmed using the Trim Galore package[40] before removing any remaining ribosomal RNA reads using the SortMeRNA package[41]. The above rRNA removal was performed by comparing each read to the SILVA 16 S rRNA v138 database. Following the *in-silico* removal of rRNA, all remaining non-rRNA reads were mapped to a *P. aeruginosa* reference genome (downloaded 14 February 2022; https://www.Pseudomonas.com/strain/download) using Bowtie2[42]. For each sample, mapped reads were converted to SAM files and sorted using SAMtools[43]. The mapped reads were quantified using the HTSeq count function[44] before undergoing DESeq2[45] analyses to determine differential expression of genes between samples. The sequences have been uploaded to Sequence Read Archive (SRA) and are currently awaiting accession number credentials. RNA sequences identified from the sequencing detailed above were further used as a template for single molecule inexpensive fluorescent in situ hybridisation (smiFISH) probe design and confocal microscopy studies. RNA sequencing was performed in three biological replicates.

## Statistics

In this study, differential expression analysis was performed using the DESeq2 package in R. DESeq2 employs statistical methods tailored for RNA-Seq data analysis. The negative binomial model was fitted to estimate size factors and dispersion. Significance was determined using two tailed statistical tests: the false discovery rate correction for adjusted p-values and a threshold for log 2-fold change and unadjusted *p*-values ($P < 0.01$). The volcano plot, created with ggplot2, comprises a plot of log 2-fold change against negative log 10 p-values. This analysis identified differentially expressed genes and provided insights into gene expression changes between conditions.

The standard deviation bars indicated in bar graphs are generated based on the mean values calculated from three biological replicates. For rheogram data, the biological triplicates are averaged for each condition and plotted against frequency.

## Staining

eDNA staining was achieved following incubation of biofilm-coated glass slides with 2 μM TOTO-1 iodide (1 mM solution in DMSO;

Thermofisher Scientific, catalogue number: T3600) for 15 min in the dark. Live-dead imaging was performed according to LIVE/DEAD™ BacLight™ Bacterial Viability Kit, for microscopy (Catalog number: L7012, Thermofisher Scientific). Briefly, SYTO 9 dye, 3.34 mM (component A), 300 μl solution in DMSO and propidium iodide, 20 mM (component B), 300 μL solution in DMSO are mixed in qual volume. Three microlitres of the mix were added to every 1 ml of bacterial cell suspension and incubated in dark at room temperature for 15 min. Five microlitres of stained suspension were deposited onto the slides and imaged using LSM780 confocal microscopy (Zeiss).

## smiFISH for biofilm RNA localisation

Briefly, primary, unlabelled oligoribonucleotide probes targeting mRNA were designed using an oligostan R software with a prewritten code available[46]. These identified multiple oligoribonucleotide probes for each gene. Each primary probe contained a shared (i.e., FLAP) sequence towards 3' end that is complementary to a secondary FLAP probe that has two fluorophores attached at both ends of the sequence. The combination of multiple probes, and two fluorophores per probes, enabled visualisation of even single RNA molecules. RNA seq data of *P. aeruginosa* cells provided the list of mRNA transcripts that were either highly up-regulated or down-regulated in both biofilm matrix and planktonic cells. The probes against the mRNA target were provided by Integrated DNA Technologies (IDT).

In total, 27 primary probes specific for *lpdV*, 13 primary probes specific for *antB* and eight probes specific for *bkDA-2* were used. The scrambled primary probes of different mRNA as a negative control were generated using online software tool Genscript by pasting the primary probe sequences onto the toolbox (https://www.genscript.com/tools/create-scrambled-sequence). All primary and secondary FLAP sequences used in the study are available within Supplementary Data 1 file of this manuscript.

## Confocal microscopy imaging

Sample slides were inverted and viewed using Carl Zeiss LSM 780 – laser scanning confocal microscope with fast spectral detection (32-GaAsP array; 63x objective lens). ImageJ-Fiji software was used for image analysis. For the time series experiment, *P. aeruginosa* biofilms were collected every day for five consecutive days, and smiFISH staining using the *lasB* probe was performed as above.

Image colocalisation analyses for all the images were performed using Imaris x64 software. Mander's colocalisation coefficient was calculated by setting threshold for both mRNA and eDNA channels. The colocalised region from both red mRNA and green eDNA channels was isolated and represented in each image. All microscopy imaging and quantification were performed across three biological replicates, except for the non-infected sputum sample where three technical replicates were used. For all images assessing eRNA colocalisation with eDNA fibres, contrast adjustment was performed relative to fibre signal rather than the signal from dead cells.

## Image quantification

For quantification of eDNA and extracellular mRNA quantification in biofilms across different days and different DNase I concentrations, total fluorescence intensity was obtained from images by calculating the raw integrated intensity using ImageJ. The Mander's coefficient, which measured the fraction of colocalisation between two fluorescence channels, was calculated by building a colocalisation channel in Imaris x64 software. This indicated the percentage of the mRNA signal (red channel) that was colocalised with the eDNA or cell signals (green channel). The percentage of live-to-dead population was calculated by measuring total fluorescent intensity using ImageJ. *ssrA* and *crcZ* mRNA positive cells in the biofilms were calculated based on the "Find Maxima" function in ImageJ. The eDNA fibre reduction was quantified using the threshold option to eliminate the cell signal based on the

fluorescence intensity difference between cells and eDNA fibres, obtained using ImageJ Fiji software.

## Sample and probe preparation

Primary probes (number varied depending on the size of mRNA) were produced in 96-well plates and delivered wet and frozen. For use, the probes were suspended in TE buffer pH 8 (final concentration 100 mM). The final concentration of individual probes was 0.833 mM after five-times dilution with double-distilled water.

The fluorescent FLAP was delivered lyophilised. The FLAP was resuspended in TE buffer to a final concentration of 100 mM. Five-day *P. aeruginosa* biofilms and clinical sputum samples were collected by gently aspirating with a shortened 1 ml pipette tip. These were subsequently transferred to 2 ml Eppendorf tubes and rinsed with double-distilled water. The biofilms were neither fixed nor permeabilised to avoid intracellular contamination. The primary probe and secondary FLAP were prehybridised using conditions described in Supplementary Tables 6 and 7. The washed biofilms were incubated in 40% formamide freshly prepared in 1X SSC buffer for 15 min at 22 °C. During incubation, Mix 1 and Mix 2 were prepared on ice (Supplementary Table 8). Mix 2 was vortexed for 30 s. Mix 1 and Mix 2 were combined and vortexed for 30s. One hundred microlitres of the hybridisation mix were added to slides with deposited biofilms and covered with a Petri dish, wrapped in parafilm and incubated at 37 °C overnight[46]. After hybridisation, biofilms were gently pipetted onto a poly-L-lysine-coated microscopic slide and covered with a coverslip.

## Clinical sample collection

Three clinical airway specimens HP0005, HP0007 and HP0017 (spontaneously expectorated sputum) were obtained from three patients with severe post infection bronchiectasis. One airway sample Nano0015 was collected from a healthy non-diseased patient. Ethics approval for sample collection was obtained from Singapore General Hospital (CIRB 2017/2109) and Nanyang Technological University, Singapore (NTU; IRB-2017-05-035). Written informed consent was obtained from the individual prior to sputum collection. The specimens were transported on ice to the laboratory at NTU within 4 h of collection.

## Microbiological detection and identification of *Pseudomonas aeruginosa* in clinical sputum

*P. aeruginosa* was detected in three of the four sputum samples by inoculation on blood agar and MacConkey agar (BD-BBL) with a loopful of sputum incubated overnight at 37 °C. Colourless colonies isolated on MacConkey agar were presumptively identified as *P. aeruginosa* by Gram staining, oxidase test and BD BBL™ Crystal™ microbial ID strips (N/H), and subsequently confirmed by MALDI TOF (Bruker MALDI Biotyper Identification system). Further confirmation of the presence of *P. aeruginosa* was performed by metagenomic sequencing of the sample. To achieve this, sputum DNA was extracted using the Roche High-pure PCR Template Preparation Kit (Roche) followed by quantification using the Qubit dsDNA High Sensitivity (HS) Assay Kit (Invitrogen, USA)[47–49]. Briefly, 1 ml of sputum DNA was added to 199 μl of qubit dsDNA HS reagent, vortexed for 10 seconds and measured for DNA concentration. Sequencing was performed on a HiSeq 2500 platform (Illumina, USA) at the Singapore Centre for Environmental Life Sciences Engineering (SCELSE)-NTU core sequencing facility according to library preparation and DNA sequencing methods as described[50,51]. Briefly, prior to library preparation, DNA quantification was repeated with Invitrogen's Picogreen assay. Sequencing libraries were prepared using the Swift Biosciences Accel-NGS 2 S Plus DNA Kit. DNA was sheared using E220 ultrasonicator to ~450 bp.

Bioanalyzer DNA 7500 chip (Agilent) was used to determine library size. Library was validated using qPCR on a ViiA-7 real-time

thermocycler (Applied Biosystems) using Kapa Biosystem's Library Quantification Kit for Illumina Platforms. Paired end sequencing with Illumina HiSeq 2500 rapid run machine was performed.

Raw sequencing reads with a minimum Phred score 20 and >30 bp in length were selected and adapter-trimmed using Cutadapt (version 1.14)[52]. Trimmed reads were mapped against the GRCh38 human reference genome with Bowtie2 (version 2.3.2)[42]. Unmapped non-host reads were separated for further analysis and aligned against the NCBI non-redundant protein database (7 August 2017) with Diamond (version 0.9.9)[53]. Based on these alignments, microbial taxonomical classification was generated using the Lowest Common Ancestor (LCA) algorithm implemented in MetaGenome Analyzer (MEGAN, v6.8.18) with a minimum score of 100 and a support of ≥25[54]. Determination of *P. aeruginosa* relative abundance was made using total microbial (non-human) reads and reported as a percentage.

## In silico binding assay

The designed *lasB* primary probe for targeting *P. aeruginosa lasB* mRNA was checked individually against a model *P. aeruginosa* culture. Each probe was tested for its hybridisation efficiency against *P. aeruginosa* A and human elastase expressing mRNA sequence using the link: http://mathfish.cee.wisc.edu/mismatch.html. The 13 probes were designed using mathFISH webtool to measure the hybridisation efficiency and Gibbs free energy of binding.

## Reporting summary

Further information on research design is available in the Nature Portfolio Reporting Summary linked to this article.

## Data availability

The source data file supporting the findings of this study are available within the paper and its Supplementary Information. The RNA sequencing data analysis from PAO1 and metagenomic data of clinical sputum sample has been deposited to sequence read archive (SRA) with a BioProject accession number PRJNA890467 and PRJNA595703 respectively. The data can be found from https://www.ncbi.nlm.nih.gov/sra. Source data are provided with this paper.

## Code availability

Pre-written source code for designing primary and secondary FLAP probes was used, which is available at https://bitbucket.org/muellerflorian/fish_quant/src/master/Oligostan/Oligostan.r.

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

## Acknowledgements

We thank Professor Christian Damsgaard and their group at Aarhus University for suggesting and advising on the inclusion of smiFISH methodology. We acknowledge Dr Poh Wee Han for providing *lasB* transposon mutants, Dr Sujatha Subramoni for protocols and assistance with elastase activity assays, Dr Sakcham Bairoliya for discussions regarding eDNA and eRNA extraction, Dr Shi Ming Tan for helping with probe hybridisation efficiency studies, Dr NG Poh Yong for RNA ribozero sequencing services, and Dr Sharon Longford for proofreading. The Singapore Centre for Environmental Life Sciences Engineering (SCELSE) is funded by the Ministry of Education, Singapore, the National Research Foundation of Singapore, Nanyang Technological University Singapore (NTU) and National University of Singapore (NUS), and hosted by NTU in partnership with NUS (S.K.).

## Author contributions

S.M., L.L.W., F.R.W. performed experiments; S.M., T.S., S.K., S.A.R. designed experiments. F.R.W. (NMR and nucleic acids), A.T.P. (nucleic acids), S.S. (sequencing), T.K.J. (clinical sample data collection), O.W.M. (clinical sample data collection), P.Y.T. (clinical sample collection provider), S.H.C (clinical sample usage), M.H.B.I. and S.M (probe design). Y.H.F helped with image processing and quantification. S.M., T.S., S.A.R., S.K. wrote the manuscript. T.S. and S.K. (project management).

## Competing interests

The authors declare no competing interests.
