## [Peer Review File · Nature Communications]

RNA is a key component of extracellular DNA networks in Pseudomonas biofilmsReviewer #1 (Remarks to the Author):

In this paper, the authors examined the role of extracellular RNA in construction of viscoelastic extracellular DNA networks in a *Pseudomonas aeruginosa* biofilm. NMR analysis demonstrated that alkalization of the eDNA gel recovered from the biofilm degraded extracellular RNA into mono-ribonucleotides. CLSM analysis confirmed that alkalization of the *P. aeruginosa* biofilm resulted in complete loss of TOTO-1-stained extracellular fibers. smFISH revealed that some specific mRNA species including *lasB* mRNA colocalized with the fibers, but the most abundant RNA in the biofilm matrix did not. Furthermore, *lasB* mRNA also colocalized with eDNA fibers in airway specimens colonized with *P. aeruginosa*.

The reviewer agrees with the authors that the clarification of the formation mechanism of eDNA networks in biofilms would be of interest to microbiologists. While some potentially interesting results have been observed, there are several concerns regarding this study that the authors need to clarify as described below.

Major comments:

1. The data presented are only partially convincing and too superficial to provide an appropriate understanding of the link between extracellular RNA and eDNA-containing extracellular fibers in the *P. aeruginosa* biofilm. It has been already reported, as mentioned by the authors, that the positive charged Pel exopolysaccharide crosslinks with eDNA through ionic interactions in the *P. aeruginosa* biofilm (Jennings et al., PNAS 2015). Interactions between exopolysaccharides and eDNA have also been reported in other bacterial biofilms (Mlynek et al. J. Bacteriol. 2020). In addition, eDNA is known to interact with fibrous proteins like extracellular amyloids (Gallo et al. Immunity. 2015), and *P. aeruginosa* also produces extracellular amyloids (Dueholm et al. Mol Microbiol. 2010). The authors should examine roles of exopolysaccharides and proteins in the formation of eDNA-containing extracellular fibers in the *P. aeruginosa* biofilm using the respective mutant strains. Effects of alkalization on stability of these extracellular polymeric substances remain unclear. While the authors mentioned that neither proteins nor polysaccharides were present in the isolated eDNA gel (page 6, line 2), such extracellular components may have been washed out during preparation of the eDNA gel sample. Could the authors observe TOTO-1-stained fibers in the eDNA gel by CLSM?

2. Although the authors analyzed the effect of alkalization on the extracellular fibers in the biofilm (Fig. 1), its effects on quantity and structure of the biofilm are not shown. If the biofilm is dispersed via dissociations of key biofilm matrix components including eDNA, Pel, and/or proteins from bacterial surface by the alkali treatment, the authors can not eliminate the possibility that alkalization not only degraded extracellular RNA into small ribonucleotides but also peeled off the biofilm matrix components, leading to disappearance of the extracellular fibers. The former possibility can be analyzed by a conventional crystal violet-staining method, while the latter one can be addressed by CLSM. The authors showed only selected x-y images of the biofilm but should show 3D images of entire biofilms. Many NMR data represented profiles of nucleotides in the isolated eDNA gel and the biofilm following alkalization (Fig. 1A, B, Supplementary Fig. 2, 3, 4), but the authors had better to compare these profiles before and after alkalization in order to examine the effect of alkalization on nucleotides. In addition, NMR analysis indicated that eDNA was not degraded into mono-deoxyribonucleotides by treatment with NaOD, while it was still unclear whether eDNA was processed into short fragments following harsh alkalization (pH 12, 37°C for 16 h), which can be confirmed by agarose gel electrophoresis.

3. Agarose gel electrophoresis revealed that the isolated eDNA gel of *P. aeruginosa* PAO1 consisted of nucleotides in size from 1,500 to 7,000 nts (Supplementary Fig. 7). These sizes are much smaller than those of genomic DNA and eDNA which was previously reported (Jennings et al., PNAS 2015). Was eDNA partially degraded during extraction from the biofilm? The authors mentioned in the text that RNase A treatment of the eDNA gel isolate did not disrupt the network structure and there was no reduction in the MW of constituent nucleic acid (Supplementary Fig. 7A, lane 4). However, there is no data to show the effect of RNase A on the network structure, and Supplementary Fig. 7A clearly shows that RNase A treatment reduced the sizes of nucleic acids to the same extent with the heated sample, indicating that RNA in the sample was actually degraded after the RNase A treatment. Nevertheless, if there was no effect of RNase A on the network

structure in the biofilm as mentioned above, roles of extracellular RNA in the network formation are puzzling. The effect of RNase A on formation of extracellular fibers can be examined by adding RNase A to the culture from the onset of biofilm formation. This experiment will strengthen the conclusion by the authors that extracellular RNA plays a key role in viscoelastic eDNA network formation.

4. It is not convincing that extracellular RNA in the eDNA gel isolated from the biofilm was compared with intracellular RNA isolated from planktonic cells by RNA-seq. Based on this comparison, the authors selected *lasB* mRNA as an abundant mRNA in the biofilm matrix for further smFISH analyses. However, extracellular RNA in the eDNA gel should be compared with intracellular RNA in the biofilm cells to reach the conclusion that some specific mRNA species were enriched in the biofilm matrix or that expression levels of mRNA in cells reflect the contents of mRNA in the biofilm matrix. The most abundant RNA in the biofilm matrix was 23S rRNA, but not *lasB* mRNA. Given that the abundance of *lasB* mRNA was much smaller than those of the others in the biofilm matrix, it is expected that the *lasB* knockout mutants still produce a biofilm and that *lasB* mRNA is not essential for biofilm formation. It is confusing that the most abundant 23S rRNA (and 16S rRNA) in the biofilm matrix was detected only inside the cells but not on the extracellular fibers (Supplementary Fig. 11 and 12). This was probably due to use of single probe for 23S rRNA and 16S rRNA (Supplementary Table 4). In contrast, multiple probes were used to detect the *lasB* mRNA (Supplementary Table 3), which could enhance the sensitivity of the detection. If multiple probes are used for 23S rRNA and 16S rRNA, colocalization of these RNA with extracellular fibers could be detected.

5. Based on the *in silico* binding affinity study, the authors concluded that a far weaker specificity and lower binding affinity of the designed *lasB* probes to the human elastase gene. However, the authors could not eliminate the possibility of cross-reaction between the *lasB* probes and non-specific RNA species or other components in the human samples, which should be confirmed experimentally. For instance, the human clinical samples without the *P. aeruginosa* infection can be used as a negative control. In addition, how many clinical samples were used and how many samples were positive for *lasB* mRNA on fibers?

6. The major weakness of the manuscript is that it completely ignores the existing literatures on extracellular RNA in biofilms (Domenech et al. *Sci. Rep.* 2016; Scherbakova et al. *bio-protocols* 2020; Chiba et al. *npj Biofilms and Microbiomes* 2022).

Minor comments:

1. The authors used DNase I purchased from Sigma Aldrich (Fig. 3-7, Supplementary Fig. 7-14), but this reviewer experienced that DNase I from Sigma Aldrich was contaminated with some protease(s). If this is also the case with DNase I used in this study, the interpretations and conclusions of various data are severely influenced. So, the authors should confirm that no detectable proteolytic activity in DNase I used in this study. In addition, treatment with 0.1 mg/ml DNase I did not degrade extracellular fibers in the biofilms completely. Does longer time incubation with this enzyme or use of the higher concentrations abolish the fibers?

2. There is no or less evidence of direct interactions between extracellular RNA and eDNA. This should be confirmed by ITC, SPR, pull-down, or other biochemical methods.

3. This study focuses on *P. aeruginosa*, and there is no data about other *Pseudomonas* species. So, the title is too broad.

4. The last paragraph in Introduction. It is unclear why understanding of the nucleic acid composition can resolve the mechanisms of eDNA release from bacteria.

5. Details of smFISH are missing in Methods. More information should be provided to reproduce the experiments. For example, conditions of fixation, permeabilization, staining, buffer compositions, etc. should be indicated.

6. Letters of scales in microscopic images should be deleted.

7. Supplementary Fig. 7A. The authors should use another DNA ladder maker covering sizes more than 7,000 nts. It is unclear whether reaction time was enough or not, as complete digestion of DNA and RNA had not been achieved by DNase/RNase treatment.
8. Supplementary Fig. 7B. PAO668.4 was incorrectly assigned to the right higher peak in the data of intracellular RNA.
9. This reviewer feels that contamination of DNA less than 5% of RNA was too high. The authors should reduce DNA contamination as much as possible.
10. Supplementary Table 2B. Definition of Log2 fold change is unclear.
11. Supplementary Fig. 9. More detailed explanation of the mutant strains PW7302 and PW7303 should be added.
12. Supplementary Fig. 11 and 12. The lasB mRNA is 16S rRNA or 23S rRNA within the figure.
13. Supplementary Fig. 13. How about clinical sputum left untreated with DNase I?

Reviewer #2 (Remarks to the Author):

To authors,

Your paper is very interesting and well written.

Please find below some comments:

- in the whole text: replace "nt" with "nucleotides".
- line 185 and 189: remove the point after the word "table".
- line 223: the name of the technique is "single molecule. inexpensive fluorescent in situ hybridisation (smiFISH)", not smFISH. Please modify it.
- line 224: please indicate the number of primary probes used for each RNA.
- line 225: the sentence is not clear. Please modify it with "probes pre-hybridized to a FLAP sequence labelled with two fluorophores".
- Line 227: remove the second parenthesis after "2A".
- in the whole text: please replace "smFISH" with "smiFISH"

Please find below some questions:

- The eDNA is not totally colocalized with mRNAs. How you explain the partial co-localization shown in figures 3 and 4?

Thank you!

Reviewer #3 (Remarks to the Author):

Here authors identified a role of eRNA in *P. aeruginosa* biofilm matrix. eRNA was found to co-localize with eDNA fibers in the matrix and stabilize this structure. This is an interesting and well-written manuscript. My main comments are to clarify points within the main text, and perhaps perform additional controls to help support the authors conclusions. My specific comments are below.

Major comments

1. L139 - 140. Can authors quantify bacterial viability after alkaline transesterification of the biofilm, to confirm that the loss of eDNA fibers and viscoelasticity is due to disruption of eRNA and eDNA interactions and not due to cell death.
2. Figure 1D. It looks like cells are mostly stained in this image. Do authors predict that the alkaline transesterification has killed most of the cells, allowing them to take up the TOTO-1? Can authors comment on this?
3. L169 - 182. How old were the biofilms when eDNA gel was isolated? Are these results consistent across eDNA isolated at different stages of biofilm develop? Do these results correlate with findings that eDNA at later stages of biofilm development is resistant to DNase?
4. L190 - 191. How do the RNA transcripts compare between the matrix and biofilm cells? Do authors predict to see the same enrichment of transcripts in the matrix? Or is the transcription of these genes elevated in the biofilm and will therefore form a high component of the eRNA?
5. L271 - 277. Was there a difference in the eDNA fiber structure of lasB mutant biofilms? Was there a difference in total eRNA concentration in lasB mutant biofilms?
6. Figure S11 and S12. Perform colocalization analysis on these images to demonstrate that the labelling localizes to the cell and not eDNA.
7. L296 - 301. Can authors clarify these results/ conclusions given that 23S RNA was identified in Figure S7B?
8. L304 - 313. I find this section confusing. RNA-Seq identified *ssrA* and *crcZ* transcripts in the eRNA of the biofilm matrix. However, smRNA FISH identified a low abundance of these transcripts. Authors conclude that this is due to specific transport of RNA sequences into the matrix. Was the RNA-Seq performed on the biofilm cells and not the matrix eRNA? Can authors clarify these points in the text?
9. L340 - 346. Was there an increase in lasB mRNA in the eRNA between days 3 - 5? Was there an increase in eDNA content overtime? Can authors quantify this by fluorescence intensity?
10. L354 - 357. To clarify, lasB mRNA was detected in sputum without DNase treatment? Can authors comment on this difference between in vivo and in vitro. Did authors see an increase in mRNA labelling with DNase treatment? Quantify fluorescence intensity.
11. L368. How did authors confirm that eDNA in the sputum was bacterial and not host derived?
12. L442 - 443. Can authors confirm if the centrifugation and lyophilization created artifacts in the analysis? It could this process affect viability of the cells, releasing RNA? Could this process influence the colocalization of eDNA and eRNA that was detected by microscopy?
13. L546 - 562. In line with the above comment, did authors perform microscopy on native biofilms? Or only on the processed biofilms? If so, how did authors stain the lyophilized samples?
14. Methods section. Include number of biological and technical replicates for all assays.

Minor comments

1. L61. Clarify the statement 'eDNA from the cytosol into membrane vesicles'. This sounds that eDNA is only released into membrane vesicles by ECL. eDNA is also released into the extracellular environment and incorporated into the biofilm matrix through this mechanism.
2. L67. Clarify that later biofilms are resistant to DNase treatment
3. Figure 2. Suggest rephrasing the title and legend of the figure as mRNA is not located on the chromosome.
4. L215. What are the gene names of these PA numbers?
5. Figure S10 is mentioned after S11 and S12.
6. Figures S11 and S12. Images are labelled with LasB mRNA rather than 16S and 23S.
7. Figure 5. Is the 10um scale bar also consistent for the zoomed insets?
8. L519. Is this subheading correct?

Reviewer #4 (Remarks to the Author):

The manuscript by Mugunthan et al. entitled "Extracellular RNA is a key component of viscoelastic eDNA networks in *Pseudomonas* biofilm" is based on the discovery of extracellular RNA (eRNA) being present in the biofilm matrix, with the eRNA being complexed with eDNA providing a viscoelastic structure. The authors also emphasize the presence of specific mRNA species with the

biofilm matrix. While the notion of extracellular RNA being a component of the biofilm matrix is exciting, the data are not that convincing. For example, the authors use confocal microscopy/smFISH to visualize eRNA but the visualization of mRNA species is only qualitative and appears to be inconsistent (spotty). Moreover, proper smFISH controls are missing (e.g. using mutant strains and scrambled probes, etc). It also appears that the conditions used by the authors result not just in eDNA, but also bacterial cells, to be stained by cell impermeable stain TOTO-1. Additional concerns are raised by the data presentation, as Figures are incorrectly referenced throughout the manuscript, and the terms "biofilm" and "biofilm matrix" are frequently used interchangeably. Additional comments and concerns are given below.

Other comments

1. In support of RNA being a component of the biofilm matrix, the authors provide 5 figures at the beginning of the manuscript, however, they are all supplementary figures. It would be nice if the first evidence for eRNA is Figure 1 instead of Supplementary Figures 1-5.
2. smFISH data, Please provide quantitative data as well. Additionally, include controls (e.g. using mutant strains, scrambled probes)
3. l. 354-357 The authors state that "... lasB mRNA also appeared in the extracellular matrix of the clinical sample (Fig. 7, red channel). The sputum sample had fewer cells staining positive for lasB mRNA ...". However, I am having a hard time detecting a difference. I highly recommend presenting the data in a quantitative manner. The qualitative approach chosen here is not convincing.
4. What is a "extracellular nucleic acid gel". Please define
5. l. 105. 1H-13C not shown in Supplementary Figure 1.
6. l. 141 The authors state that "there was also a loss in biofilm viscoelasticity (Supplementary Fig. 6). " However, the data shown in Supplementary Fig. 6 do not support loss of viscoelasticity upon transesterification of RNA/loss of DNA. The authors should consider revising the figure legend and/or data labels
7. l. 144. The authors state that "RNA transesterification upon alkalinisation was associated with biofilm and matrix dissolution". However, no evidence has been provided for biofilm dissolution or matrix dissolution.
8. l. 186-189 Some of the mRNA transcripts listed here appear to be decreased rather than enriched in the matrix
9. l. 197, enriched in biofilm or biofilm matrix, please clarify. Please also make sure that the correct figures and tables are referred to in this section
10. l. 211 what is the meaning of "RNA sequenced from the extracellular matrix of five-day-old *P. aeruginosa* biofilms and the chromosomes of planktonic *P. aeruginosa* cells".
11. l. 242-244 recommend showing data as graph
12. l. 298 text does not match content shown in Suppl figure 12
13. l. 335-336, values don't match data; authors provide 6 values for 5 time points
14. Figure 3 please provide means of presenting data in a quantitative manner
15. Figure 5 no eDNA fibers are visible, Instead, only bacterial cells are shown despite DNase treatment. Please clarify why the appearance of the biofilm (matrix) is so different from those shown in previous figures. Also, please clarify if the samples are from planktonic or biofilm cells. Also, please provide means of presenting data in a quantitative manner
16. Figure 1A-B, not convincing evidence of RNA being present in matrix
17. Figure 1C-D, Figure D does not appear to show eDNA fibers. Unclear why in Figure 1C-D bacterial cells also appear to be stained with the cell impermeable dye TOTO-1
18. Figure 7 control without DNase treatment is missing
19. Supplementary Table 1. Transcript abundance relative to what?. Same applies to data shown in Supplementary Tables 2A and 2B
20. Material and methods - The authors indicate that biofilms were grown in 2L flasks under static conditions. I assume this means that the biofilms formed as pellicles at the air-liquid interface. If this is the case, it is unclear why "...Biofilms were concentrated by centrifugation at 10,000 x g for 15 min" Equally puzzling is that the entire content of the flask was centrifuged, with "...Centrifugation stratified the biofilm mixture into three layers. The bottom two layers were collected and lyophilized"
21. Please clarify what kind of biofilms were used in this study, why biofilms were harvested in such a manner, and most importantly, what were the bottom layers used for?
22. Material and methods - One would assume when analyzing the biofilm matrix by microscopy, that the goal is to have the matrix in an intact state as possible rather than all compressed and out

of context, as in the case of centrifugation. Please clarify

23. Material and methods - RNA extraction. How were biofilms grown for subsequent RNA extraction?

Authors' response

In our response to reviewers, our text is in blue and excerpts of text from the manuscript are italicised in bold

We thank all the reviewers for their constructive comments.

Reviewer #1 (Remarks to the Author):

In this paper, the authors examined the role of extracellular RNA in construction of viscoelastic extracellular DNA networks in a *Pseudomonas aeruginosa* biofilm. NMR analysis demonstrated that alkalization of the eDNA gel recovered from the biofilm degraded extracellular RNA into mono-ribonucleotides. CLSM analysis confirmed that alkalization of the *P. aeruginosa* biofilm resulted in complete loss of TOTO-1-stained extracellular fibers. smFISH revealed that some specific mRNA species including *lasB* mRNA colocalized with the fibers, but the most abundant RNA in the biofilm matrix did not. Furthermore, *lasB* mRNA also colocalized with eDNA fibers in airway specimens colonized with *P. aeruginosa*.

The reviewer agrees with the authors that the clarification of the formation mechanism of eDNA networks in biofilms would be of interest to microbiologists. While some potentially interesting results have been observed, there are several concerns regarding this study that the authors need to clarify as described below.

Major comments:

1. The data presented are only partially convincing and too superficial to provide an appropriate understanding of the link between extracellular RNA and eDNA-containing extracellular fibers in the *P. aeruginosa* biofilm. It has been already reported, as mentioned by the authors, that the positive charged Pel exopolysaccharide crosslinks with eDNA through ionic interactions in the *P. aeruginosa* biofilm (Jennings et al., PNAS 2015). Interactions between exopolysaccharides and eDNA have also been reported in other bacterial biofilms (Mlynek et al. J. Bacteriol. 2020). In addition, eDNA is known to interact with fibrous proteins like extracellular amyloids (Gallo et al. Immunity. 2015), and *P. aeruginosa* also produces extracellular amyloids (Dueholm et al. Mol Microbiol. 2010). The authors should examine roles of exopolysaccharides and proteins in the formation of eDNA-containing extracellular fibers in the *P. aeruginosa* biofilm using the respective mutant strains.

The absence of polysaccharides and proteins in the eDNA gel isolate was confirmed by solid state NMR in our previous publication (Seviour et al 2021). In this earlier study we also demonstrated that the polysaccharides were not essential for *P. aeruginosa* biofilm elasticity by performing rheological assays on Pel and Psl knockouts. The knockout mutants displayed a modified viscoelastic response relative to the wildtype, which supported the observations that polysaccharides interact with the DNA to modify the bulk viscoelastic response, but also indicated that the polysaccharides and proteins were not essential for the eDNA to form viscoelastic networks.

Here, we sought to explain the fundamental viscoelastic network structure of the *P. aeruginosa* biofilm. Importantly, the viscoelastic eDNA gel responded the same way to pH, temperature, and DNase treatment as the biofilm from which it was extracted, suggesting similar network assembly. Polysaccharides were

therefore determined to be non-essential for the fundamental biopolymer network and biofilm viscoelasticity.

Nonetheless, we performed additional experiments that further demonstrated an eDNA gel could be isolated from biofilms of Psl and Pel polysaccharide knockout strains as well as after proteinase K digestion (Supplementary Figure 2E). eDNA fibres of Psl and Pel knockout strains and proteinase k treated biofilm were observed.

We have clarified this with the following text:

Commencing line 119:

*“The full ^1H - ^{13}C HSQC spectrum indicated that neither proteins nor polysaccharides were present in the NA gel isolate (Supplementary Figure 1)¹. eDNA fibre networked structures were visible in the polysaccharide mutants and the proteinase K digested wildtype PAO1 biofilms (Supplementary Figure 2A-C). The viscoelastic NA gel isolate from *P. aeruginosa* contained eDNA fibres (Supplementary Figure 2D), and viscoelastic NA gels were isolated from the ΔpelA and ΔpsIF polysaccharide mutants as well as the proteinase K treated PAO1 biofilms (Supplementary Figure 2E). These observations further suggest that polysaccharides and proteins are not essential for the fundamental eDNA network structure in *P. aeruginosa* biofilms.”*

Effects of alkalization on stability of these extracellular polymeric substances remain unclear.

We demonstrated that alkalinisation transesterifies eRNA, while the eDNA chain structure is preserved. It is possible that alkalinisation also destabilises extracellular proteins or polysaccharides, but they were not present in the viscoelastic isolate, so it was not possible to assess their stability upon alkalinisation. Based on the observations that dissolution of eDNA gel and biofilm occurred at the same pH (and temperature), we posit that the mechanism for network disruption is the same in both, and we now provide additional evidence that RNA contributes to the viscoelasticity. RNaseH treatment degraded *P. aeruginosa* biofilms, and RNaseA reduced their viscosity, but only following partial DNaseI treatment (Figure 5A-C). This is in agreement with our microscopy results showing the binding of the oligoribonucleotide probes only after mild DNaseI treatment (Figure 3). Partial DNaseI treatment alone does not modify the rheology of the biofilm (Figure 5C).

We have described this with the following text excerpts:

Commencing line 309:

*“However, despite the observation that the eRNA co-localizes with eDNA fibres and that eRNA transesterification upon alkalinisation occurs coincident with biofilm and matrix dissolution, eDNA fibres in 5 d *P. aeruginosa* biofilms were not affected by RNaseA treatment (Supplementary Figure 10B).”*

Commencing line 268:

“Nonetheless, the eDNA fibre structures were preserved even after prolonged incubation with DNaseI at 0.1 mg ml^{-1} (16 h) (Supplementary Figure 10A).”

Commencing line 387:

“Extracellular RNA is key to viscoelasticity of *Pseudomonas* biofilms

“The agarose gel electrophoresis image of alkalinised nucleic acid gel isolate shows that the eDNA was preserved and there was a slight reduction in DNA length to 8000 bp (Supplementary Figure 14B), indicating some loss of mass or secondary structure. To assess directly whether eRNA contributes to P. aeruginosa biofilm viscoelasticity, RNaseA digestion of the P. aeruginosa biofilm was coupled with mild pre-DNaseI treatment (0.1 mg ml⁻¹), as per the smiFISH microscopy. This increased tan δ for the biofilm relative to the untreated biofilm in the rheology frequency sweep (Figure 5A-C), although tan δ remained below 1, indicating that viscoelasticity was preserved. However, when RNaseA was replaced with RNaseH, targeting the RNA strand of RNA:DNA hybrids, the model P. aeruginosa biofilm dissolved and lost its viscoelasticity, as indicated by the increase in tan δ to > 1 (Figure 5C). The same effect was not observed without mild pre-DNaseI treatment for RNaseA or RNaseH and mild DNaseI treatment (0.1 mg ml⁻¹) with RNaseH buffer alone did not dissolve the biofilm (Figure 5C) or result in the loss of eDNA fibres.”

Commencing line 400:

“Additionally, eDNA fibres disappeared following RNaseH digestion only in conjunction with mild DNaseI pre-treatment (Figure 5A-B).”

Commencing Line 412

“The same effect of RNaseH digestion with mild DNaseI pre-treatment was observed for the nucleic acid gel isolate from P. aeruginosa, as well as P. putida and P. protegens biofilms (Supplementary Figures 8A and 14C respectively). These observations further indicate that eRNA is an integral part of eDNA fibre formation in the biofilm matrix by complexing to eDNA as extracellular RNA:DNA hybrids, and hence an important contributor to Pseudomonas biofilm viscoelastic networks. This likely explains why the eRNA was shielded from the oligoribonucleotide smiFISH probes.”

While the authors mentioned that neither proteins nor polysaccharides were present in the isolated eDNA gel (page 6, line 2), such extracellular components may have been washed out during preparation of the eDNA gel sample.

The objective in isolating the eDNA was to clean up the sample, to systematically remove constituents and find out which materials were essential for the extract to form a viscoelastic network. Polysaccharides and proteins were found to be non-essential, so removing the proteins and polysaccharides was consistent with the objective.

This is clarified with the following text:

Commencing line 119:

“The full ¹H-¹³C HSQC spectrum indicated that no proteins or polysaccharides were present in the NA gel isolate (Supplementary Figure 1) ¹. eDNA fibre networked structures were also present in the polysaccharide mutants and the proteinase K digested wildtype PAO1 biofilms (Supplementary Figure 2A-C). The viscoelastic NA gel isolate from P. aeruginosa was also characterized by the presence of eDNA fibres (Supplementary Figure 2D), and similar viscoelastic NA gels could be isolated from the ΔpelA and ΔpslF polysaccharide mutants, and proteinase K treated PAO1 biofilms (Supplementary Figure 2E). These observations further suggest that polysaccharides and proteins are not essential for the fundamental eDNA network structure in P. aeruginosa biofilms.”

Could the authors observe TOTO-1-stained fibers in the eDNA gel by CLSM?

We now provide evidence for this in Supplementary Figure 2D, showing that fibres are observed by TOTO-1 staining of the eDNA gel isolated, as clarified with the following text:

Commencing line 122:

“The viscoelastic NA gel isolate from P. aeruginosa was also characterized by the presence of eDNA fibres (Supplementary Figure 2D)”

2. Although the authors analyzed the effect of alkalization on the extracellular fibers in the biofilm (Figure 1), its effects on quantity and structure of the biofilm are not shown. If the biofilm is dispersed via dissociations of key biofilm matrix components including eDNA, Pel, and/or proteins from bacterial surface by the alkali treatment, the authors cannot eliminate the possibility that alkalization not only degraded extracellular RNA into small ribonucleotides but also peeled off the biofilm matrix components, leading to disappearance of the extracellular fibers. The former possibility can be analyzed by a conventional crystal violet-staining method, while the latter one can be addressed by CLSM.

Firstly, we provide extensive evidence that Pel, Psl and proteins were not essential for the eDNA fibres, as described in our response to the reviewer’s comment. To demonstrate that extracellular RNA degradation was responsible for the disappearance of eDNA fibres, in addition to this RNA transesterification experiment, we performed confocal microscopy on *P. aeruginosa* biofilms treated following enzymatic treatments targeting RNA. RNaseH (targeting the RNA part of RNA:DNA hybrids) treatment of biofilms following mild DNaseI treatment that by itself did not degrade the biofilm, resulted in loss of viscoelasticity, as illustrated in Figure 5C.

We have described this with following text:

Commencing line 268:

“Nonetheless, the eDNA fibre structures were preserved even after prolonged incubation with DNaseI at 0.1 mg ml⁻¹ (16 h) (Supplementary Figure 10A).”

Commencing line 390:

“To assess directly whether eRNA contributes to P. aeruginosa biofilm viscoelasticity, RNaseA digestion of the P. aeruginosa biofilm was coupled with mild pre-DNaseI treatment (0.1 mg ml⁻¹), as per the smiFISH microscopy. This increased $\tan \delta$ for the biofilm relative to the untreated biofilm in the rheology frequency sweep (Figure 5A-C) although $\tan \delta$ remained below one indicating that viscoelasticity was preserved. However, when RNaseA was replaced with RNaseH, targeting the RNA strand of RNA:DNA hybrids, the model P. aeruginosa biofilm dissolved and lost its viscoelasticity, as indicated by the increase in $\tan \delta$ to > 1 (Figure 5C). The same effect was not observed without mild pre-DNaseI treatment for RNaseA or RNaseH, and mild DNaseI treatment (0.1 mg ml⁻¹) with RNaseH buffer alone did not dissolve the biofilm (Figure 5C) or result in the loss of eDNA fibres. Additionally, eDNA fibres disappeared following RNaseH digestion only in conjunction with mild DNaseI pre-treatment (Figure 5A-B).”

The authors showed only selected x-y images of the biofilm but should show 3D images of entire biofilms.

We now provide 3-D images of both untreated biofilm and RNA transesterified biofilm in Supplementary Figure 5A and B with the following text:

Commencing line 146:

“Transesterification of eRNA in P. aeruginosa biofilms additionally resulted in complete loss of eDNA fibres (Figure 1C-D), with the 3-D image of the entire biofilm (13 µm thick) (Supplementary Figure 5A-B) showing that eDNA fibres disappear throughout the biofilm only following RNA transesterification.”

Many NMR data represented profiles of nucleotides in the isolated eDNA gel and the biofilm following alkalization (Figure 1A, B, Supplementary Figure 2, 3, 4), but the authors had better to compare these profiles before and after alkalization in order to examine the effect of alkalization on nucleotides.

We now provide the ³¹P NMR spectrum (Supplementary Figure 3A) that shows pre and post alkalisation profiles of eDNA gel isolate and *P. aeruginosa* biofilm, where the monoesterified phosphate peaks only become visible and resolved in the alkalised samples.

This is clarified using text,

Commencing line 129:

“The ³¹P NMR spectrum of the biofilm and nucleic acid gel isolate displayed no monoribonucleotide phosphate peaks pre-alkalinisation (Supplementary Figure 3A). Furthermore, the monoribonucleotide ¹H peaks could only be resolved in the 1-D ¹H NMR spectrum of the P. aeruginosa biofilm after alkalisation (Supplementary Figure 3B).”

In addition, NMR analysis indicated that eDNA was not degraded into mono-deoxyribonucleotides by treatment with NaOD, while it was still unclear whether eDNA was processed into short fragments following harsh alkalization (pH 12, 37°C for 16 h), which can be confirmed by agarose gel electrophoresis.

We now provide the agarose gel image showing the MW profile of DNA in the eDNA gel isolate before and after alkalisation (Supplementary Figure 14B). This shows that the chain structure is preserved following alkalisation with a resultant DNA length of 8000 bp (i.e. down from 10,000 bp).

We have described this with following text.

Commencing line 388:

“The agarose gel electrophoresis image of alkalised nucleic acid gel isolate shows that the eDNA chain structure was preserved and there was a slight reduction in DNA length to 8000 bp (Supplementary Figure 14B), indicating some loss of mass or secondary structure.”

eDNA was degraded slightly, which could also contribute to the loss of biofilm structure and eDNA fibre network. Nonetheless, as demonstrated by the new data showing that RNaseH degrades the biofilm, whereby eRNA is key for *P. aeruginosa* biofilm viscoelasticity, alkaline RNA transesterification degrades this foundational structure with biofilm dissolution ensuing, as described in Figure 1. We have therefore maintained a focus on the role of eRNA, in light of the new data, as per the following text:

Commencing line 488:

“eRNA is a key viscoelastic component of Pseudomonas biofilms, and biofilm alkalinisation, results in eRNA transesterification and therefore degrades a foundation structural material. This accounts for Pseudomonas biofilm dissolution by alkanisation.”

3. Agarose gel electrophoresis revealed that the isolated eDNA gel of *P. aeruginosa* PAO1 consisted of nucleotides in size from 1,500 to 7,000 nts (Supplementary Figure 7). These sizes are much smaller than those of genomic DNA and eDNA which was previously reported (Jennings et al., PNAS 2015). Was eDNA partially degraded during extraction from the biofilm? The authors mentioned in the text that RNaseA treatment of the eDNA gel isolate did not disrupt the network structure and there was no reduction in the MW of constituent nucleic acid (Supplementary Figure 7A, lane 4). However, there is no data to show the effect of RNaseA on the network structure, and Supplementary Figure 7A clearly shows that RNaseA treatment reduced the sizes of nucleic acids to the same extent with the heated sample, indicating that RNA in the sample was actually degraded after the RNaseA treatment. Nevertheless, if there was no effect of RNaseA on the network structure in the biofilm as mentioned above, roles of extracellular RNA in the network formation are puzzling. The effect of RNaseA on formation of extracellular fibers can be examined by adding RNaseA to the culture from the onset of biofilm formation. This experiment will strengthen the conclusion by the authors that extracellular RNA plays a key role in viscoelastic eDNA network formation.

We apologise that there was some confusion regarding the length of the eDNA as the molecular marker ladder was presented as nucleotides instead of base pairs. This former ladder has now been replaced with one using base-pair representation, showing that the size of the eDNA is consistent with the observations of Jennings et al., PNAS 2015 in terms of molecular weight of eDNA, as shown in Supplementary Figure 8A.

Our results suggest that the inefficacy of RNaseA treatment on biofilms is due to DNA shielding RNA. Thus, mild DNaseI pre-treatment is required to expose RNA to RNaseA. This has been described and addressed in response to an earlier comment and supported with additional rheology data and confocal microscopy. These observations clearly indicate that RNaseA upon mild DNaseI pretreatment reduces viscoelasticity in *P. aeruginosa* biofilms. RNaseH treatment under the same conditions, on the other hand, completely dissolves *P. aeruginosa* biofilms.

This has been described previously in lines 412

Commencing line 387:

“Extracellular RNA is key to viscoelasticity of Pseudomonas biofilms

“The agarose gel electrophoresis image of alkalinised nucleic acid gel isolate shows that the eDNA chain structure was preserved and there was a slight reduction in DNA length to 8000 bp (Supplementary Figure 14B), indicating some loss of mass or secondary structure. To assess directly whether eRNA contributes to P. aeruginosa biofilm viscoelasticity, RNaseA digestion of the P. aeruginosa biofilm was coupled with mild pre-DNaseI treatment (0.1 mg ml⁻¹), as per the smiFISH microscopy. This increased $\tan \delta$ for the biofilm relative to the untreated biofilm in the rheology frequency sweep (Figure 5A-C) although $\tan \delta$ remained below 1 indicating that viscoelasticity was preserved. However, when RNaseA was replaced with RNaseH, targeting the RNA strand of RNA:DNA hybrids, the model P. aeruginosa biofilm dissolved and lost its

viscoelasticity, as indicated by the increase in $\tan \delta$ to > 1 (Figure 5C). The same effect was not observed without mild pre-DNaseI treatment for RNaseA or RNaseH and mild DNaseI treatment (0.1 mg ml^{-1}) with RNaseH buffer alone did not dissolve the biofilm (Figure 5C) or result in the loss of eDNA fibres.”

Commencing line 400:

*“Additionally, eDNA fibres disappeared following RNaseH digestion only in conjunction with mild DNaseI pre-treatment (Figure 5A-B). However, mild DNaseI pre-treatment alone did not result in the loss of eDNA fibres or dissolve the biofilm, (Figure 5A). The same effect of RNaseH digestion with mild DNaseI pre-treatment was observed for the nucleic acid gel isolate from *P. aeruginosa*, as well as *P. putida* and *P. protegens* biofilms (Supplementary Figures 8A and 14C respectively). These observations further indicate that eRNA is an integral part of eDNA fibre formation in the biofilm matrix by complexing to eDNA as extracellular RNA:DNA hybrids, and hence an important contributor to *Pseudomonas* biofilm viscoelastic networks. This likely explains why the eRNA was shielded from the oligoribonucleotide smiFISH probes”.*

It is not convincing that extracellular RNA in the eDNA gel isolated from the biofilm was compared with intracellular RNA isolated from planktonic cells by RNA-seq. Based on this comparison, the authors selected *lasB* mRNA as an abundant mRNA in the biofilm matrix for further smiFISH analyses. However, extracellular RNA in the eDNA gel should be compared with intracellular RNA in the biofilm cells to reach the conclusion that some specific mRNA species were enriched in the biofilm matrix or that expression levels of mRNA in cells reflect the contents of mRNA in the biofilm matrix.

Intracellular RNA from planktonic cells was used as a base reference to identify RNA that were upregulated/essential during biofilm matrix formation, as this allowed us to make meaningful comparisons on gene expression in the extracellular matrix over time. Hence, we could normalise the level of individual eRNA on each day against the same value, which was the planktonic intracellular RNA level.

However, as per the reviewer’s suggestion we now provide this comparison between the biofilm matrix and biofilm cells. The count plot profiles of *bkDA-2*, *lpdV*, *antB* and *lasB*, shown to exist in the eDNA fibres, on days 1, 2, 3 and 5 in the biofilm matrix and biofilm cells (Figure 2B-G), demonstrate that each gene was more abundant in the biofilm matrix than biofilm cells throughout the growth assay. The counts of each of the above four genes are presented as a fraction of the total reads in each sample.

In contrast, the count plot profile of *crcZ* and *ssrA*, which were identified as intracellular and not present in the matrix, show higher counts within the biofilm cells than in the matrix throughout biofilm growth. Given that they were detected in the biofilm matrix sample, it is likely that there is some intracellular contamination. Nonetheless, these results were consistent with smiFISH results, and demonstrate that comparing counts between the biofilm matrix and either biofilm or planktonic cells can distinguish intra and extracellular RNA in these biofilms.

We have clarified this with the following text:

Commencing line 220:

*“To further investigate whether eRNA might be predicted from gene counts in the biofilm matrix, RNA from the biofilm matrix and cells within the biofilm (i.e. biofilm cells) were sequenced on days 1, 2, 3 and 5 of biofilm growth (Figure 2B-G). For putative extracellular mRNA *lasB*, *bkDA-2*, *lpdV* and *antB*, gene counts*

*standardised against total reads per sample were higher in the biofilm matrix than in biofilm cells and planktonic cells. These extracellular mRNA are therefore enriched in the biofilm matrix relative to both planktonic and biofilm cells. *ssrA* and *crcZ*, on the other hand, were higher within the biofilm cells than the biofilm matrix, yet present in similar abundance in the biofilm matrix and planktonic cells (Figure 2F-G). One explanation for this could be contamination from intracellular RNA in the biofilm matrix samples resulting from cell death.”*

Commencing line 365:

*“However, neither *ssrA* nor *crcZ* mRNA were observed to be associated with the eDNA fibres, consistent with their low abundance in the biofilm matrix extract relative to the biofilm cells. This further suggests that there is contamination of intracellular RNA in the biofilm matrix samples, possibly due to the high number of dead cells in the model biofilm. Nonetheless, the observations by smiFISH confocal microscopy of highly expressed biofilm matrix RNA along the eDNA fibres, and highly expressed biofilm cellular RNA inside cells, validates the methods described for identifying eRNA.”*

The most abundant RNA in the biofilm matrix was 23S rRNA, but not *lasB* mRNA. Given that the abundance of *lasB* mRNA was much smaller than those of the others in the biofilm matrix, it is expected that the *lasB* knockout mutants still produce a biofilm and that *lasB* mRNA is not essential for biofilm formation. It is confusing that the most abundant 23S rRNA (and 16S rRNA) in the biofilm matrix was detected only inside the cells but not on the extracellular fibers (Supplementary Figure 11 and 12). This was probably due to use of single probe for 23S rRNA and 16S rRNA (Supplementary Table 4). In contrast, multiple probes were used to detect the *lasB* mRNA (Supplementary Table 3), which could enhance the sensitivity of the detection. If multiple probes are used for 23S rRNA and 16S rRNA, colocalization of these RNA with extracellular fibers could be detected.

While 23S ribosomal RNA (PA0668.4) had the third highest base mean average across both sample sets (biofilm matrix and planktonic) the log₂-fold change in the matrix was -1.2. Hence, while 23S was highly enriched in cells, this was not the case for the matrix.

As previously mentioned, while we strived for an extraction method that was specific for eRNA, there is inevitably some presence of intracellular RNA, as indicated by the high counts of *ssrA* and *crcZ* in the matrix and their observation within cells.

However, as per the reviewer’s suggestions, we performed fluorescent *in situ* hybridisation (FISH) targeting 23s and 16s ribosomal RNA with multiple probes as per smiFISH, which showed no signal in the eDNA fibres (Supplementary Figure 12A(iii) and 12B (iii)). This further suggests that 23S rRNA is not present in the matrix eDNA network structure.

We have described this with the following text:

Commencing line 348:

“Additionally, both 16S and 23S rRNA were targeted by smiFISH using multiple probes to increase the sensitivity of detection (Supplementary Figure 12A(iii) and 12B (iii)).”

5. Based on the in-silico binding affinity study, the authors concluded that a far weaker specificity and lower binding affinity of the designed *lasB* probes to the human elastase gene. However, the authors could not eliminate the possibility of cross-reaction between the *lasB* probes and non-specific RNA species or other

components in the human samples, which should be confirmed experimentally. For instance, the human clinical samples without the *P. aeruginosa* infection can be used as a negative control. In addition, how many clinical samples were used and how many samples were positive for *lasB* mRNA on fibers?

As per the reviewer's suggestion we performed smiFISH targeting *lasB* on a human clinical sample that was not infected with *P. aeruginosa* as a negative control. The image showed no *lasB* mRNA signal in the sample (Supplementary Figure 15C)

This is described with the following text:

Commencing line 436:

"No lasB fluorescence was observed by smiFISH confocal microscopy performed on a human clinical sputum sample without P. aeruginosa infection, demonstrating the specificity for the lasB mRNA probes (Supplementary Figure 15C) to P. aeruginosa-derived lasB mRNA and not human elastase mRNA. Additionally, no eDNA fibres were observed in the non-infected sample (Supplementary Figure 15C), and lasB can only be observed when eDNA fibres were present in the sputum sample (Figure 6A)."

In total, we used three human clinical sputum samples positive for *P. aeruginosa* and one sample negative for *P. aeruginosa* infection. Out of three positive samples, two stained positive for *lasB* mRNA along the eDNA fibres, but eDNA fibres could not be detected from the third positive sample, as eDNA fibres were absent in this sample.

This is clarified with following text:

Commencing line 430:

"eDNA fibres were observed in two of the three clinical sputum samples positive for P. aeruginosa and lasB mRNA was seen only when eDNA fibres were present in the sample."

We have now replaced Figure 7 with a new Figure 6 obtained using another clinical sample positive for *P. aeruginosa* infection. The image analysis shows clear colocalisation of *lasB* mRNA to eDNA fibres, with a Mander's coefficient of colocalisation of about 56.7% (n = 8; Figure 6).

This is clarified with the change of Figure 6 and accompanying figure legend.

Commencing line 452:

"A Mander's coefficient of 56.72 % (n = 8 images) was determined for the clinical sample"

We have added a new Supplementary Figure 15B obtained using another DNaseI treated clinical sample positive for *P. aeruginosa* infection (n = 8), to illustrate the lack of *lasB* mRNA signal.

6. The major weakness of the manuscript is that it completely ignores the existing literatures on extracellular RNA in biofilms (Domenech et al. Sci. Rep. 2016; Scherbakova et al. bio-protocols 2020; Chiba et al. npj Biofilms and Microbiomes 2022).

We have addressed this shortcoming by including the reviewer's recommended references:

Commencing line 88:

“Extracellular RNA (eRNA) is present in high concentrations in eDNA-containing biofilm matrices and may also contribute to eDNA viscoelastic networks². Ribonuclease treatment inhibited eDNA-containing Haemophilus influenzae (NTHI) biofilm formation³. A structural function for RNA was also described in Staphylococcus aureus biofilms where it was found to associate with eDNA and stabilise the polysaccharide-rich matrix². Nonetheless, the lack of robust eRNA extraction protocols and its low stability⁴ has made it difficult to understand the precise structural role of eRNA and assess whether it could possibly also associate with eDNA in biofilm matrices.”

Minor comments:

1. The authors used DNaseI purchased from Sigma Aldrich (Figure. 3-7, Supplementary Figure. 7-14), but this reviewer experienced that DNaseI from Sigma Aldrich was contaminated with some protease(s). If this is also the case with DNaseI used in this study, the interpretations and conclusions of various data are severely influenced. So, the authors should confirm that no detectable proteolytic activity in DNaseI used in this study.

We have addressed this concern by performing SDS-page on a protein standard gamma globulin before and after 0.4 mg ml⁻¹ DNaseI treatment to verify the proteolytic activity of DNaseI enzyme (Figure 1). This showed no change in band length (i.e., 158 kDa) of gamma globulin before and after DNaseI treatment, suggesting no proteolytic activity of DNaseI. Here we have provided an SDS gel image.

Figure 1 (A) SDS-PAGE of gamma globulin protein standard before (second lane) and after 0.4 mg/ml DNaseI treatment (Third lane) and prestained Nu-PAGE protein ladder (ThermoFisher) (First lane).

In addition, treatment with 0.1 mg/ml DNase I did not degrade extracellular fibers in the biofilms completely. Does longer time incubation with this enzyme or use of the higher concentrations abolish the fibers?

The incubation time for DNaseI pretreatment is normally 30 min. We have now addressed the effect of longer incubation time by performing confocal microscopy on 0.1 mg ml⁻¹ DNaseI treated biofilm after prolonged incubation at 37°C for 16 h (Supplementary Figure 10A). eDNA fibres were still observed after 16 h suggesting that our short-duration DNaseI pretreatment did not degrade the biofilm matrix. The incubation period of 16 h was chosen as it was consistent with the total experimental time required for microscopy analysis.

Importantly, the negative control for the smFISH assay also showed that the fibres were preserved. This includes additional incubation time for hybridisation etc and shows that within the time scales relevant to our observations, the DNaseI treatment does not degrade the fibres.

This was described by adding Supplementary Figure 10A:

Commencing line 268:

“Nonetheless, the eDNA fibre structures were preserved even after prolonged incubation with DNaseI at 0.1 mg ml⁻¹ (16 h) (Supplementary Figure 10A).”

2. There is no or less evidence of direct interactions between extracellular RNA and eDNA. This should be confirmed by ITC, SPR, pull-down, or other biochemical methods.

We now provide direct evidence that eRNA is structurally important, because RNaseH degrades the biofilm, but only after DNA is partially degraded. RNaseH targets the RNA strand of RNA:DNA hybrids, and so this result demonstrates the presence of RNA:DNA hybrids in the matrix. Further, this shows that RNA is bound to DNA, which is consistent with our smFISH results. eRNA and eDNA are both important for biofilm viscoelasticity and the structural role of RNA:DNA hybrids is dependent on eDNA integrity.

ITC, SPR etc, would further inform the nature of the interaction, however we believe that at this stage it is sufficient to draw attention to the interactive effect of eRNA and eDNA on biofilm viscoelasticity.

We have made the following amendments

Commencing line 40: *“Thus, specific mRNA species associated with the biofilm matrix enable eDNA to form networks through extracellular DNA:RNA hybrids.”*

Line 98: *“Such information is required to understand how the eDNA network is assembled, which could help explain P. aeruginosa biofilm matrix formation, the viscoelastic phenotype of P. aeruginosa biofilms, and establish a basis for new biofilm control strategies.”*

3. This study focuses on P. aeruginosa, and there is no data about other Pseudomonas species. So, the title is too broad.

We demonstrated in our previous publication that an eDNA gel could be extracted from other Pseudomonas biofilms (i.e., P. putida and P. protegens). We provide additional evidence here that eRNA is likely important for other Pseudomonas species, where RNA transesterification also leads to

disintegration of *P. protegens* and *P. putida* biofilms. This was deduced using confocal microscopy, which illustrated the loss of eDNA fibres, as shown in Supplementary Figure 6A-B.

This is described with the following text:

Commencing line 152:

“The same response upon alkalinisation was also observed in 5-day P. protegens and P. putida biofilms (Supplementary Figure 6A-B).”

Furthermore, we demonstrate the *P. protegens* and *P. putida* biofilms are also dissolved following RNaseH digestion with DNaseI pretreatment, as discussed in the following text:

Commencing line 412:

“The same effect of RNaseH digestion with mild DNaseI pre-treatment was observed for the nucleic acid gel isolate from P. aeruginosa, as well as P. putida and P. protegens biofilms (Supplementary Figures 8A and 14C respectively). These observations further indicate that eRNA is an integral part of eDNA fibre formation in the biofilm matrix by complexing to eDNA as extracellular RNA:DNA hybrids, and hence an important contributor to Pseudomonas biofilm viscoelastic networks. This likely explains why the eRNA was shielded from the oligoribonucleotide smiFISH probes.”

4. The last paragraph in Introduction. It is unclear why understanding of the nucleic acid composition can resolve the mechanisms of eDNA release from bacteria.

We have clarified this with the following amendment:

Line 98:

“Such information is required to understand how the eDNA network is assembled, which could help explain P. aeruginosa biofilm matrix formation, the viscoelastic phenotype of P. aeruginosa biofilms, and establish a basis for new biofilm control strategies.”

5. Details of smFISH are missing in Methods. More information should be provided to reproduce the experiments. For example, conditions of fixation, permeabilization, staining, buffer compositions, etc. should be indicated.

We have included these details in the materials and methods and new Supplementary Tables 6, 7, and 8 as detailed in the following text:

Commencing line 693:

“Five-day native P. aeruginosa biofilms or clinical sputum samples were collected by gently aspirating the samples using shortened pipette tips prior to transferring to 2 ml Eppendorf tubes and washing with double distilled water. The biofilms were not fixed and permeabilised to avoid intracellular contamination. The primary probe and secondary FLAP were prehybridised using conditions described in (Supplementary Tables 6 and 7). The washed biofilms were incubated in 40 % formamide freshly prepared in 1X SSC buffer for 15 min at 22°C. During incubation, Mix 1 and Mix 2 were prepared on ice (Supplementary Table 8). A volume of 100 µl (Mix 1 + Mix 2) was sufficient for two slides (22 x 22 mm). Mix 2 was vortexed for 30 seconds. Mix 1 was added to Mix 2 and vortexed again for 30 seconds. 100 µl of the hybridisation mix was

added to the top of the slide with deposited biofilms and covered with a 10 cm Petri dish, which was wrapped in a parafilm sheet and incubated at 37°C overnight⁵. After hybridisation, biofilms were deposited gently pipetted onto the poly-L-Lysine coated microscopic slide and covered with a coverslip.”

6. Letters of scales in microscopic images should be deleted.

This has been addressed in all microscopy images.

7. Supplementary Figure 7A. The authors should use another DNA ladder maker covering sizes more than 7,000 nt. It is unclear whether reaction time was enough or not, as complete digestion of DNA and RNA had not been achieved by DNase/RNase treatment.

This has been addressed by including a new MW marker in the corresponding Supplementary Figure 8A.

8. Supplementary Figure 7B. PAO668.4 was incorrectly assigned to the right higher peak in the data of intracellular RNA.

This has been addressed in the corresponding Supplementary Figure 8B.

9. This reviewer feels that contamination of DNA less than 5% of RNA was too high. The authors should reduce DNA contamination as much as possible.

Before submitting the RNA samples for Illumina sequencing, the extracted RNA went through two rounds of turbo DNase digestion to remove traces of DNA. A 5% of DNA contamination threshold is acceptable in Illumina/NGS protocols, as the random hexamers that are added to the RNA during the library preparation will only bind to single-stranded RNA so there is little to no carry-over of DNA in the final library. Also, additional checks are performed during data analysis to ensure that DNA reads are being excluded from analysis (i.e., removing noisy reads).

10. Supplementary Table 2B. Definition of Log2 fold change is unclear.

A log2FoldChange represents the effect size estimate. This value indicates how much the gene or transcript expression have changed between the planktonic cellular RNA and biofilm matrix RNA. This value is reported on a logarithmic scale to base 2.

This has been addressed in corresponding Supplementary Table 2B.

“The log₂-fold change is the measure of extent of change in gene or transcript's expression between the biofilm matrix sample and planktonic cell RNA sample. This value is reported on a logarithmic scale to base 2. “

11. Supplementary Figure 9. More detailed explanation of the mutant strains PW7302 and PW7303 should be added.

This information is now provided in the caption of Supplementary Figure 9B-C.

“PW7302 AND PW7303 are transposon mutants derived from the mPAO1 isolate of P. aeruginosa strain PAO1. The mutant strains were generated using the transposon ISphoA/hah of 4.83kbp, with a tetracycline resistant gene (Jacobs et al PNAS 2003)“.

12. Supplementary Figure 11 and 12. The lasB mRNA is 16S rRNA or 23S rRNA within the Figure

We previously misrepresented the 16S and 23S rRNA as lasB. We apologise for the confusion and have now provided the amended Supplementary Figure 12A and B.

13. Supplementary Figure 13. How about clinical sputum left untreated with DNase I?

Using microscopy analysis, we determined that lower amounts of lasB mRNA signal in the sputum sample without DNaseI treatment. Image quantification showed < 10% of lasB mRNA detected without DNaseI treatment. We have described this by providing a figure (Figure 6B) showing DNaseI concentration-dependent response in a sputum sample as similar to our in vitro P. aeruginosa model.

This is included with the following text:

Commencing line 427:

“While some lasB mRNA could be detected without DNaseI treatment, fluorescence intensity was increased more than ten-fold following pre-treatment with DNaseI in sputum sample (Figure 6B).“

Reviewer #2 (Remarks to the Author):

To authors,

Your paper is very interesting and well written.

Please find below some comments:

- in the whole text: replace "nt" with "nucleotides".

This has been addressed throughout the manuscript.

- line 185 and 189: remove the point after the word "table".

This has been addressed throughout the manuscript.

- line 223: the name of the technique is "single molecule. inexpensive fluorescent in situ hybridisation (smiFISH)", not smFISH. Please modify it.

This has been addressed throughout the manuscript.

- line 224: please indicate the number of primary probes used for each RNA.

This has been addressed with the following text:

Line 251:

"13 lasB, RNA-specific oligoribonucleotide primary probes"

Commencing line 658:

"In total, 27 primary probes specific for ipdV, 13 primary probes specific for antB and eight probes specific for bkdA-2 were used".

Line 298: (Figure3H)

"(A) IpDv (27 primary probes) (B) antB (13 primary probes), and (C) bkdA-2 (8 primary probes)".

Line 380:

"stained with 10 and six smiFISH primary probes specific for transfer messenger RNA"

- line 225: the sentence is not clear. Please modify it with "probes pre-hybridized to a FLAP sequence labelled with two fluorophores".

This has been clarified with the following text:

Line 247:

"specific primary probes pre-hybridised to a FLAP sequence labelled with two fluorophores."

- Line 227: remove the second parenthesis after "2A".

This has been addressed in the corresponding section.

- in the whole text: please replace "smFISH" with "smiFISH"

This has been addressed throughout the manuscript.

Please find below some questions:

- The eDNA is not totally colocalized with mRNAs. How you explain the partial co-localization shown in Figures 3 and 4?

Thank you!

Partial colocalisation could result from differential mRNA expression throughout biofilm formation, likely contributing to fibre formation and functional redundancy with mRNA as a matrix component. *lasB* mRNA

was one among many extracellular RNA identified in the biofilm matrix through RNA sequencing (Figure 2B, Supplementary Table 2A). However, *lasB* mRNA was not identified in eDNA fibres in earlier biofilm development stages (Supplementary Figure 13). The partial colocalisation of *lasB* therefore indicates that other mRNAs were present in those regions. This also highlights the specificity of smiFISH probes to their respective RNA. It is also possible that the local concentration gradient of *lasB* contributes to colocalisation. Similarly, this could also apply to other extracellular mRNAs such as *bkdA-2*, *ipdV* and *antB* (Figure 3H).

We have provided the following explanation:

Commencing line 348:

“However, despite being the most abundant eRNA in the five-day biofilm, the lasB knockout mutants still produced a biofilm under the same growth conditions and the mRNA transcript profile of the lasB knockout biofilm was different to the wildtype biofilm (Supplementary Figure 14A). Thus, lasB is not essential for P. aeruginosa biofilm formation, and there is likely redundancy with regards to the mRNA P. aeruginosa uses in eDNA fibre assembly. Accordingly, lasB does not completely cover the eDNA fibres (Figure 3A-E) and other mRNAs contribute to a greater extent than lasB to eDNA fibre formation at earlier stages of P. aeruginosa biofilm growth.”

Reviewer #3 (Remarks to the Author):

Here authors identified a role of eRNA in *P. aeruginosa* biofilm matrix. eRNA was found to co-localize with eDNA fibers in the matrix and stabilize this structure. This is an interesting and well-written manuscript. My main comments are to clarify points within the main text, and perhaps perform additional controls to help support the authors conclusions. My specific comments are below.

Major comments

1. L139 - 140. Can authors quantify bacterial viability after alkaline transesterification of the biofilm, to confirm that the loss of eDNA fibers and viscoelasticity is due to disruption of eRNA and eDNA interactions and not due to cell death.

Biofilm alkalisation reduced viability by 9% of the whole population compared to just 5% for DNaseI and RNaseA treatment, as shown in the Figure 2 below.

Figure 2 Confocal micrograph of live-dead staining (Green- Syto9 and Red-Propidium iodide) of alkalinized five-day old *P. aeruginosa* biofilm (i), DNase1+ RNaseA treated five-day old *P. aeruginosa* biofilm (ii). The scale bars represent 10 μm . (iii) Fluorescent intensity quantification (n=5) for biofilm viability of non-treated, alkalinized and DNase1+RNaseA treated five-day old *P. aeruginosa* biofilm using ImageJ fiji software (n=5)

Additionally, there were many dead cells in the model biofilm prior to the treatments (8% on day 2, increasing to 68% on day 5, Supplementary Figure 7C) due it being a static biofilm with no media change. Despite this, the biofilm displayed viscoelastic behaviour on days 1-5 indicating that the behaviour was independent of the number of dead cells. This, coupled with the relatively low reduction in cell viability following alkalinisation, indicates that RNA transesterification and not cell death is responsible for disruption of eDNA-eRNA viscoelastic networks.

We have clarified this in the following text:

Commencing line 153:

“Furthermore, the biofilm displayed viscoelastic properties from days 1-5 of growth, with the number of dead cells increasing from 8 to 68% across days 2 to 5, suggesting that the viscoelastic behaviour was independent of the number of dead cells (Supplementary Figure 7A-C).”

Additionally, as described for reviewer 1 we now provide direct evidence for the structural importance of eRNA, whereby *P. aeruginosa* biofilm disintegration follows digestion with RNaseH that targets the RNA strand of RNA:DNA hybrids, as per the following text included in a new section:

Commencing line 387:

“Extracellular RNA is key to viscoelasticity of Pseudomonas biofilms

*“The agarose gel electrophoresis image of alkalinised nucleic acid gel isolate shows that the eDNA chain structure was preserved and there was a slight reduction in DNA length to 8000 bp (Supplementary Figure 14B), indicating some degradation. To assess directly whether eRNA contributes to *P. aeruginosa* biofilm viscoelasticity, RNaseA digestion of the *P. aeruginosa* biofilm was coupled with mild pre-DNase1 treatment (0.1 mg ml⁻¹), as per the smiFISH microscopy. This increased $\tan \delta$ for the biofilm relative to the untreated*

biofilm in the rheology frequency sweep (Figure 5A-C) although $\tan \delta$ remained below one indicating that viscoelasticity was preserved. However, when RNaseA was replaced with RNaseH, targeting the RNA strand of RNA:DNA hybrids, the model *P. aeruginosa* biofilm dissolved and lost its viscoelasticity, as indicated by the increase in $\tan \delta$ to greater than 1 (Figure 5C). The same effect was not observed without mild pre-DNaseI treatment for RNaseA or RNaseH, and mild DNaseI treatment (0.1 mg ml^{-1}) with RNaseH buffer alone did not dissolve the biofilm (Figure 5C) or result in the loss of eDNA fibres. Additionally, eDNA fibres disappeared following RNaseH digestion only in conjunction with mild DNaseI pre-treatment (Figure 5A-B).

Commencing line 268:

“Nonetheless, the eDNA fibre structures were preserved even after prolonged incubation with DNaseI at 0.1 mg ml^{-1} (16 h) (Supplementary Figure 10A).”

Commencing line 412:

“The same effect of RNaseH digestion with mild DNaseI pre-treatment was observed for the nucleic acid gel isolate from *P. aeruginosa*, as well as *P. putida* and *P. protegens* biofilms (Supplementary Figures 8A and 14C respectively). These observations further indicate that eRNA is an integral part of eDNA fibre formation in the biofilm matrix by complexing to eDNA as extracellular RNA:DNA hybrids, and hence an important contributor to *Pseudomonas* biofilm viscoelastic networks. This likely explains why the eRNA was shielded from the oligoribonucleotide smiFISH probes.”

2. Figure 1D. It looks like cells are mostly stained in this image. Do authors predict that the alkaline transesterification has killed most of the cells, allowing them to take up the TOTO-1? Can authors comment on this?

The live-dead stain of the five-day-old biofilm provided here shows that alkaline transesterification only reduces cell viability by 9% compared to just 5% for DNaseI and RNaseA treatment, as shown in Figure 3 below.

Figure 3 Confocal micrograph of live-dead staining (Green- Syto9 and Red-Propidium iodide) of alkalinized five-day old *P. aeruginosa* biofilm (i), DNaseI+ RNaseA treated five-day old *P. aeruginosa* biofilm (ii). The scale bars represent $10 \mu\text{m}$. (iii) Fluorescent intensity quantification ($n=5$) for biofilm viability of non-treated, alkalinized and DNaseI+RNaseA treated 5 d *P. aeruginosa* biofilm using ImageJ fiji software ($n=5$)

Additionally, the number of dead cells was only 8% on day 2, increasing to 68% on day 5, with a viscoelastic response observed throughout different days (Supplementary Figure 7C). Hence, dead cells do not explain the loss of biofilm viscoelasticity, as described with the following:

Commencing line 153:

“Furthermore, the biofilm displayed viscoelastic properties from days 1-5 of growth, with the number of dead cells increasing from 8 to 68% across days 2 to 5, suggesting that the viscoelastic behaviour was independent of the number of dead cells (Supplementary Figure 7A-C).”

3. L169 - 182. How old were the biofilms when eDNA gel was isolated? Are these results consistent across eDNA isolated at different stages of biofilm develop? Do these results correlate with findings that eDNA at later stages of biofilm development is resistant to DNase?

The biofilms were grown for 5 d. However, we provide additional data showing expression levels of the RNA investigated in this study on days 1, 2, 3 and 5, showing low levels of *lasB* mRNA upon the onset of biofilm growth. This is consistent with the smiFISH micrographs targeting *lasB*, and high expression levels of *lasB*, *bkdA-2*, *ipdV* and *antB*, and low levels of *crcZ* and *ssrA* were maintained in the biofilm matrix relative to biofilm cells, for each of the days measured (new Figure 2B-G). This is described in the following text:

Commencing line 340:

“lasB RNA only became visible in the biofilm on day 3 after the matrix was established (Figure 4A-B) and its expression was maintained across days 4 and 5 (Figure 4A-B). There was a 5% increase in lasB mRNA detection levels on day 3 compared to days 1 and 2 and their levels increased until day 4 before reducing slightly (Figure 4B). This is consistent with the increase in lasB mRNA gene counts detected in the P. aeruginosa biofilm matrix from days 1 to 3 (Figure 2B) validating the smiFISH microscopy method for eRNA quantification.”

The effect of DNaseI on mature biofilms was not addressed in this study.

4. L190 - 191. How do the RNA transcripts compare between the matrix and biofilm cells?

We have now addressed this by providing the gene counts of the mRNA used in this study, i.e., *lasB*, *bkdA-2*, *IpdV*, *antB*, *crcZ* and *ssrA* across different days (days 1,2,3 and 5) in both biofilm matrix and biofilm cells. This is addressed with following text:

Commencing line 220:

“To further investigate whether eRNA might be predicted from gene counts in the biofilm matrix, RNA from the biofilm matrix and cells within the biofilm (i.e. biofilm cells) were sequenced on days 1, 2, 3 and 5 of biofilm growth (Figure 2B-G). For putative extracellular mRNA lasB, bkdA-2, IpDV and antB, gene counts standardised against total reads per sample were higher in the biofilm matrix than in biofilm cells and planktonic cells. These extracellular mRNA are therefore enriched in the biofilm matrix relative to both

*planktonic and biofilm cells. *ssrA* and *crcZ*, on the other hand, were higher within the biofilm cells than the biofilm matrix, yet present in similar abundance in the biofilm matrix and planktonic cells (Figure 2F-G). One explanation for this could be contamination from intracellular RNA in the biofilm matrix samples resulting from cell death (Supplementary Figure 7)."*

And commencing line 364:

"Nonetheless, the observations by smiFISH confocal microscopy of highly expressed biofilm matrix RNA along the eDNA fibres, and highly expressed intracellular RNA in biofilm cells, validates the methods described for identifying eRNA."

Do authors predict to see the same enrichment of transcripts in the matrix? Or is the transcription of these genes elevated in the biofilm and will therefore form a high component of the eRNA?

As discussed, we now provide standardised counts of the four mRNA used in our study, i.e., *lasB*, *bkdA-2*, *ipdV* and *antB*, on days 1, 2, 3 and 5 in the matrix and within cells (Figure 2B-G). This shows a steady and parallel increase in gene counts, both intra- and extracellularly, up to day 3 of biofilm growth for each of these four genes. This was followed by a slight decrease in extracellular counts of these genes until day 5, suggesting that, as per the reviewer's comment, such eRNA species are elevated in the matrix. Their appearance in the matrix is further demonstrated using smiFISH with confocal microscopy.

We have described this in line 220,

Commencing line 220:

*"To further investigate whether eRNA might be predicted from gene counts in the biofilm matrix, RNA from the biofilm matrix and cells within the biofilm (i.e. biofilm cells) were sequenced on days 1, 2, 3 and 5 of biofilm growth (Figure 2B-G). For putative extracellular mRNA *lasB*, *bkdA-2*, *IpDV* and *antB*, gene counts standardised against total reads per sample were higher in the biofilm matrix than in biofilm cells and planktonic cells. These extracellular mRNA are therefore enriched in the biofilm matrix relative to both planktonic and biofilm cells. *ssrA* and *crcZ*, on the other hand, were higher within the biofilm cells than the biofilm matrix, yet present in similar abundance in the biofilm matrix and planktonic cells (Figure 2F-G). One explanation for this could be contamination from intracellular RNA in the biofilm matrix samples resulting from cell death."*

Also, we acknowledge some confusion in the terminology used in this section, and now use biofilm matrix to describe samples from our extraction targeting extracellular RNA, and biofilm cells to describe samples from our extraction targeting intracellular RNA.

5. L271 – 277. Was there a difference in the eDNA fiber structure of *lasB* mutant biofilms? Was there a difference in total eRNA concentration in *lasB* mutant biofilm

No difference in eDNA fibre structure of the *lasB* mutant was observed through confocal microscopy. However, RNA sequencing of *lasB* mutant PW7302 indicated a difference in the eRNA profile of eDNA fibres in the *lasB* mutant compared to the wildtype biofilm (Supplementary Figure 14A). The volcano plot shows extracellular RNA highly enriched in the wildtype matrix (left hand side of the plot) and extracellular RNA highly enriched in the matrix of *lasB* mutant PW7302 (right hand side of the plot). This revealed

higher enrichment of different extracellular RNA such as PA0718, PA0726, PA0723 (bacteriophage Pf1 related) in the mutant as compared to PA3724 (*lasB*, elastase), PA1494 (*muiA*, mucoidy inhibitor gene A), PA3049 (*rmf*, ribosomal modulation factor) in wildtype.

This shows that *P. aeruginosa* compensates for an absent mRNA in the matrix by modifying its RNA expression profile, suggesting that using RNA as a scaffold-building agent allows for some degree of functional redundancy.

We have clarified this with following text:

Commencing line 348:

“However, despite being the most abundant eRNA in the five-day biofilm, the lasB knockout mutants still produced a biofilm under the same growth conditions and the mRNA transcript profile of the lasB knockout biofilm was different to the wildtype biofilm (Supplementary Figure 14A).”

6. Figure S11 and S12. Perform colocalisation analysis on these images to demonstrate that the labelling localizes to the cell and not eDNA.

We now provide an updated Supplementary Figure 12A(iii) and 12B (iii) with the results of the colocalisation analyses and Mander’s coefficients for Supplementary Figure 12A (i), (ii) and 12B (i), (ii). The colocalised panel shows regions of colocalisation of FISH probes with cells. The Mander’s coefficients were 66% and 76%, for 16S rRNA at 37°C and 46°C, respectively, and 62% and 60.4% for 23S rRNA at 37°C and 46°C, respectively.

This is clarified with the following text:

Commencing line 327:

“Colocalization analysis of 16S rRNA and 23S rRNA to cells showed Mander’s coefficients of about 66% and 76% for 16S rRNA at 37°C and 46°C respectively, and 62% and 60.4% for 23S rRNA at 37°C and 46°C respectively (Supplementary Figure 12A (i), (ii) and 12B (i), (ii)). Additionally, both 16S and 23S rRNA were targeted by smiFISH using multiple probes to increase the sensitivity of detection (Supplementary Figure 12A(iii) and 12B (iii)). The absence of both 16S and 23S ribosomal RNA signals in the eDNA fibre along with negative log₂-fold change of 23S rRNA is seen in Supplementary Figure 12A(iii) and 12B (iii)), supports our observation from total RNA sequencing that mRNA but not ribosomal RNA is dominant in the extracellular biofilm.”

7. L296 – 301. Can authors clarify these results/ conclusions given that 23S RNA was identified in Figure S7B?

We thank the reviewer for helping us identify an error with our headings in our original submission. We have amended this, and the Supplementary Figure 8B now shows the base mean coverage of 23S rRNA on the Y- axis, which defines the overall expression of 23s rRNA across both planktonic intracellular and biofilm matrix samples. The 23S rRNA shows a negative log₂-fold change of -1.2 in the biofilm matrix compared to planktonic cells, which explains why they were observed in the cells through conventional FISH microscopy as seen in Supplementary Figure 12A (i),(ii) and 12B (i),(ii). We have additionally now performed smiFISH microscopy with multiple probes specific for 23S rRNA to increase the sensitivity of

detection (Supplementary Figure 12A(iii) and 12B (iii)), to provide further evidence that 23S rRNA is absent in the eDNA fibres.

We have clarified this with following text:

Commencing line 331:

“The absence of both 16S and 23S ribosomal RNA signals in the eDNA fibre along with negative log2-fold change of 23S rRNA is seen in Supplementary Figure 12A(iii) and 12B (iii)), supports our observation from total RNA sequencing that mRNA but not ribosomal RNA is dominant in the extracellular biofilm matrix of the P. aeruginosa”

8. L304 – 313. I find this section confusing. RNA-Seq identified *ssrA* and *crcZ* transcripts in the eRNA of the biofilm matrix. However, smRNA FISH identified a low abundance of these transcripts. Authors conclude that this is due to specific transport of RNA sequences into the matrix. Was the RNA-Seq performed on the biofilm cells and not the matrix eRNA? Can authors clarify these points in the text?

We acknowledge confusion regarding the use of the term ‘biofilm’ in this section. Originally, ‘biofilm’ referred to our extracellular extract (so matrix) and we did not include the comparison with biofilm cells. We now provide this comparison, as well as between the biofilm matrix and planktonic cells, and therefore introduce this terminology, and more clearly identify samples derived from the biofilm matrix, biofilm cells and planktonic cells.

With regard to the low abundance, the log2-fold change for both *ssrA* and *crcZ* was close to zero, which indicates that they were not enriched in the eRNA relative to the planktonic cells, unlike *lasB* and other genes. As discussed previously, smFISH-based observations indicated that there is also a presence of intracellular RNA, possibly due to cell death in the model system, as discussed above. Nonetheless, by comparing total counts in the matrix sample extracted by our method with total planktonic cell count, revealed which mRNA were extracellular based on log2-fold difference between biofilm matrix and either biofilm or planktonic cells. This was validated using smFISH, as is clarified with the following text:

Commencing line 220:

*“To further investigate whether eRNA might be predicted from gene counts in the biofilm matrix, RNA from the biofilm matrix and cells within the biofilm (i.e. biofilm cells) were sequenced on days 1, 2, 3 and 5 of biofilm growth (Figure 2B-G). For putative extracellular mRNA *lasB*, *bkDA-2*, *IpDV* and *antB*, gene counts standardised against total reads per sample were higher in the biofilm matrix than in biofilm cells and planktonic cells. These extracellular mRNA are therefore enriched in the biofilm matrix relative to both planktonic and biofilm cells. *ssrA* and *crcZ*, on the other hand, were higher within the biofilm cells than the biofilm matrix, yet present in similar abundance in the biofilm matrix and planktonic cells (Figure 2F-G). One explanation for this could be contamination from intracellular RNA in the biofilm matrix samples resulting from cell death.”*

Commencing line 361:

*“However, neither *ssrA* nor *crcZ* mRNA associated with the eDNA fibres, which is consistent with their low abundance in the biofilm matrix relative to the biofilm cells. This further suggests that there is some contamination of intracellular RNA in the biofilm matrix. Nonetheless, the observations by smFISH*

confocal microscopy of highly expressed biofilm matrix RNA along the eDNA fibres, and highly expressed biofilm cellular RNA inside cells, validate the methods described in this study for identifying eRNA.”

9. L340 – 346. Was there an increase in *lasB* mRNA in the eRNA between days 3 – 5? Was there an increase in eDNA content overtime? Can authors quantify this by fluorescence intensity?

We now provide standardised counts of *lasB* mRNA obtained from RNA sequencing over the different days of biofilm growth, as discussed previously and illustrated with the new Figure 2B. This shows an increase in *lasB* mRNA content over the first three days of growth before stabilising, whereas eDNA content increased from days 1 to 5, which is consistent with that described using smiFISH image quantification (Figure 4B).

This additional information is described with the following text:

Commencing line 344:

*“This corroborates the increase in *lasB* mRNA gene counts detected in the *P. aeruginosa* biofilm matrix from days 1 to 3 (Figure 2B).”*

10. L354 – 357. To clarify, *lasB* was detected in sputum without DNase treatment? Can authors comment on this difference between in vivo and in vitro. Did authors see an increase in mRNA labelling with DNase treatment? Quantify fluorescence intensity.

To clarify, we now provide *lasB* mRNA fluorescence levels in the sputum following 0, 0.05 and 1 mg ml⁻¹ DNaseI pre-treatment (Figure 6B). This shows that the *lasB* mRNA fluorescence signal without DNaseI treatment was less than 10% of total *lasB* mRNA following pre-treatment with 0.1 mg ml⁻¹ DNaseI (Figure 6B). Hence, there is a large increase in oligoribonucleotide binding following the pre-treatment.

Moreover, the increase in *lasB* mRNA signal with the DNaseI concentration was similar to our *in vitro* studies results (Figure 3F). This further suggests the presence of DNA:RNA hybrids in the extracellular matrix of COPD infected clinical sputum sample.

We have clarified this with following text:

Commencing line 427:

*“While some *lasB* mRNA could be detected without DNaseI treatment, fluorescence intensity increased more than ten-fold following pre-treatment with DNaseI in the sputum sample (Figure 6B). with a final Mander’s coefficient of colocalization of 56.72% (Figure 6A, colocalised region panel, seen in yellow).”*

11. L368. How did authors confirm that eDNA in the sputum was bacterial and not host derived?

We do not address the origin of the eDNA in this paper, and only refer to the eRNA of *Pseudomonas* complexing to eDNA fibres, and the importance of eDNA fibres for viscoelasticity.

Nonetheless, we now provide an additional Supplementary Figure 15C that eDNA fibres are absent in the non-infected control, as discussed with the following text:

Commencing line 438:

“Additionally, no eDNA fibres were found in the non-infected sample (Supplementary Figure 15C) and lasB can only be observed when eDNA fibres were present in the sputum sample (Figure 6A).”

12. L442 – 443. Can authors confirm if the centrifugation and lyophilization created artifacts in the analysis? Could this process affect viability of the cells, releasing RNA? Could this process influence the colocalization of eDNA and eRNA that was detected by microscopy?

Centrifugation and lyophilisation of biofilms were only performed for biofilm matrix extraction purposes. For microscopy analysis of biofilm cells, the native biofilm was directly collected from flasks using gentle pipetting and deposited onto the poly-L-lysine slide for further processing. This minimised biofilm disturbance and preserved the native structure of the biofilm matrix.

This has been clarified in the text, by describing the purposes of centrifugation and lyophilisation, as per the following:

Commencing line 517:

“All biofilm growth experiments were performed with three biological replicates. For extracellular nucleic acid gel isolation, NMR analysis and RNA sequencing⁶, viscoelastic biofilms that were previously shown to contain eDNA as key networking agents, were concentrated by centrifugation at 10,000 x g for 15 min in 50 ml Falcon tubes. Biofilm pellets were collected and lyophilised. For microscopy and rheology assays, the biofilms were collected directly from the Erlenmeyer flasks by gently aspirating the viscoelastic biofilm edge using a cut-off 1 ml pipette tip to minimise disturbance caused to the biofilms and specifically to preserve the native structure of biofilm matrix.”

The list of bacterial strains used in the study is included with following text:

Commencing line 503:

“P. aeruginosa wildtype PAO1, lasB knockout mutants PW7302, PW7303, polysaccharide mutants ΔpslF and ΔpelA, as well as P. putida and P. protogens strains were grown on Luria-Bertani agar plates”

Furthermore, we have reorganised the Material and Methods to describe sample preparation more clearly for eDNA and live dead staining microscopy, as per the following text:

Commencing line 636:

New section- **“Staining”**

“eDNA staining was achieved following incubation of biofilm-coated glass slides with 2 μM TOTO-1 iodide (1 mM solution in DMSO, Thermofisher Scientific, Catalog No: T3600) for 15 min in the dark. Live-dead imaging was performed according to LIVE/DEAD™ BacLight™ Bacterial Viability Kit, for microscopy (Catalog number: L7012, Thermofisher Scientific). Briefly, SYTO 9 dye, 3.34 mM (component A), 300 μl solution in DMSO and Propidium iodide, 20 mM (component B), 300 μL solution in DMSO are mixed in equal volume. Three microlitres of the mix were added to every 1 ml of bacterial cell suspension and incubated in dark at room temperature for 15 min. Five microlitres of stained suspension were deposited onto the slides and imaged using LSM780 confocal microscopy (Zeiss).”

For smiFISH microscopy methodology,

Commencing line 693:

“Five-day native P. aeruginosa biofilms and clinical sputum samples were collected by gently aspirating with a shortened 1 ml pipette tip, they were subsequently transferred to 2 ml Eppendorf tubes and rinsed with double distilled water. The biofilms were not fixed and permeabilised to avoid intracellular contamination. The primary probe and secondary FLAP were prehybridised using conditions (Supplementary Tables 6 and 7). The washed biofilms were incubated in 40% formamide freshly prepared in 1X SSC buffer for 15 min at 22°C. During incubation, Mix 1 and Mix 2 were prepared on ice (Supplementary Table 8). Mix 2 was vortexed for 30 s. Mix 1 was added to Mix 2 and vortexed again for 30 s. One hundred microlitres of the hybridisation mix were added to slides with deposited biofilms and covered with a Petri dish, wrapped in parafilm and incubated at 37°C overnight⁵. After hybridisation, biofilms were deposited onto the poly-L-Lysine coated microscopic slide using gentle pipetting and covered with a coverslip.”

13. L546 – 562. In line with the above comment, did authors perform microscopy on native biofilms? Or only on the processed biofilms? If so, how did authors stain the lyophilized samples?

We performed all microscopy-related experiments on native biofilms without lyophilisation. We have clarified this under the headings of biofilm growth, staining and sample and probe preparation respectively (lines 543, 648 and 693). Lyophilised samples were used only for biofilm matrix and nucleic acid gel extraction purposes, as water should be removed completely from biofilms to solubilise and extract the matrix when using ionic liquid as a solvent.

14. Methods section. Include number of biological and technical replicates for all assays.

We have included this information in the methods section. Briefly, all experimental procedures were performed on biological replicates and the findings were consistent and reproducible in all replicates. *In vitro* wildtype and mutant *P. aeruginosa* biofilms were collected from three biological replicates for confocal microscopy and RNA sequencing. Due to limited availability of clinical samples, confocal microscopy was performed on three technical replicates for each of the three *P. aeruginosa* infected sputum samples.

Composition and quantification of purified nucleic acid isolate from biofilms were performed using a one-dimensional (1-D) NMR study across three biological replicates. Detailed structural analysis of nucleic acid higher order structures and presence of mononucleotides were resolved using two-dimensional NMR across a single replicate, and this validated spectrum was representative of all biological replicates using the 1-D spectra.

The following text has been provided:

Line 517:

“All biofilm growth experiments were performed on three biological replicates.”

Line 547:

“All NA gel isolate and enzymatic digestion experiments were performed on three biological replicates.”

Line 594:

“Biofilms were grown under model static conditions in biological triplicate, and sampled on days 1, 2, 3 and 5 for extraction of intracellular and extracellular RNA. “

Commencing line 604:

“Qualitative and quantitative analyses of the purified nucleic acid isolate from biofilms were undertaken from one-dimensional NMR analyses across three biological replicates, while the detailed structural analysis was achieved using two-dimensional NMR on a single.”

Line 633:

“RNA sequencing was performed on three biological replicates.”

Commencing line 674:

“All microscopy imaging was performed across three biological replicates, except for the non-infected sputum sample where three technical replicates were used.”

Commencing line 430:

“eDNA fibres were observed in two of the three clinical sputum samples positive for P. aeruginosa.”

Additionally, we introduce to the reviewer four different terminologies (planktonic RNA, biofilm cellular RNA, biofilm matrix RNA and nucleic acid gel isolate) and their extraction procedures for easier understanding.

The above four terms are segregated into three separate sections in Materials and Methods with the following text:

Lines 556:

“Extracellular nucleic acid gel isolation”

Line 563:

“Planktonic and biofilm cell RNA extraction for sequencing”

Line 578:

“Biofilm matrix RNA extraction for sequencing”

Minor comments

1. L61. Clarify the statement ‘eDNA from the cytosol into membrane vesicles. This sounds that eDNA is only released into membrane vesicles by ECL. eDNA is also released into the extracellular environment and incorporated into the biofilm matrix through this mechanism.

We have addressed this with the following text:

Line 56:

“It was thus shown that a cryptic prophage endolysin activated explosive cell lysis, resulting in the release of eDNA from the cytosol into the extracellular biofilm matrix environment^{7, 8.}”

2. L67. Clarify that later biofilms are resistant to DNase

We did not observe the DNase resistance in mature biofilms. We have modified the previously described sentence:

Line 63:

“This viscoelastic behaviour is completely removed for P. aeruginosa biofilms in early stages of growth by DNaseI treatment, resulting in biofilm dissolution^{9,6.}”

3. Figure 2. Suggest rephrasing the title and legend of the Figure as mRNA is not located on the chromosome.

We have rephrased this as follows:

Line 230:

“The abundance of specific mRNA transcripts increases in the extracellular matrix of P. aeruginosa biofilms relative to planktonic cells”

4. L215. What are the gene names of these PA numbers?

These are identified in the following text:

Line 237:

“PA3724 (lasB), PA2250 (IpdV), PA2248 (bkdA-2) and PA2513 (antB),”

5. Figure S10 is mentioned after S11 and S12.

We have addressed this in the main text (i.e., all Supplementary Figures have been renumbered, reorganised and now appear in the correct sequence)

6. Figures S11 and S12. Images are labelled with *LasB* mRNA rather than 16S and 23S.

We have addressed in the main text.

7. Figure 5. Is the 10um scale bar also consistent for the zoomed insets?

This has been amended.

8. L519. Is this subheading correct?

To clarify, we have made the following heading amendment:

Line 525:

“Biofilm and nucleic acid gel alkalisation for NMR, rheology and agarose gel electrophoresis”

Reviewer #4 (Remarks to the Author):

The manuscript by Mugunthan et al. entitled “Extracellular RNA is a key component of viscoelastic eDNA networks in *Pseudomonas* biofilm” is based on the discovery of extracellular RNA (eRNA) being present in the biofilm matrix, with the eRNA being complexed with eDNA providing a viscoelastic structure. The authors also emphasize the presence of specific mRNA species with the biofilm matrix. While the notion of extracellular RNA being a component of the biofilm matrix is exciting, the data are not that convincing. For example, the authors use confocal microscopy/smFISH to visualize eRNA but the visualization of mRNA species is only qualitative and appears to be inconsistent (spotty).

We thank Reviewer 4 for their positive comments and have addressed concerns regarding the clarity of the microscopy with a series of eDNA and *lasB* mRNA quantification and control experiments, as detailed below.

Moreover, proper smiFISH controls are missing (e.g. using mutant strains and scrambled probes, etc).

We now provide additional controls involving scrambled oligo probes against the four highly enriched extracellular RNA used in this study (*lasB*, *bkdA2*, *lpdV* and *antB*) to validate their specificity (Supplementary Figure 11). The outcome is described with the following text:

Commencing line 321:

“Additionally, no signal was observed from scrambled probes of extracellular mRNA such as lasB, lpdV, bkdA-2 and antB in eDNA fibres of P. aeruginosa biofilms (Supplementary Figure 11).”

lasB scrambled probes were also tested on clinical samples to validate the specificity of *lasB* mRNA probes to their target *lasB* mRNA in COPD clinical sputum samples (Supplementary Figure 15D). The outcome is described with the following text.

Commencing line 441:

“The lasB scrambled probes did not give a fluorescent signal in the P. aeruginosa-positive sputum sample (Supplementary Figure 15D).”

lasB mRNA probes were previously tested on two *lasB* mutant strains (PW7302 and PW7303) (Supplementary Figure 9B-C). This showed that the *lasB* mRNA signal was absent in the mutant strain, further indicating probe specificity. We emphasise this point with the following text:

Commencing line 259:

“Furthermore, smiFISH performed on lasB knockout mutants with the lasB-specific oligo probes showed no fluorescent signal in the biofilm extracellular matrix (Supplementary Figure 9B-C).”

A clinical sputum sample negative for *P. aeruginosa* is now included in the study and was tested for cross reactivity with the *lasB* mRNA probe. The absence of *lasB* mRNA in the above sample, further demonstrates the specificity of *lasB* mRNA probe (Supplementary Figure 15C) and provides additional and critical data supporting our claims regarding the presence of *lasB* mRNA in the eDNA fibres of clinical sputum sample positive for *P. aeruginosa* (Figure 6A), as described here:

Commencing line 436:

“No lasB fluorescence was observed by smiFISH confocal microscopy performed on a human clinical sputum sample without P. aeruginosa infection, demonstrating the specificity for the lasB mRNA probes (Supplementary Figure 15C) to P. aeruginosa-derived lasB mRNA and not human elastase mRNA. Additionally, no eDNA fibres were observed in the non-infected sample (Supplementary Figure 15C), and lasB can only be observed when eDNA fibres were present in the sputum sample (Figure 6A).”

We also provide the quantity of *lasB* mRNA and eDNA fibres on days 2, 3, 4 and 5 of biofilm growth as determined by the fluorescent signal intensity (Figure 4B).

Commencing line 342:

“There was a 5% increase in lasB mRNA detection levels on day 3 compared to days 1 and 2, and its level increased until day 4 before reducing slightly (Figure 4B). Further, eDNA levels consistently increased across these days. This corroborates the increase in lasB mRNA gene counts detected in the P. aeruginosa biofilm matrix from days 1 to 3 (Figure 2B).”

We previously quantified the percentage of mRNA colocalisation to eDNA fibres in terms of the Mander's coefficient of colocalisation. We submit that this method is robust and an efficient way of quantifying the percentage of colocalisation, as outlined below:

We have previously provided information on Mander's coefficient for different RNA as follows:

Commencing line 278:

“Mander’s coefficients for pre-treatment with 0, 0.01, 0.05 and 0.1 mg ml⁻¹ DNase1 were 0%, 8.6±2%, 29±7% and 49±1%, respectively (Figure 3G), further indicating the DNase1 concentration dependent binding of the lasB oligoribonucleotide probes to the eDNA fibres.”

Commencing line 300:

“Mander’s coefficients for lpdv, antB and bkdA-2 of 16± 5%, 11± 3%, and 18± 6% respectively (n = 5 images each) were determined based on the overlap of the mRNA signal with TOTO-1 stained green eDNA fibres (colocalised region panel at bottom in yellow)”

Commencing line 375:

“Mander’s coefficients for day-1, 2, 3, 4 and 5 biofilms were 0%, 0%, 14 ± 2%, 22± 4%, and 24 ± 8% respectively (n = 5 images each) based on the overlap of the mRNA signal with TOTO-1 stained green eDNA fibres (colocalised region panels in yellow).”

Line 453:

“A Mander’s coefficient of 56.72 % (n = 8 images) was determined for the clinical sample based on the overlap of the lasB mRNA signal with TOTO-1 stained green eDNA fibres (colocalised region panel in yellow).”

It also appears that the conditions used by the authors result not just in eDNA, but also bacterial cells, to be stained by cell impermeable stain TOTO-1.

As discussed above, staining of the cells by TOTO-1, as well as the observation of *ssrA* and *crcZ* mRNA by smiFISH within the cells only were achieved, despite the fact they were detected in the matrix samples as well. Nonetheless, this study suggests that while there is some contamination with intracellular RNA, the eRNA extraction method does enable us to identify eRNA based on log₂-fold differences with planktonic cells, as verified by smiFISH for eRNA (*lasB* etc) and intracellular RNA (*ssrA* and *crcZ*), which we have explained as follows:

Commencing line 225:

“ssrA and crcZ, on the other hand, were higher in the biofilm cells than biofilm matrix, yet present in similar abundances in the biofilm matrix and planktonic cells (Figure 2F-G). One explanation for this could be contamination from intracellular RNA in the biofilm matrix samples.”

And commencing line 362:

“However, neither ssrA nor crcZ mRNA associated with the eDNA fibres, which is consistent with their low abundance in the biofilm matrix relative to the biofilm cells. This further suggests that there is some contamination of intracellular RNA in the biofilm matrix. Nonetheless, the observations by smiFISH confocal microscopy of highly expressed biofilm matrix RNA along the eDNA fibres, and highly expressed biofilm cellular RNA inside cells, validate the methods described in this study for identifying eRNA.”

Additionally, cell death resulting from alkalisation does not explain the viscoelastic response of the biofilm, as indicated by the slight reduction in viability following alkalisation and enzymatic treatment

(9% and 5 % decrease respectively, extract from Figure 4), and the high numbers of dead cells in the five-day-old model biofilm (68%).

Figure 4 Confocal micrograph of live-dead staining (Green- Syto9 and Red-Propidium iodide) of alkalinized five-day old *P. aeruginosa* biofilm (i), DNase1+ RNaseA treated five-day old *P. aeruginosa* biofilm (ii). The scale bars represent 10 μ m. (iii) Fluorescent intensity quantification (n=5) for biofilm viability of non-treated, alkalinized and DNase1+RNaseA treated five-day old *P. aeruginosa* biofilm using ImageJ Fiji software (n=5)

It is worth noting, however, that there were also many viable cells in the biofilm. This point is clarified with the following text.

Commencing line 153:

“Furthermore, the biofilm displayed viscoelastic properties from days 1-5 of growth, with the number of dead cells increasing from 8 to 68% across days 2 to 5, suggesting that the viscoelastic behaviour was independent of the number of dead cells (Supplementary Figure 7A-C).”

Additional concerns are raised by the data presentation, as Figures are incorrectly referenced throughout the manuscript, and the terms “biofilm” and “biofilm matrix” are frequently used interchangeably.

We acknowledge confusion regarding the use of the term ‘biofilm’ in this section. Originally, ‘biofilm’ referred to our extracellular extract (i.e., the matrix) and we did not include the comparison with biofilm cells. We now distinguishing the biofilm matrix from planktonic cells, and more clearly identify samples derived from the matrix, biofilm cells and planktonic cells. The RNA extraction procedures from these different sources are included for clarity.

The above terms are separated into three sections in Materials and Methods with the following text:

Line 557:

“Extracellular nucleic acid gel isolation”

Line 564:

“Planktonic and biofilm cell RNA extraction for sequencing”

Line 579:

“Biofilm matrix RNA extraction for sequencing”

Additional comments and concerns are given below.

Other comments

1. In support of RNA being a component of the biofilm matrix, the authors provide 5 Figures at the beginning of the manuscript, however, they are all Supplementary Figures. It would be nice if the first evidence for eRNA is Figure 1 instead of Supplementary Figures 1-5.

To address these additional comments, we have reorganised the first section of the results text such that the first evidence for eRNA is the $1\text{H}-^{13}\text{C}$ HSQC, showing the existence of eight 2' and 3' monoribonucleotides resulting from alkaline transesterification, along with the chain eDNA.

This has been described in following text:

Commencing line 106:

“Total correlation spectroscopy (TOCSY) and $^1\text{H}-^{13}\text{C}$ heteronuclear single quantum coherence (HSQC) NMR can identify proton NMR correlations within individual ribose sugars and their proton-carbon single bond correlations, respectively. Using this approach, we showed that at pH 12 NA peaks dominate the solution $^1\text{H}-^{13}\text{C}$ HSQC spectra for the alkalinised NA gel isolate. The NA spectrum was resolved as two clusters of sugar proton peaks ($\text{C}1'-\text{H}1'$), according to differing correlations to neighbouring carbons ($\text{C}2'-\text{H}1'$) (Figure 1A). The first cluster (rectangles), with $\text{C}2'$ chemical shift values of ~ 40 ppm, represent deoxyribose (i.e., DNA) and the second cluster (ovals) with $\text{C}2'$ shifts of ~ 70 ppm, are associated with ribose sugar conformations (i.e., RNA). “

2. smFISH data, Please provide quantitative data as well. Additionally, include controls (e.g. using mutant strains, scrambled probes)

As per the reviewer's suggestions, we now provide the abundance of relevant mRNA (*lasB*, *bkdA-2*, *lpdV*, *antB*, *ssrA* and *crcZ*) in the biofilm matrix (Figure 2(B-G) and Figure 3I and 4D), as well as the raw integrated fluorescence intensity of *lasB* mRNA in both *in vitro* PAO1 biofilm and COPD clinical sputum sample (Figure 3F, 4B and 6B respectively), as determined from microscopy images.

The image quantification indicated that there were about 22% *crcZ* and 26% *ssrA* positive cells in the five-day-old biofilms (Figure 4D). This has been described with the following figure 4D legend text:

Line 359:

*“(F) smiFISH confocal microscopy performed on five-day biofilms using 10 and six oligo primary probes targeting *ssrA* and *crcZ* mRNA respectively, showed about 26% and 22% of *ssrA* and *crcZ* mRNAs positive cells within the biofilms (seen in yellow in Figures 4C (i) ROI and 4C (ii), ROI respectively and 4D”*

Additional validation is provided for mRNA probe specificity for target RNA, with scrambled probes (designed using GenScript online tool) for the highly enriched extracellular RNA (*lasB*, *bkDA-2*, *lpdV*, *antB*) (Supplementary Figure 11) that did not bind to the target RNA, as explained in the following text.

Commencing line 322:

“Additionally, no signal was observed from scrambled probes of extracellular mRNA such as lasB, lpdV, bkDA-2 and antB in eDNA fibres of P. aeruginosa biofilms, validating the specificity of primary probes to the target mRNA (Supplementary Figure 11).”

Commencing line 442:

“The lasB scrambled probes did not produce a fluorescent signal in the P. aeruginosa-positive sputum sample (Supplementary Figure 15D).”

In our original submission we also validated the specificity of *lasB* mRNA probe by using *lasB* transposon mutants (PW7302 and PW7303). We have included the text here:

Commencing line 259:

“Furthermore, smiFISH performed on lasB knockout mutants with the lasB-specific oligo probes showed no fluorescent signal in the biofilm extracellular matrix (Supplementary Figure 9B-C).”

Finally, we also observed a fluorescent intensity increase in the *lasB* mRNA across different days in *P. aeruginosa* biofilms (days 2 to 5) (Figure 4B) showing that the *lasB* mRNA levels identified from sequencing (Figure 2B) correlate with the intensity and coverage of smiFISH fluorescence, as explained in the following text:

Commencing line 345:

“This corroborates the increase in lasB mRNA gene counts detected in the P. aeruginosa biofilm matrix from days 1 to 3 (Figure 2B).”

Commencing line 365:

“Nonetheless, the observations by smiFISH confocal microscopy of highly expressed biofilm matrix RNA along the eDNA fibres, and highly expressed biofilm cell RNA inside cells, validate the methods described for identifying eRNA.”

3. I. 354-357 The authors state that “... LasB mRNA also appeared in the extracellular matrix of the clinical sample (Figure 7, red channel). The sputum sample had fewer cells staining positive for lasB mRNA”. However, I am having a hard time detecting a difference. I highly recommend presenting the data in a quantitative manner. The qualitative approach chosen here is not convincing.

The previous Figure 7 now becomes Figure 6. We have improved the resolution in this new Figure 6 to more clearly show that *lasB* mRNA colocalises with eDNA fibres in the *P. aeruginosa*-positive sputum sample.

Additionally, by image quantification, we observed an increase in the *lasB* mRNA signal across different DNaseI concentrations in both *in vitro* *P. aeruginosa* model and clinical sputum samples positive for *P. aeruginosa* (Figure 3F, 4B and 6B respectively).

LasB mRNA quantification across different days of biofilm growth in the *in vitro* PAO1 model is included, with following text:

Commencing line 272:

“The increasing intensity of the lasB RNA-specific oligoribonucleotide probe upon DNaseI treatment, where lasB intensity increases five-fold upon DNaseI pre-treatment (Figure 3F), suggests that higher concentrations of DNaseI enzyme (0.1 mg ml⁻¹) improve the accessibility of the lasB mRNA probe to eRNA.”

lasB mRNA quantification of human sputum treated with DNaseI is discussed as follows:

Commencing line 428:

“While some lasB mRNA could be detected without DNaseI treatment, fluorescence intensity increased more than ten-fold following pre-treatment with DNaseI in sputum samples (Figure 6B) with a final Mander’s coefficient of colocalization of 56.72% (Figure6A, colocalised region panel, seen in yellow).”

We also provide a negative control for the sputum sample (i.e., no *P. aeruginosa* infection) (Supplementary Figure 15C) with the following description:

Commencing line 437:

“No lasB fluorescence was observed by smiFISH confocal microscopy performed on a human clinical sputum sample without P. aeruginosa infection, demonstrating the specificity for the lasB mRNA probes (Supplementary Figure 15C) to P. aeruginosa-derived lasB mRNA and not human elastase mRNA. Additionally, no eDNA fibres were observed in the non-infected sample (Supplementary Figure 15C), and lasB can only be observed when eDNA fibres were present in the sputum sample (Figure 6A).”

We have amended the original statement in line 447 from “*lasB* mRNA also appeared in the extracellular matrix of the clinical sample (Figure 6, red channel)” to “*lasB* mRNA is present in the eDNA fibres of the clinical sample”

4. What is a “extracellular nucleic acid gel”. Please define

The extracellular nucleic acid isolate is the material recovered from a five-day-old *P. aeruginosa* following ionic liquid extraction, HPLC purification, and polishing, which assembles into a viscoelastic gel material when returned to water. This was defined in our previous publication (include citation #6 here), which we have clarified in the current text as follows:

Commencing line 103:

“Using this method, our model biofilm, as well as the extracellular nucleic acid (NA) gel recovered from the biofilm (i.e. NA gel isolate)⁶, were dissolved.”

5. L. 105. 1H-13C not shown in Supplementary Figure 1.

The full length 1H-13C HSQC of extracellular nucleic acid gel isolate is now provided in Supplementary Figure 1.

6. L. 141 The authors state that “there was also a loss in biofilm viscoelasticity (Supplementary Figure 6). “However, the data shown in Supplementary Figure 6 do not support loss of viscoelasticity upon transesterification of RNA/loss of DNA. The authors should consider revising the Figure Legend and/or data labels.

We have addressed the reviewer’s concern by now providing rheology data measuring viscoelasticity after biofilm alkalinisation. Such measurements showed an increase in $\tan \delta$ value greater than 1, indicating that the loss modulus is greater than the storage modulus, which is the typical profile of fluids (Supplementary Figure 5B, C), demonstrating a loss of viscoelasticity for biofilms after alkalinisation. This is described in the revised manuscript as follows:

Commencing line 149:

“Alkalinization increased $\tan \delta$ in the rheology frequency sweep for the biofilm from 0.2 to greater than 1 (Supplementary Figure 5C), indicating that the loss modulus subsequently exceeded the storage modulus, and that removal of eDNA fibres is coincident with loss of viscoelasticity.”

7. I. 144. The authors state that “RNA transesterification upon alkalinisation was associated with biofilm and matrix dissolution”. However, no evidence has been provided for biofilm dissolution or matrix dissolution.

We now provide rheograms of the wildtype biofilm before and after RNA transesterification showing that G'' (loss modulus) only exceeds G' (storage modulus), hence increasing $\tan \delta$ greater than 1 across different frequency intervals, following transesterification. This shows that RNA transesterification was associated with a loss of biofilm viscoelasticity, demonstrating that the biofilm was dissolved.

Furthermore, biofilm confocal micrographs in the original submission showed that RNA transesterification leads to loss of eDNA network (Figure 1D).

We have added the new supplementary data with the following text:

Commencing line 149:

“Alkalinization increased $\tan \delta$ in the rheology frequency sweep for the biofilm from 0.2 to greater than 1 (Supplementary Figure 5C), indicating that the loss modulus subsequently exceeded the storage modulus, and that removal of eDNA fibres is coincident with loss of viscoelasticity.”

Crucially, we also provide direct evidence that RNA contributes to the biofilm structure, whereby RNaseH digestion, targeting the RNA strand in DNA:RNA hybrids, dissolved the biofilm, as described in response to Reviewer 1 (Figure 5A-C, Supplementary Figure 14C and Supplementary Figure 8A, lane 6-8).

8. I. 186-189 Some of the mRNA transcripts listed here appear to be decreased rather than enriched in the matrix

Some of the mRNA transcripts mentioned in lines 186-189 (original manuscript), such as *rnpB* and three bacteriophage related proteins (PA0640, PA0621, PA0622), have negative log₂-fold changes. This suggests a reduced expression in biofilm matrix compared to planktonic cells.

Higher base mean average is an indication of their enrichment in planktonic cells compared to the biofilm matrix.

We have clarified this with the following text:

Commencing line 195:

“RNA sequencing revealed several mRNA transcripts to be highly enriched in RNA from planktonic P. aeruginosa cells (i.e. planktonic) and the biofilm matrix extracted from five-day P. aeruginosa biofilm, as indicated by the high base mean average (Figure 2A, Supplementary Figure 8B, Supplementary Table 2B).”

Additionally, the *ssrA* and *crcZ* mRNA count plot (Figure 2F-G) showed these mRNA transcripts are highly enriched in the biofilm cells compared to the matrix.

9. L. 197, enriched in biofilm or biofilm matrix, please clarify. Please also make sure that the correct Figures and tables are referred to in this section

“Biofilm matrix” is now used here and throughout the manuscript.

10. L. 211 what is the meaning of “RNA sequenced from the extracellular matrix of five-day-old P. aeruginosa biofilms and the chromosomes of planktonic P. aeruginosa cells”.

We apologise for the confusion. We have extracted RNA from both the biofilm matrix and planktonic cells of *P. aeruginosa*. Volcano plots were generated by sequencing extracted RNA to compare the highly enriched RNA across both the biofilm matrix and planktonic cells.

We have clarified the above statement with the following text:

Commencing line 234:

“(A) volcano plot of total RNA sequenced from the biofilm matrix of five-day old P. aeruginosa biofilms and planktonic P. aeruginosa cells (16 h)”

11. L. 242-244 recommend showing data as graph

As per reviewer’s suggestion, we now include an extra component to Figure 3 (Figure 3G) showing a bar graph for Mander’s coefficient of colocalisation of *lasB* mRNA to eDNA fibre across different DNaseI concentrations.

We have clarified the above suggestion with the following explanation:

Commencing line 278:

“Mander’s coefficients for pre-treatment with 0, 0.01, 0.05 and 0.1 mg ml⁻¹ DNase1 were 0%, 8.6±2%, 29±7% and 49±1%, respectively (Figure 3G), further indicating the DNase1 concentration dependent binding of the lasB oligoribonucleotide probes to the eDNA fibres).”

12. L. 298 text does not match content shown in Suppl Figure 12

We have amended the image caption. Previous Supplementary Figures 10 and 11 have now been changed to Supplementary Figure 12A and B.

13. L. 335-336, values don’t match data; authors provide 6 values for 5 time points

We apologise for the mistake and have included the data as Figure 3G.

We have rectified this with the following text:

Commencing line 375:

“Mander’s coefficients for day 1, 2, 3, 4 and 5 biofilms were 0%, 0%, 14 ± 2%, 22± 4%, and 24 ± 8% respectively (n = 5 images each) based on the overlap of the mRNA signal with TOTO-1 stained green eDNA fibres (colocalised region panels in yellow).”

14. Figure 3 please provide means of presenting data in a quantitative manner

We have addressed this by quantifying *lasB* mRNA detection in eDNA fibre as a result of DNase1 treatment across different concentrations. While *lasB* was not detected without DNase1 treatment, there was a five-fold increase in *lasB* mRNA detection upon 0.1 mg ml⁻¹ DNase1 pretreatment. Quantification was performed using ImageJ Fiji software by calculating raw integrated fluorescence intensity of *lasB* mRNA and eDNA for each condition.

We provide this additional data in the following explanation:

Commencing line 272:

“The increasing intensity of the lasB RNA-specific oligoribonucleotide probe upon DNase1 treatment, where lasB intensity increases five-fold upon DNase1 pre-treatment (Figure 3F), suggests that higher concentrations of DNase1 enzyme (0.1 mg ml⁻¹) improve the accessibility of the lasB mRNA probe to eRNA.”

15. Figure 5 no eDNA fibers are visible, Instead, only bacterial cells are shown despite DNase1 treatment. Please clarify why the appearance of the biofilm (matrix) is so different from those shown in previous Figures Also, please clarify if the samples are from planktonic or biofilm cells. Also, please provide means of presenting data in a quantitative manner

We now provide a revised Figure 4C that shows cells stained with *crcZ* and *ssrA* oligoribonucleotide probes in the presence of eDNA fibres.

The samples were collected from five-day-old biofilms.

We have now clarified this with the following text:

Line 359:

“smiFISH confocal microscopy performed on the five-day old biofilms”

As per the reviewer’s suggestions, we have quantified of *ssrA* and *crcZ* positive cells using ImageJ Fiji software, calculating the maxima points based on the fluorescence intensity. We identified ~22% *crcZ* positive cells and ~26% *ssrA* positive cells (Figure 4D). The detailed information on the method of quantification is described in the materials and method section.

New section,

Commencing line 678:

“Image quantification:

*For quantification of eDNA and extracellular mRNA quantification in biofilms across different days and different DNaseI concentrations, total fluorescent intensity was obtained from images by calculating the raw integrated intensity using ImageJ. The Mander’s coefficient, which measured the fraction of colocalization between two fluorescence channels, was calculated by building a colocalization channel in Imaris x64 software. This indicated the percentage of the mRNA signal (red channel) that was colocalised with the eDNA or cell signals (green channel). The percentage of live to dead population was calculated by measuring total fluorescent intensity using ImageJ and *ssrA* and *crcZ* mRNA positive cells in the biofilms were calculated based on the “Find Maxima” function in ImageJ.”*

We have now provided the additional quantification data with the following text:

Figure 4C legend, commencing line 380:

*“Confocal micrograph of DNaseI (0.1 mg/ml) pretreated five-day-old P. aeruginosa biofilms stained with 10 and six smiFISH primary probes specific for transfer messenger RNA (tmRNA) *ssrA* and (B) *crcZ* respectively showing 22% of cells with *crcZ* and 26% with *ssrA*.”*

16. Figure. 1A-B, not convincing evidence of RNA being present in matrix

Solution state NMR spectroscopy is incontrovertible with regards to describing the biomolecular content in solution. This showed, through the cross-referencing of 1-D ¹H NMR and 2-D ¹H-¹³C HSQC, ¹H-¹³C HSQC-TOCSY and ¹H-³¹P HETCOR spectra, that eRNA in the nucleic acid gel isolate underwent alkaline transesterification.

smiFISH targeting the mRNA detected in our nucleic acid gel isolate was used to verify the presence of eRNA in the biofilm matrix and overlap with eDNA structural fibres.

Additionally, we demonstrate that these eDNA structural fibres disappear, along with biofilm viscoelasticity, upon RNA alkaline transesterification (Figure 1C-D and Supplementary Figure 5A, B and C)

Figure 1A-B does not, however, demonstrate that eRNAs are necessarily responsible for the monoribonucleotide peaks observed as a result of transesterification. However, we can now directly implicate eRNA as a key structural component of the biofilm by providing additional data showing that

biofilm disintegration occurs upon RNaseH treatment of the biofilm, whereby RNaseH targets the RNA strand of RNA:DNA hybrids (Figure 5A-C and Supplementary Figure 14C). This is explained in the new text added to the existing section titled:

Line 388: ***“Extracellular RNA is key to viscoelasticity of Pseudomonas biofilms”***

Commencing line 396:

“However, when RNaseA was replaced with RNaseH, targeting the RNA strand of RNA:DNA hybrids, the model P. aeruginosa biofilm dissolved and lost its viscoelasticity, as indicated by the increase in $\tan \delta$ to greater than 1 (Figure 5C).”

Commencing line 415:

“These observations further indicate that eRNA is an integral part of eDNA fibre formation in the biofilm matrix by complexing to eDNA as extracellular RNA:DNA hybrids, and hence an important contributor to Pseudomonas biofilm viscoelastic networks. This likely explains why the eRNA was shielded from the oligoribonucleotide smiFISH probes.”

17. Figure 1C-D, Figure D does not appear to show eDNA fibers. Unclear why in Figure 1C-D bacterial cells also appear to be stained with the cell impermeable dye TOTO-1

Figure 1D is the image of the biofilm after RNA alkaline transesterification and was included to demonstrate that the loss of biofilm elasticity with RNA transesterification occurs coincident with the loss of eDNA fibres. Hence, Figure 1D highlights their absence, as clarified in the following:

Commencing line 149:

“Alkalinisation increased $\tan \delta$ in the rheology frequency sweep for the biofilm from 0.2 to greater than 1 (Supplementary Figure 5C), indicating that the loss modulus subsequently exceeded the storage modulus, and that removal of eDNA fibres is coincident with loss of viscoelasticity.”

As per our response to Reviewer 3, the high number of dead and viable cells in our biofilm reflects the static biofilm model used. To clarify this, we have quantified the live-dead cells of the day-5-old biofilm before and after alkaline transesterification, showing only a 9% reduction in biofilm viability (excerpt from Figure 5, below) following alkalinisation, suggesting that the loss of biofilm viscoelasticity is unlikely to be due to cell death.

Figure 5 Confocal micrograph of live-dead staining (Green- Syto9 and Red-Propidium iodide) of alkalinized five-day-old *P. aeruginosa* biofilm (i), DNase1+ RNaseA treated five-day old *P. aeruginosa* biofilm (ii). The scale bars represent 10 μm . (iii) Fluorescent intensity quantification (n=5) for biofilm viability of non-treated, alkalinized and DNase1+RNaseA treated five-day old *P. aeruginosa* biofilm using ImageJ fiji software (n=5)18.

Additionally, we have also quantified the percentage of dead cells across day-2 and day-5 biofilm showing an increase from 8% to 68% (Supplementary Figure 7A-C), as described in the manuscript as follows:

Commencing line 153: *“Furthermore, the biofilm displayed viscoelastic properties from days 1-5 of growth, with the number of dead cells increasing from 8 to 68% across days 2 to 5, suggesting that the viscoelastic behaviour was independent of the number of dead cells (Supplementary Figure 7A-C).”*

18. Figure 7 control without DNase treatment is missing

As per the reviewer’s suggestion, we performed smiFISH confocal microscopy on the *P. aeruginosa*-positive clinical sputum sample without DNaseI treatment. The images show low levels of *lasB* mRNA signal without DNaseI treatment, which we now describe as less than 10% of the total *lasB* mRNA fluorescence signal intensity following pre-treatment with the maximum DNaseI concentration (0.1 mg ml^{-1}) (Figure 6B).

We have provided a more detailed explanation of these results in response to Reviewer 3’s comments.

This is addressed with the following text:

Commencing line 428:

*“While some *lasB* mRNA could be detected without DNaseI treatment, fluorescence intensity increased greater than ten-fold following pre-treatment with DNaseI in sputum samples (Figure 6B) with a final Mander’s coefficient of colocalization of 56.72% (Figure6A, colocalised region panel, seen in yellow)”*

19. Supplementary Table 1. Transcript abundance relative to what? Same applies to data shown in Supplementary Tables 2A and 2B

We have calculated the mRNA enrichment in biofilms relative to planktonic cells. This has been addressed in the corresponding Supplementary Tables 1, 2A and 2B with new text highlighted in blue.

“Supplementary Table 2A: mRNAs highly detected in extracellular 5-day biofilm matrix compared to chromosomal extract from 16 h overnight culture (i.e. planktonic) from P. aeruginosa cells in terms of higher log 2-fold change with a minimum base mean threshold of 30.”

20. Material and methods - The authors indicate that biofilms were grown in 2L flasks under static conditions. I assume this means that the biofilms formed as pellicles at the air-liquid interface. If this is the case, it is unclear why “...Biofilms were concentrated by centrifugation at 10,000 x g for 15 min” Equally puzzling is that the entire content of the flask was centrifuged, with “...Centrifugation stratified the biofilm mixture into three layers. The bottom two layers were collected and lyophilized”

Our model wildtype *P. aeruginosa* biofilm forms a viscoelastic gel-like biofilm throughout the growth medium rather than air-liquid pellicles. Therefore, centrifugation is performed to enable biofilm recovery from the growth medium. Following centrifugation, the biofilm stratifies according to density, where the more-dense part is darker and the less dense part transparent. Both are collected and dried as this is the starting material for nucleic acid gel extraction using ionic liquid, and the top layer containing water is discarded. This is described in more detail in our earlier publication presenting the extraction and recovery method ⁶. In the current manuscript, we describe this as follows:

Commencing line 518:

“All biofilm growth experiments were performed in three biological replicates. For extracellular nucleic acid gel isolation, NMR analysis and RNA sequencing ⁶, viscoelastic biofilms, shown previously to contain eDNA as key networking agents, were concentrated by centrifugation at 10,000 x g for 15 min in 50 ml Falcon tubes. Biofilm pellets were collected and lyophilised. For microscopy and rheology assays, the biofilms were collected directly from the Erlenmeyer flasks by gently aspirating the viscoelastic biofilm edge using a cut-off 1 ml pipette tip to minimise disturbance caused to the biofilms and specifically to preserve the native structure of biofilm matrix ⁶.”

21. Please clarify what kind of biofilms were used in this study, why biofilms were harvested in such a manner, and most importantly, what were the bottom layers used for?

Pseudomonas biofilms grown under static conditions were used in this study, as described in our publication presenting the extraction and recovery method ⁶. Our current study addresses how the assembly of eDNA provides biofilm viscoelasticity. This model is therefore used because it produces a lot of eDNA that we can extract and do more detailed chemical analyses on, but also because we can demonstrate by means of DNaseI digestion that eDNA is the critical viscoelastic material.

Commencing line 518:

“For extracellular nucleic acid gel isolation, NMR analysis and RNA sequencing, viscoelastic biofilms, shown previously to contain eDNA as key networking agents⁶, were concentrated by centrifugation at 10,000 x g for 15 min in 50 ml Falcon tubes.”

The detailed explanation of biofilm growth is described in lines 518.

“All biofilm growth experiments were performed with three biological replicates. For extracellular nucleic acid gel isolation, NMR analysis and RNA sequencing, viscoelastic biofilms, shown previously to contain eDNA as key networking agents⁹, were concentrated by centrifugation at 10,000 x g for 15 min in 50 ml Falcon tubes. Biofilm pellets were collected and lyophilised. For microscopy and rheology assays, the biofilms were collected directly from the Erlenmeyer flasks by gently aspirating the viscoelastic biofilm edge using a cut-off 1 ml pipette tip to minimise disturbance caused to the biofilms and specifically to preserve the native structure of biofilm matrix.”

Biofilm pellets resulting from centrifugation were used for extracellular nucleic acid extraction, NMR analyses and RNA sequencing.

22. Material and methods - One would assume when analyzing the biofilm matrix by microscopy, that the goal is to have the matrix in an intact state as possible rather than all compressed and out of context, as in the case of centrifugation. Please clarify

As described for Reviewer 3, biofilm centrifugation or lyophilisation only preceded matrix extraction (i.e., for NMR analyses and sequencing) and not smiFISH or any microscopy sample preparation. For microscopy-based analyses, the native biofilm was directly collected from flasks using cut-off 1 ml pipette, transferred to 2 ml Eppendorf tubes, processed further and finally deposited onto the poly-L-lysine slide for imaging. This protocol minimised the disturbance caused to the biofilms and preserve the native structure of biofilm matrix.

This has been clarified, firstly by describing the purpose of centrifugation and lyophilisation, as per the following:

Commencing line 521:

“Biofilm pellets were collected and lyophilised for further extracellular nucleic acid extraction, NMR analysis and RNA sequencing⁶.”

Furthermore, we have reorganised the Material and Methods to describe sample preparation more clearly for microscopy, and the tests dependent on this, as per the following text:

New section, commencing line 636:

“Staining

“eDNA staining was achieved following incubation of biofilm coated glass slides with 2 µM TOTO-1 iodide (1 mM solution in DMSO, Thermofisher Scientific, Catalog No: T3600) for 15 min in the dark. Live-dead imaging was performed according to LIVE/DEAD™ BacLight™ Bacterial Viability Kit, for microscopy (Catalog number: L7012, Thermofisher Scientific). Briefly, SYTO 9 dye, 3.34 mM (component A), 300 µl solution in DMSO and propidium iodide, 20 mM (component B), 300 µl solution in DMSO were mixed in equal volume. Three millilitres of the mix were added to every 1 ml of bacterial cell suspension and incubated in dark at room temperature for 15 min. Five millilitres of stained suspension were deposited onto the slide and imaged using LSM780 confocal microscopy (Zeiss).”

For smiFISH microscopy methodology:

Commencing line 694:

“Five-day-old native *P. aeruginosa* biofilms or clinical sputum sample were collected by gently aspirating them with shortened 1 ml pipette tip, transferring them to 2 ml Eppendorf tubes and washing once with double distilled water. The biofilms were not fixed nor permeabilised to avoid intracellular contamination. The primary probe and secondary FLAP were prehybridised using conditions described in Supplementary Tables 6 and 7. The washed biofilms were incubated in 40% formamide freshly prepared in 1X SSC buffer for 15 min at 22°C. During incubation, Mix 1 and Mix 2 were prepared on ice, as detailed in Supplementary Table 8. Mix 2 was vortexed for 30 s. Mix 1 was added to Mix 2 and vortexed again for 30 s. One hundred microlitres of the hybridisation mix were added to the deposited biofilms and covered with a 10 cm Petri dish. The Petri dish was wrapped in a Parafilm sheet and incubated at 37°C overnight⁵. After hybridisation, biofilms were deposited onto the poly-L-Lysine coated microscopic slide using gentle pipetting and covered with a coverslip.”

23. Material and methods - RNA extraction. How were biofilms grown for subsequent RNA extraction?

We performed subsequent RNA extractions on biofilms grown under model static growth conditions, on days 1, 2, 3 and 5 of the growth experiment, in biological triplicate (i.e., in total 12 for intracellular and 12 for extracellular RNA extraction).

We have clarified this with following text:

Commencing line 605:

“For the subsequent RNA extraction experiment, biofilms were grown under model static conditions, on days 1, 2, 3 and 5 of the growth experiment, in biological triplicate for both intracellular and extracellular RNA extraction. Hence, six samples (three for intracellular and three for extracellular RNA extraction) were collected every day.”

References (references newly included in manuscript highlighted in bold):

1. Wüthrich K. NMR with proteins and nucleic acids. *Europhys News* **17**, 11-13 (1986).
2. Chiba A, Seki M, Suzuki Y, Kinjo Y, Mizunoe Y, Sugimoto S. ***Staphylococcus aureus* utilizes environmental RNA as a building material in specific polysaccharide-dependent biofilms. *npj Biofilms Microbiomes* 8, 17 (2022).**
3. Domenech M, Pedrero-Vega E, Prieto A, García E. Evidence of the presence of nucleic acids and β -glucan in the matrix of non-typeable *Haemophilus influenzae in vitro* biofilms. *Sci Rep* **6**, 36424 (2016).
4. Scherbakova AE, Rykova VS, Danilova KV, Solovyev AI, Egorova DA. Extracellular RNA isolation from biofilm matrix of *Pseudomonas aeruginosa*. *Bio-protocol* **10**, e3810 (2020).

5. Tsanov N, *et al.* smiFISH and FISH-quant - a flexible single RNA detection approach with super-resolution capability. *Nucleic Acids Res* **44**, e165-e165 (2016).
6. Seviour T, *et al.* The biofilm matrix scaffold of *Pseudomonas aeruginosa* contains G-quadruplex extracellular DNA structures. *npj Biofilms Microbiomes* **7**, 27 (2021).
7. Schwechheimer C, Kuehn MJ. Outer-membrane vesicles from Gram-negative bacteria: Biogenesis and functions. *Nat Rev Microbiol* **13**, 605-619 (2015).
- 8. Kaparakis-Liaskos M, Ferrero RL. Immune modulation by bacterial outer membrane vesicles. *Nat Rev Immunol* **15**, 375-387 (2015).**
9. Whitchurch CB, Tolker-Nielsen T, Ragas PC, Mattick JS. Extracellular DNA required for bacterial biofilm formation. *Science* **295**, 1487 (2002).

Reviewer #1 (Remarks to the Author):

This reviewer would like to thank the authors for addressing previous concerns and questions raised by the reviewers. After carefully reading all their responses (also to the other reviewers) and re-evaluating their data, the reviewer has even more serious concerns about the presented data as mentioned below. As there are many errors, the authors should carefully check the text and figures.

Major concern:

1. The main issue with this study is the lack of biological significance to compare RNA between in the NA gels isolated from the 5-day biofilm and in 16-h planktonic cells (Figure 2A, log₂-fold change in Supplementary Table 2A and 2B). From these data, the authors conclude that some specific RNA species (e.g., *lasB*) were enriched in the NA gels, but it is difficult to conclude this from the results. There was no or less enrichment of *lasB* mRNA in the extracellular matrix of the 5-day biofilm compared with intracellular of the 5-day biofilm (Fig. 2B).

2. Supplementary 8B: The authors changed the labels of the upper (Intracellular) and lower (Extracellular) panels from Supplementary Figure 7B of the original manuscript with no explanation. Importantly, assignment of peaks seems to be incorrect and mismatched with Supplementary Tables 2A and 2B. The X-axis represents gene names aligned in ascending order (from left to right). Nevertheless, for example, the authors assigned a peak with 30,000-40,000 base mean coverage at the position around PA4428 in the X-axis as *lasB* mRNA (PA3724) in the upper panel. However, the base mean coverage of *lasB* was only 2,685 as shown in Supplementary Table 2A. Other peaks may have also been assigned incorrectly. The Y-axis represents base mean coverage, but these values are inconsistent with those in Supplementary Table 2A and 2B.

3. Many data of smiFISH are still not convincing. The authors selected some of mRNA with low abundance identified in extracellular NA gel by RNA-seq as targets for smiRNA (*lasB*, *antB*, *bkdA2*, and *idpV*). The authors concluded that these mRNA were enriched in the isolated extracellular NA gel, but this interpretation is improper as mentioned in the current comment #1. Supplementary Tables 2A and 2B show that base mean coverage of these mRNA are much lower than other abundant RNA (e.g., *ssrA*, *rnpB*, 23S rRNA, *crcZ*), indicating that abundance of *lasB*, *antB*, *bkdA2*, and *idpV* are very low. Nevertheless, these low abundant mRNA could be detected on extracellular NA fibers by smiFISH, whereas abundant RNA (*ssrA*, *crcZ*, 23S rRNA) could not (Figure 4C and Supplementary Figure 12B). The authors explain that the presence of intracellular RNA in the biofilm matrix samples could result from cell death, but it is unusual that only some specific intracellular RNA contaminated into the biofilm matrix samples. One of the most reasonable explanations could be due to some problems in smiFISH experiments. Can the authors detect abundant RNA (*ssrA*, *crcZ*, 23S rRNA) colocalized with the isolated NA fibers by smiFISH? The reviewer also wonder why the authors removed rRNA (16S rRNA, not 23S rRNA?, tRNA?) in RNA-seq analysis. These ribosomal RNA should not be removed in the data analysis as they may exist in the isolated NA gel samples. In addition, both 16S rRNA and 23S rRNA were not detected on extracellular fibers by smiFISH using multiple probes, but why these probes did not stain bacterial cells (Supplementary Figures 12A and 12 B)? According to the results of FISH (Supplementary Figures 12A and 12B) and smiFISH for the other RNA (Figures 3 and 4) it is reasonable that the multiple probes specific for 16S rRNA and 23S rRNA can also stain some bacterial cells.

4. Supplementary Figure 8A seems to be inappropriately processed by the authors. The image of DNA ladder (lane 1) was copied and pasted from the outside (probably, the website of Fisher Scientific?). A similar copy/paste is found in Supplementary Figure 14B. The middle image (lane 2 to 5) is originated from Supplementary Figure 7A, extended vertically, and connected to the right image that may have been newly acquired (lane 6 to 8). This is bad practice.

5. There are many microscopic images without constant fluorescent intensity. For example, some data show clear extracellular fibers stained with TOTO-1 in control samples (Figures 3A, 4A, and Supplementary Figures 9, 11A, 11C, 11D), while others represent faint or no fibers even in the similar samples (Figures 3H, 4C, and Supplementary Figure 11B). In addition, the contrast in many images (Figures 1C, 3D, 3E, 5A, 6A, Supplementary Figures 2A-C, 5A, 10, 11, 15C, 15D) seems to be inappropriately high, which makes the reviewer wonder how the lookup tables were

chosen for all the images and if they allow for a fair comparison between the different treatments and samples. Maybe it would make sense to show the images with a lookup table such that the imaging noise/background can be seen as a reference?

6. The authors show that microscopic images of the TOTO-1-stained biofilms formed by Δ psf and Δ pelA (Supplementary Figures 2A and 2B). As mentioned above, the contrast adjustment in these images seems to be inappropriately high, which makes a fair comparison difficult. Again, it would make sense to show the images with a lookup table such that the imaging noise/background can be seen as a reference. If the contrast is reduced similar to a level of Figure 3A and 4A, eDNA fibers would become invisible or much less. In addition, to eliminate a possibility that Psl and Pel play a redundant role in formation of TOTO-1 positive fibers, a double knockout strain should be used. Furthermore, the authors show a wild-type biofilm treated with proteinase K (Supplementary Figure 2C), but the contrast of this image is also too high, and less fibers are shown compared with the control figures as mentioned above.

Importantly, proteinase K is not able to degrade amyloid fibers in general. Alternatively, Δ fapC mutant can be used to investigate an importance of extracellular amyloids in formation of extracellular fibers. In Supplementary Figure 2E, extracellular NA gel from wild type treated with proteinase K looks very different from those of Δ psf and Δ pelA. But, no picture of extracellular NA gel isolated from wild type without enzyme treatment is shown here. When compared with NA gel isolated from wild type without enzyme treatment, do proteinase K and deletion of psf and pelA affect characteristics of NA gel? These structures can be compared more in detail by CLSM with TOTO-1 staining.

7. Thank you for providing new data of the NA gel isolate analyzed by CSLM (Supplementary Figure 2D) and agarose gel electrophoresis (Supplementary Figure 14B). On the one hand, it is nicely shown that there are many TOTO-1 positive fibers in this sample (Supplementary Figure 2D). On the other hand, the statement in the text (lines 388-390) is incorrect as there is a remarkable (but not slight) reduction in DNA length. An average size of eDNA isolate was likely more than 10,000 base, and it seemed to be reduced to less than around 2,000 bp in average (not 8,000 bp) after alkali treatment (Supplementary Figure 14B). This result suggests that fragmentation of eDNA in the isolated NA gel may occur after alkali treatment, which can be addressed by CLSM with TOTO-1 staining as shown in Supplementary Figure 2D.

8. Use of RNase H is a good idea to examine the role of eRNA bound to eDNA in integrity of extracellular NA fibers, but the presented data are not convincing enough to the reviewer. Although accuracy of data presented in Supplementary Figure 8A may be doubtful as mentioned above (see current comment # 2), RNase A and RNase H reduced the size of extracellular nucleic acids to a similar extent, and their combination with DNase I did so more remarkably (Supplementary Figure 8A). In addition, 0.4 mg/ml DNase I alone degraded eDNA for 30 min at 37°C. These results make the reviewer wonder how the same enzymatic treatments affect extracellular fibers in the biofilm stained with TOTO-1. In Figures 5A and 5B, only extracellular fibers treated with 0.1 mg/ml DNase I and 0.1 mg/ml DNase I + uncertain concentration of RNase H are shown. In addition, Figure 2 in the response letter shows combination of uncertain concentrations of DNase I and RNase A degraded extracellular fibers in the 5-day biofilm. To compare effects of enzymes on fragmentation and fiber structure of extracellular nucleic acids in the biofilm, it is better to show CLSM images of biofilm without enzyme treatment (control) as well as ones treated with 0.4 mg/ml DNase I alone, 0.3 mg/ml RNase A alone, 0.3 mg/ml RNase H alone, 0.4 mg/ml DNase I + 0.3 mg/ml RNase A, 0.4 mg/ml DNase I + 0.3 mg/ml RNase H in the same figure. Of course, it is good to show the images with lookup tables for fair comparison.

9. The authors did not fully address the concern on the role of eRNA in formation of the biofilm and extracellular NA fibers, which was pointed out by this reviewer in the previous comment #3. Do 0.3 mg/ml RNase A and 0.3 mg/ml RNase H inhibit formation of biofilm and extracellular NA fibers when they are added to the culture media at the onset of biofilm formation?

Minor comments

1. Supplementary Figure 14B: The authors may have done copying/pasting the DNA ladder maker from outside (some web site of company?). As mentioned above, this is very bad practice. If DNA makers are indistinguishable at high molecular sizes, the authors have to perform the same

experiment again using low concentration of agarose.

2. Lines 181-191 are not correct: (i) The size of nucleic acids can not be estimated as the marker is illegally connected to the image. (ii) There is less evidence that RNase A treatment of the NA gel did not disrupt the network structure. Supplementary Figure 10B shows effects on extracellular fibers in the biofilm treated with RNase A but lacks fair comparison with untreated samples as mentioned above. (iii) There is a reduction in molecular size of nucleic acids after RNase A treatment (Supplementary Figure 8A, lane 4). (iv) The extracellular nucleic acids in the gel isolate from *P. aeruginosa* biofilms were sensitive to all enzymes to different extents and highly susceptible to combined enzymes compared with single enzymes (Supplementary Figure 8A). (v) These results do not suggest that RNA was protected from enzymatic hydrolysis.

3. Thank you for providing a result showing degradation of gamma globulin (Figure 1 in the response letter). However, the band intensity of a protein larger than 170 kDa seems to be slightly reduced in the lane 3. In addition, no band corresponding to DNase I is visible. How much concentration of gamma globulin was treated with 0.4 mg/ml DNase I and how long? The catalog numbers of the other enzymes are indicated in the methods section but that of DNase I is missing. Furthermore, gamma globulin is a tightly folded protein and thus some how resistant to proteases. Unfolded proteins (e.g., casein) is generally used as a substrate to examine activities of proteases in biochemical studies.

4. It is strange that not only vertical error bars but also horizontal error bars are shown (Figure 2D, day 1 white square)

5. Supplementary Figure 15C: Obviously, this is also minor, but some lasB-positive signals are visible in the region close to the scale bar.

Original Supplementary Figure 8A

A

Brightness-changed Supplementary Figure 8A

From Supplementary Figure 7A
of the initial submission

From the web site of Fisher Scientific

https://www.google.co.jp/search?q=1+kb+ladder+in+vitrogen&tbm=isch&chips=q:1+kb+ladder+in+vitrogen,online_chips:thermo+scientific+generuler:fnfFF92VEeU%3D&hl=ja&sa=X&ved=2ahUKEwjB-aOmp63-AhXyt1YBHxmBjjQ4IYoAnoECAEQMA&biw=1010&bih=448#imgrc=Y3xh6HicyzawPM

Brightness-changed Supplementary Figure 8A

At least two images are connected here

A

Original Supplementary Figure 8B in the revised manuscript

Original Supplementary Figure 7B in the initial manuscript

B

Reviewer #3 (Remarks to the Author):

Authors have successfully addressed my previous comments.

Reviewer #4 (Remarks to the Author):

The revised manuscript by Mugunthan et al has much improved with the additional (clarifying) experiments and the more detailed explanations.

However, I still have a few concerns

1. The majority of the data appears to be shown in the supplement but there are several results that should be shown in the main manuscript. For example, RNAseH only treated biofilms should be shown in Figure 5. I strongly encourage the authors to carefully review their data and move pertinent information including controls to the main manuscript
2. None of the data including the next gen sequencing data include statistics. Please include throughout. Also, please indicate statistical methods in the Methods section
3. Data only show averages but should really show individual data points as well
4. Figure 3
 - Fig. 3E, indicate how biofilms were treated prior to image acquisition
 - Fig. 3G, Mander's coefficient is only based on 4 images. Are these images from biological or technical replicates, please specify. If the images are from technical replicates, additional biological replicates will be required
 - Fig. 3H-I, same as for Figure 3G; also, what is the meaning of (A) in legend for Fig 3I?
5. Inconsistency, e.g. 5 days vs 5 d
6. Awkward wording or excess of commas/periods, see l. 305, l. 369, l. 488-489
7. Section on lasB and contaminating eRNAs is not very clear/confusing, with the section on contaminating eRNA and the role of lasB (lasB mutant data) not being very supportive of the major findings reported here. It actually brings the overall findings into question. Suggest revising
8. suggest moving section l. 387-401 after section l. 419-445

Reviewer #1 (Remarks to the Author):

This reviewer would like to thank the authors for addressing previous concerns and questions raised by the reviewers. After carefully reading all their responses (also to the other reviewers) and re-evaluating their data, the reviewer has even more serious concerns about the presented data as mentioned below. As there are many errors, the authors should carefully check the text and figures.

Major concern:

1. The main issue with this study is the lack of biological significance to compare RNA between in the NA gels isolated from the 5-day biofilm and in 16-h planktonic cells (Figure 2A, log₂-fold change in Supplementary Table 2A and 2B). From these data, the authors conclude that some specific RNA species (e.g., *lasB*) were enriched in the NA gels, but it is difficult to conclude this from the results. There was no or less enrichment of *lasB* mRNA in the extracellular matrix of the 5-day biofilm compared with intracellular of the 5-day biofilm (Fig. 2B).

To identify whether the viscoelastic behaviour of nucleic acids in *P. aeruginosa* biofilms could be explained by a distinct RNA expression profile, RNA expression levels were compared for matrix and planktonic *P. aeruginosa* cells. Expression levels in the matrix embedded (biofilm) cells were considered stable over the 5-day growth period based on their consistent mechanical stability and viscoelasticity (Supplementary Figure 6C). This is included in the manuscript with following text

Line 156:

“Furthermore, the biofilm displayed viscoelastic properties from days 1-5 of growth (Supplementary Figure 6C), with the number of dead cells increasing from 8 to 68% from days 2 to 5, suggesting that the viscoelastic behaviour was independent of the number of dead cells (Supplementary Figure 7A-E).”

The biofilm cells were compared to planktonic cells at a stage preceding the onset of EPS production in planktonic cells. *P. aeruginosa* planktonic cells begin to produce EPS and aggregate at an early stage of growth (i.e. approximately 1 d), even with shaking. Hence, although not a direct biological comparison, assessing established biofilm cells with 16 h planktonic cells provided the means to ascertain and compare the RNA produced in the different phases, i.e., planktonic vs biofilm, without the interference of aggregate formation in the plankton.

In our first revision, we provided the expression levels of the eRNA in biofilms over time, showing that the amount of eRNA was greater in the biofilm matrix on day 5 than that associated with their planktonic cell counterparts. This was also true for days 1-3. On the other hand, iRNA levels were less in the biofilm matrix throughout than in the planktonic cells. The relative proportions remain consistent, such that greater amounts of eRNA in the biofilm matrix compared to planktonic cells were evident despite the day of growth on which the sample was taken. This is clarified with the following:

Line 194:

“To identify whether the viscoelastic behaviour of nucleic acids in P. aeruginosa biofilms could be explained by a distinct RNA expression profile, RNA expression levels were compared between the matrix from a five-day P. aeruginosa biofilm with planktonic P. aeruginosa. Assessing established biofilm cells with 16 h planktonic cells (i.e., cells sampled prior to their onset of EPS production) provided the means to ascertain

the level of RNA produced in the different phases, that is, biofilm versus planktonic, without the interference of aggregate formation. Expression levels in the matrix were considered stable, based on their consistent mechanical stability and viscoelasticity (Supplementary Figure 6C). Evaluating eRNA as described above revealed several mRNA transcripts that were highly enriched in both planktonic cells and the biofilm matrix cells, as indicated by their high base mean average (Figure 2A, Supplementary Figure 8B, Supplementary Table 2B)."

Line 238:

"Nonetheless, while expression levels of putative iRNA and eRNA in the matrix varied from days 2 to 5, the relative proportions of iRNA and eRNA remained constant. Greater amounts of eRNA were detected in the biofilm matrix compared to those in association with planktonic cells, despite the age of the biofilm. Further, the iRNA was also consistently lower in proportion in the biofilm matrix than the planktonic cells. Thus, this approach for assessing iRNA and eRNA associated with microbial cells is valid."

The reviewer correctly notes that the discrepancy between intracellular and extracellular RNA content narrows towards day 5. This is most likely due to cell death and lysis, where the distinction between intracellular and extracellular is less pronounced due to increased solubility of intracellular RNA. We now provide data for dead cells over time to demonstrate this, as discussed with the following text:

Line 230:

*"gene counts standardised against total reads per sample were higher in the biofilm matrix than in biofilm cells and planktonic cells throughout the biofilm growth assay. These extracellular mRNA are therefore enriched in the biofilm matrix compared to its abundance in both planktonic and biofilm cells. *ssrA* and *crcZ* transcripts, on the other hand, were higher in the biofilm cells than in the biofilm matrix, yet present in similar abundances in the biofilm cells and planktonic cells (Figure 2F, G). The difference between eRNA levels in the biofilm matrix and its cells became increasingly less pronounced from days 2 to 5. This correlates with increased cell death and lysis with time (Supplementary Figure 7) and is likely a consequence of increased iRNA solubility in the matrix."*

Finally, and crucially, these methods for differentiating eRNA and iRNA are validated microscopically, showing that those RNA identified as eRNA are seen in extracellular fibres and those as iRNA in cells. This reduction in the level of *lasB* in the matrix referred to here for day 5 also coincides with a reduction in *lasB* signal intensity using smiFISH, which we highlight with the following text:

Line 363:

*"This reduction in *lasB* signal intensity from days 4 to 5 was also observed in the mRNA count for the biofilm (Figure 2B)."*

2. Supplementary 8B: The authors changed the labels of the upper (Intracellular) and lower (Extracellular) panels from Supplementary Figure 7B of the original manuscript with no explanation. Importantly, assignment of peaks seems to be incorrect and mismatched with Supplementary Tables 2A and 2B. The X-axis represents gene names aligned in ascending order (from left to right). Nevertheless, for example, the authors assigned a peak with 30,000-40,000 base mean coverage at the position around PA4428 in the X-axis as *lasB* mRNA (PA3724) in the upper panel. However, the base mean coverage of *lasB* was only

2,685 as shown in Supplementary Table 2A. Other peaks may have also been assigned incorrectly. The Y-axis represents base mean coverage, but these values are inconsistent with those in Supplementary Table 2A and 2B.

We thank the reviewer for identifying this oversight that arose during the last revision with regards to labelling of the panels. We have amended this, and all lines now connect with the corresponding peaks.

We would like to clarify that the base mean numbers showed in Supplementary Figure 8B, indicate the raw copy number of each RNA in planktonic and biofilm matrix samples (i.e. intracellular and extracellular respectively). Supplementary Tables 2A and B, on the other hand, show the base mean average of RNA (i.e. copy number) across both sets of samples, which was used as the basis to assess relative expression levels. This explains the different base mean numbers showcased in Supplementary Tables 2A and B compared to Supplementary Figure 8B. Nonetheless, to avoid confusion we have now removed the base mean average and log2fold change numbers from Supplementary Figure 8B.

3. Many data of smiFISH are still not convincing. The authors selected some of mRNA with low abundance identified in extracellular NA gel by RNA-seq as targets for smiRNA (*lasB*, *antB*, *bkdA2*, and *idpV*). The authors concluded that these mRNA were enriched in the isolated extracellular NA gel, but this interpretation is improper as mentioned in the current comment #1. Supplementary Tables 2A and 2B show that base mean coverage of these mRNA are much lower than other abundant RNA (e.g., *ssrA*, *rnpB*, 23S rRNA, *crcZ*), indicating that abundance of *lasB*, *antB*, *bkdA2*, and *idpV* are very low. Nevertheless, these low abundant mRNA could be detected on extracellular NA fibres by smiFISH, whereas abundant RNA (*ssrA*, *crcZ*, 23S rRNA) could not (Figure 4C and Supplementary Figure 12B). The authors explain that the presence of intracellular RNA in the biofilm matrix samples could result from cell death, but it is unusual that only some specific intracellular RNA contaminated into the biofilm matrix samples. One of the most reasonable explanations could be due to some problems in smiFISH experiments.

Just to clarify, as would be expected, many intracellular RNA were also detected in the matrix. We focused only on those iRNA with greatest base mean coverage, including *ssrA* which is typically detected in high abundance in the stressful conditions found in biofilms, as stated in the following text/reference:

Line 204:

“these include ssrA, a transfer messenger RNA involved in the trans-translation process for protein stability under stress, which is often detected in high abundance in biofilm transcriptomics studies¹ ”.

smiFISH is a published method. Our findings, methods, controls etc were conducted as described by Tsanov et al., NAR, 2016, and approved by Reviewer 2 (identified as the smFISH expert). Regarding the identification of a subset of iRNA, we investigated only those with the greatest log-fold increase in the matrix relative to planktonic cells. Significantly, our method for assigning eRNA and iRNA was validated by smiFISH microscopy, which provides the best available validation for our assignment of iRNA and eRNA.

Nonetheless, we accept that the focus on *lasB* could be misconstrued as a statement that it is a fundamental component of the eDNA fibers, which the time-series and *lasB* knockout biofilm growth assays show to not be the case. Our observations of eRNA in the eDNA fibers in model and clinical biofilms, as illustrated most clearly by *lasB*, using validated microscopic methods (smiFISH), is, to our knowledge, the first time any defined nucleic acid sequence has been detected in the extracellular matrix of any biofilm. It represents a significant advance from non-specific eDNA staining methods. Furthermore, it

validates our method for determining iRNA and eRNA, provides direct evidence of the presence of eRNA in eDNA fibers, and also shows that *lasB* is a suitable biomarker for detecting the presence of eRNA in model and clinical *P. aeruginosa* biofilms. This is reflected with the following text:

Line 531:

“While lasB appears not to be essential for P. aeruginosa biofilms, it is nonetheless a suitable biomarker for observing eRNA in viscoelastic eDNA networks of clinical and model P. aeruginosa biofilms.”

Can the authors detect abundant RNA (*ssrA*, *crcZ*, 23S rRNA) colocalized with the isolated NA fibres by smiFISH?

We performed smiFISH on isolated NA fibres targeting *ssrA*, *crcZ* and 23s rRNA as per the reviewer’s suggestion. The experiment showed an absence of *ssrA*, *crcZ* and 23s rRNA signal across the NA fibres (included below is the relevant figure, for the reviewer’s perusal only).

Rejoinder Figure A: smiFISH confocal micrograph of an extracellular nucleic acid (NA) gel isolate from five-day old *P. aeruginosa* biofilms stained with transfer messenger RNA *ssrA* and small RNA *crcZ* specific oligoribonucleotide (red). eDNA fibres are stained using TOTO-1 dye (green). Scale bars represent 10 μ m.

The reviewer also wonder why the authors removed rRNA (16S rRNA, not 23S rRNA?, tRNA?) in RNA-seq analysis. These ribosomal RNA should not be removed in the data analysis as they may exist in the isolated NA gel samples.

smFISH microscopy targeting 16S and 23S rRNA showed that ribosomal RNA were absent in the NA gel samples as shown in response to next question below.

In addition, both 16S rRNA and 23S rRNA were not detected on extracellular fibres by smiFISH using multiple probes, but why these probes did not strain bacterial cells (Supplementary Figures 12A and 12 B)?

We performed smiFISH on NA fibres in the purified NA gel extract targeting 16S and 23S ribosomal RNA as per reviewer suggestion. This showed the absence of rRNA signal along the NA fibres in the purified NA extract, further suggesting that the ribosomal RNA was not enriched in the matrix (included in the figure below for the reviewer's perusal only).

smiFISH uses a lower hybridisation temperature than conventional FISH (37 cf 48°C) thereby explaining the lack of staining of bacterial cells (expanded upon in response to the next point).

Rejoinder Figure B: smiFISH confocal micrograph of extracellular nucleic acid (NA) gel isolate from five-day old *P. aeruginosa* biofilms stained with 16S ribosomal RNA and 23S ribosomal RNA specific oligoribonucleotide (red). eDNA fibres are stained using TOTO-1 dye (green). Scale bars represent 10 μ m.

According to the results of FISH (Supplementary Figures 12A and 12B) and smiFISH for the other RNA (Figures 3 and 4) it is reasonable that the multiple probes specific for 16S rRNA and 23S rRNA can also stain some bacterial cells.

We agree with the reviewer that the cells are not stained by the 16S and 23S rRNA probes. This is likely because a much lower hybridization temperature for smiFISH of 37°C (reference) is used as compared to 46°C for conventional FISH, where the higher temperature is required to denature dsDNA and enable access of the probe to the single stranded form^{2, 3, 4}.

This is clarified with the following text:

Line 348:

“The lower hybridisation temperature compared to conventional FISH precludes staining of genomic DNA by 16S and 23S probes following smiFISH⁴.”

4. Supplementary Figure 8A seems to be inappropriately processed by the authors. The image of DNA ladder (lane 1) was copied and pasted from the outside (probably, the website of Fisher Scientific?). A similar copy/paste is found in Supplementary Figure 14B. The middle image (lane 2 to 5) is originated from

Supplementary Figure 7A, extended vertically, and connected to the right image that may have been newly acquired (lane 6 to 8). This is bad practice.

As per the reviewer's suggestion, both figures are now presented along with nucleic acid ladder on the same gel (new Supplementary Figures 8A (i), (ii) and 14B).

5. There are many microscopic images without constant fluorescent intensity. For example, some data show clear extracellular fibres stained with TOTO-1 in control samples (Figures 3A, 4A, and Supplementary Figures 9, 11A, 11C, 11D), while others represent faint or no fibres even in the similar samples (Figures 3H, 4C, and Supplementary Figure 11B). In addition, the contrast in many images (Figures 1C, 3D, 3E, 5A, 6A, Supplementary Figures 2A-C, 5A, 10, 11, 15C, 15D) seems to be inappropriately high, which makes the reviewer wonder how the lookup tables were chosen for all the images and if they allow for a fair comparison between the different treatments and samples. Maybe it would make sense to show the images with a lookup table such that the imaging noise/background can be seen as a reference?

some data show clear extracellular fibres stained with TOTO-1 in control samples (Figures 3A, 4A, and Supplementary Figures 9, 11A, 11C, 11D), while others represent faint or no fibres even in the similar samples (Figures 3H, 4C, and Supplementary Figure 11B).

the contrast in many images (Figures 1C, 3D, 3E, 5A, 6A, Supplementary Figures 2A-C, 5A, 10, 11, 15C, 15D) seems to be inappropriately high.

The objective of the study is to account for the ability of the eDNA to form fibre networks. Hence, it is appropriate to relate the presence and coverage of RNA to the eDNA fibres. The signal from dead cells can be much higher than the fibres, and hence contrast adjustments are based on the eDNA fibre signal rather than the dead cells, in order to visualise the fibres and assess which RNA align with them. We have explained the application of contrasting with the following text:

Line 723:

“For all images assessing eRNA colocalisation with eDNA fibres, contrast adjustment was performed relative to the fibre signal rather than the signal from dead cells. “

6. The authors show that microscopic images of the TOTO-1-stained biofilms formed by Δ psIF and Δ pelA (Supplementary Figures 2A and 2B). As mentioned above, the contrast adjustment in these images seems to be inappropriately high, which makes a fair comparison difficult. Again, it would make sense to show the images with a lookup table such that the imaging noise/background can be seen as a reference. If the contrast is reduced similar to a level of Figure 3A and 4A, eDNA fibres would become invisible or much less.

Similar to our response to point 5 above, the contrast in Supplementary Figures 2A and 2B was adjusted in order for the fibres to be visible, due to the presence of a higher number of dead cells stained with TOTO-1 in S2A and 2B. Our intention was not to compare the amount of eDNA fibres in the wildtype (Fig.3) and the mutants (Supplementary Figure 2), but to confirm that the eDNA fibres are also visible in the

mutants. Therefore, adding a LUT, which again would be different for each figure would not provide information of relevance to this study.

In addition, to eliminate a possibility that Psl and Pel play a redundant role in formation of TOTO-1 positive fibres, a double knockout strain should be used.

The absence of polysaccharide peaks in the solution and solid-state NMR spectra provides evidence that polysaccharides were not present in the gel isolate, negating the need for double-knockout strain analysis. This research focuses on eRNA as a key component of viscoelastic eDNA networks in *Pseudomonas*, and the stated objective is to resolve the nucleic acid composition of the *Pseudomonas* biofilm eDNA. Firstly, we sought to understand how eDNA assembles into gels, which led to the observation that eRNA was also present. Secondly, following this observation, we then determined the nucleic acid composition of the biofilm and confirmed eRNA in the matrix. This addressed our stated objective of explaining eDNA viscoelasticity in terms of its nucleic acid content. We respectfully contend that incorporating the potential role of matrix polysaccharides is beyond the scope of our study. Nonetheless, we demonstrate that fibres form in the absence of extracellular polysaccharides, Pel or Psl, and we provide additional information demonstrating that fibres form without amyloid FapC, as described below in response to the Reviewer's next comment.

With regards to the Pel Psl double knockout mutant, we have previously described the lack of biofilm formation (and hence eDNA fibres) in this strain. This observation generated significant interest for the role of Pel and Psl in *P. aeruginosa* biofilms⁵. The use of genetic knockouts is clearly a key tool in molecular biology and biofilm studies. However, in this case, we did not perform a polysaccharide addback experiment, owing to the lack of isolated Pel and Psl. Indeed, there is currently no methodology available for such phenotypic experimentation using complex polysaccharides, especially with regards to their integration into the intricate EPS and developing such protocols would be a substantial study in itself. This is currently a drawback to assigning EPS functions in biofilms for these molecules. The lack of biofilm formation in the double knockout therefore does not provide explicit evidence that these molecules directly contribute to viscoelastic structures. Thus, to our knowledge, the only way to definitely assign function is to employ the analytically complex method of extracting, isolating, and performing physicochemical analyses of isolated materials, as described in our study.

Furthermore, the authors show a wild-type biofilm treated with proteinase K (Supplementary Figure 2C), but the contrast of this image is also too high, and less fibres are shown compared with the control figures as mentioned above.

Importantly, proteinase K is not able to degrade amyloid fibres in general. Alternatively, Δ fapC mutant can be used to investigate an importance of extracellular amyloids in formation of extracellular fibres.

We include here for the reviewer's perusal only a TOTO-1 stained image of a Δ fapC mutant biofilm, showing clearly the presence of a network eDNA fibres in the amyloid mutant, indicating that amyloid fibres are not required for eDNA fibre formation in *P. aeruginosa* biofilms.

Δ fapC mutant

Rejoinder Figure C: Confocal micrograph of five-day old Δ fapC amyloid mutant of *P. aeruginosa* biofilms stained with eDNA specific TOTO-1 dye seen in green. Scale bars represent 10 μ m.

In Supplementary Figure 2E, extracellular NA gel from wild type treated with proteinase K looks very different from those of Δ pslF and Δ pelA. But, no picture of extracellular NA gel isolated from wild type without enzyme treatment is shown here. When compared with NA gel isolated from wild type without enzyme treatment, do proteinase K and deletion of pslF and pelA affect characteristics of NA gel? These structures can be compared more in detail by CLSM with TOTO-1 staining.

The image provided was one of an NA gel, but the difference in the image portrayed was due to light reflection from the cover of the well plate. A new image of an NA gel has now been included without the interference from the plate cover.

The image of the extracellular NA gel isolate without enzyme treatment was provided in our previous publication on this topic, and we now refer to this with the following text:

Line 124:

*“The viscoelastic NA gel isolate from *P. aeruginosa* contained eDNA fibres (Supplementary Figure 2D) and disappeared upon alkalinisation (Supplementary Figure 2E). Viscoelastic NA gels were isolated from the Δ pelA and Δ pslF polysaccharide mutants as well as the proteinase K treated PAO1 biofilm (Supplementary Figure 2F), as for NA from the untreated PAO1².“*

These results are included to demonstrate only that biofilm without Pel, Psl and proteins all include eDNA that can form gel networks. There are multiple factors during extraction, processing and imaging that likely have far more effect on the crosslink density of the gel than whether they were extracted from e.g. the *pslF* or *pelA* knockouts. Such analyses therefore are not pertinent to this study. We thank the reviewer for their suggestion and accept that this approach might be useful, for example in studying the optimisation of NA gel preparation, or the effect of *pslF* and *pelA* knockouts on NA content.

7. Thank you for providing new data of the NA gel isolate analyzed by CSLM (Supplementary Figure 2D) and agarose gel electrophoresis (Supplementary Figure 14B). On the one hand, it is nicely shown that there are many TOTO-1 positive fibres in this sample (Supplementary Figure 2D). On the other hand, the statement in the text (lines 388-390) is incorrect as there is a remarkable (but not slight) reduction in DNA length. An average size of eDNA isolate was likely more than 10,000 base, and it seemed to be reduced to less than around 2,000 bp in average (not 8,000 bp) after alkali treatment (Supplementary Figure 14B). This result suggests that fragmentation of eDNA in the isolated NA gel may occur after alkali treatment, which can be addressed by CLSM with TOTO-1 staining as shown in Supplementary Figure 2D.

We have amended the text as suggested by the reviewer to the following:

Line 460:

“There was a reduction in DNA length of approximately 8000 bp”

As per the reviewer’s suggestion we now provide a TOTO-1 stained image of alkalinised NA gel isolate showing the complete disappearance of eDNA fibres as result of alkalinisation (Supplementary Figure 2E).

This is clarified with following text.

Line 125:

“disappeared upon alkalinisation (Supplementary Figure 2E)”

Furthermore, we acknowledged that loss of DNA mass or secondary structure could also explain the disintegration upon alkalinisation, and thus provided additional results showing directly that RNA is a key structural component, by means of the RNase H assays. This text is included here again for the convenience of the reviewer:

Line 462:

“To assess directly whether eRNA contributes to P. aeruginosa biofilm viscoelasticity, 0.3 mg ml⁻¹ of RNase A digestion of the P. aeruginosa biofilm was coupled with mild pre-DNase I treatment (0.1 mg ml⁻¹), as per the smiFISH microscopy. This increased $\tan \delta$ for the biofilm relative to the untreated biofilm in the rheology frequency sweep and reduced eDNA fibre fluorescence intensity by 47% (Figure 6A-E); although $\tan \delta$ remained below 1. This is consistent with the observation that the viscoelastic network was preserved. However, when RNase A was replaced with 0.3 mg ml⁻¹ of RNase H, targeting the RNA strand of RNA:DNA hybrids, the model P. aeruginosa biofilm dissolved and lost its viscoelasticity, as indicated by the increase in $\tan \delta$ to > 1 and 73% reduction in eDNA fibre fluorescence intensity (Figure 6D and E). Without mild pre-DNase I treatment for 0.3 mg ml⁻¹ of RNase A and RNase H, there was a slight increase in $\tan \delta$, and 18 and 51% reductions in eDNA fibre fluorescent intensity respectively, demonstrating that their viscoelastic networks were preserved (Figure 6D and E). Mild DNase I treatment (0.1 mg ml⁻¹) with RNase H buffer alone additionally did not dissolve the biofilm and resulted in about less than 4% loss of eDNA fibre fluorescent intensity (Figure 6D and E).”

Line 522:

“eRNA is a key viscoelastic component of Pseudomonas biofilms. Biofilm alkalinisation, resulting in eRNA transesterification, therefore degrades a foundation structural material. This could account for Pseudomonas biofilm dissolution by alkanisation.”

8. Use of RNase H is a good idea to examine the role of eRNA bound to eDNA in integrity of extracellular NA fibres, but the presented data are not convincing enough to the reviewer. Although accuracy of data presented in Supplementary Figure 8A may be doubtful as mentioned above (see current comment # 2), RNase A and RNase H reduced the size of extracellular nucleic acids to a similar extent, and their combination with DNase I did so more remarkably (Supplementary Figure 8A). In addition, 0.4 mg/ml DNase I alone degraded eDNA for 30 min at 37°C. These results make the reviewer wonder how the same enzymatic treatments affect extracellular fibres in the biofilm stained with TOTO-1.

Lanes 2-6 and 7-9 are now presented in different figure panels to enable them to be presented with the respective ladders from the analyses.

This is clarified by changing the text describing size of nucleic acids after enzymatic treatment,

Line 185:

*“The NA gel isolate consisted of nucleic acids 2000-10000 base pairs (bp) in length (Supplementary Figure 8A(i), lane 2). Heating to 55°C was shown previously to disrupt the network eDNA structure⁹ and here slightly reduced NA size to 1500-5000 bp (Supplementary Figure 8A(i), lane 3). RNase A treatment also reduced NA size to a similar degree (Supplementary Figure 8A(i), lane 4). DNase I treatment, on the other hand, reduced the NAs to 400-1000 bp (Supplementary Figure 8A(i), lane 5) and subsequent RNase A treatment further reduced them to 100 bp (Supplementary Figure 8A(i), lane 6). Thus, the extracellular nucleic acid gel isolate from *P. aeruginosa* biofilms was additionally sensitised to RNase A treatment after DNase I treatment, which is consistent with eRNA being complexed to eDNA in the reconstituted extracellular NA gel extract.”*

As per the reviewers suggestion, we assessed the effect of the RNase A and RNase H treatments on eDNA fibre fluorescence signal intensity, showing complete disappearance of eDNA fibres after 0.4 mg ml⁻¹ DNase I treatment, 51% reduction of eDNA fibres of *P. aeruginosa* biofilms and 17.6% reduction of eDNA fibre of *P. aeruginosa* biofilms (Figure 6D), which is consistent with our rheological data showing an increase in tan δ following 0.3 mg ml⁻¹ of both RNase H and RNase A treatments respectively, and that only DNase I treatment increased tan δ to > 1 (Figure 6E).

This is clarified with following text

Line 471:

“Without mild pre-DNase I treatment with 0.3 mg ml⁻¹ of RNase A and RNase H, there was a slight increase in tan δ , and 18 and 51% reductions in eDNA fibre fluorescent intensity respectively, demonstrating that their viscoelastic networks were preserved (Figure 6D and E). Mild DNase I treatment (0.1 mg ml⁻¹) with RNase H buffer alone additionally did not dissolve the biofilm and resulted in less than 4% loss of eDNA fibre fluorescent intensity (Figure 6D and E).”

In Figures 5A and 5B, only extracellular fibres treated with 0.1 mg/ml DNase I and 0.1 mg/ml DNase I + uncertain concentration of RNase H are shown.

These concentrations have now been included in the figure caption. As per reviewer 4 suggestion, sections and 5 and 6 have been swapped.

In addition, Figure 2 in the response letter shows combination of uncertain concentrations of DNase I and RNase A degraded extracellular fibres in the 5-day biofilm.

This figure is included here with the DNase I and RNase A concentrations stated in the figure caption.

Rejoinder Figure D: Confocal micrograph of live-dead staining (Green- Syto9 and Red-Propidium iodide) of alkalinised five-day *P. aeruginosa* biofilm (i), 0.4 mg ml⁻¹ DNase I+ 0.3 mg ml⁻¹ RNase A treated five-day old *P. aeruginosa* biofilm (ii). The scale bars represent 10 μ m. (iii) Fluorescence intensity quantification (n = 5) for biofilm viability of non-treated, alkalinised and DNase1+RNase A treated five-day *P. aeruginosa* biofilm using ImageJ Fiji software (n=5)

To compare effects of enzymes on fragmentation and fibre structure of extracellular nucleic acids in the biofilm, it is better to show CLSM images of biofilm without enzyme treatment (control) as well as ones treated with 0.4 mg/ml DNase I alone, 0.3 mg/ml RNase A alone, 0.3 mg/ml RNase H alone, 0.4 mg/ml DNase I + 0.3 mg/ml RNase A, 0.4 mg/ml DNase I + 0.3 mg/ml RNase H in the same figure. Of course, it is good to show the images with lookup tables for fair comparison.

As per the reviewers suggestion we include this figure with various combinations of DNase I, RNase A and RNase H, although we used 0.1 mg ml⁻¹ DNase I only in the manuscript figure as 0.4 mg ml⁻¹ DNase I (targeting duplexed, single stranded and hybrid DNA) achieved full degradation of eDNA fibres by itself, as already shown in our previous manuscript⁶ and included again here for the convenience of the reviewer (i.e., 98% fibre degradation, Figure 6D and Supplementary Figure 10). This is explained in the following text:

Line 464:

*“This increased the $\tan \delta$ for the biofilm, relative to the untreated biofilm, in the rheology frequency sweep and reduced eDNA fibre fluorescence intensity by 47% (Figure 6A-E);, although $\tan \delta$ remained below 1. This is consistent with the observation that the viscoelastic network was preserved. However, when RNase A was replaced with 0.3 mg ml⁻¹ of RNase H, targeting the RNA strand of RNA:DNA hybrids, the model *P. aeruginosa* biofilm dissolved and lost its viscoelasticity, as indicated by the increase in $\tan \delta$ to greater than 1 and 73% reduction in eDNA fibre fluorescence intensity (Figure 6D and E). Without mild pre-DNase I treatment and 0.3 mg ml⁻¹ of RNase A and RNase H, there was a slight increase in $\tan \delta$, and 18, and 51%*

reductions in eDNA fibre fluorescent intensity respectively. This demonstrates that the viscoelastic networks were preserved (Figure 6D and E). Mild DNase I treatment (0.1 mg ml⁻¹) with RNase H buffer alone additionally did not dissolve the biofilm and resulted in less than 4% loss of eDNA fibre fluorescence intensity (Figure 6D and E)."

The methodology for eDNA fibre reduction after enzymatic treatments is included with the following text:

Line 734:

"The eDNA fibre reduction was quantified using the threshold option to eliminate the cell signal based on the fluorescence intensity difference between cells and eDNA fibres, obtained using ImageJ Fiji software."

9. The authors did not fully address the concern on the role of eRNA in formation of the biofilm and extracellular NA fibres, which was pointed out by this reviewer in the previous comment #3. Do 0.3 mg/ml RNase A and 0.3 mg/ml RNase H inhibit formation of biofilm and extracellular NA fibres when they are added to the culture media at the onset of biofilm formation?

We agree it is an interesting experiment to do, but it wouldn't address the question of whether RNA is a key viscoelastic compound. Firstly, in this case, biofilms did not form until after 16 h, so it is likely that the RNases would be degraded by proteolysis from the planktonic cells before the matrix is established. Secondly, if we could get RNase to persist in the broth, they would not necessarily be specific for eRNA. This would lead to other effects on cells not necessarily related to the matrix. Finally, we would have to employ both RNase and mild DNase I treatment in order to assign mechanism of action to the eRNA fibres, which would invoke additional experimental complications.

Minor comments

1. Supplementary Figure 14B: The authors may have done copying/pasting the DNA ladder maker from outside (some web site of company?). As mentioned above, this is very bad practice. If DNA makers are indistinguishable at high molecular sizes, the authors have to perform the same experiment again using low concentration of agarose.

We have amended the corresponding image by including the ladder in the same image.

2. Lines 181-191 are not correct: (i) The size of nucleic acids cannot be estimated as the marker is illegally connected to the image.

We have now presented lanes 2-6 and 7-9 in separate images allowing them to be presented along with their respective ladders (Supplementary Figure 8A and B). The above lines are thus amended to the nucleic acid sizes as determined from this new ladder.

(ii) There is less evidence that RNase A treatment of the NA gel did not disrupt the network structure. Supplementary Figure 10B shows effects on extracellular fibres in the biofilm treated with RNase A but lacks fair comparison with untreated samples as mentioned above.

We now quantify the reduction in eDNA fibre fluorescence signal intensity in both untreated and 0.3 mg ml⁻¹ RNase A treated *P. aeruginosa* biofilms (Figure 6D) showing an 18% reduction in eDNA fibre intensity as compared to untreated biofilms, consistent with only a slight increase in tan δ (Figure 6D and 6E).

This is clarified with following text.

Line 328:

“Only an 18% reduction in eDNA fibre fluorescence intensity was observed in five-day P. aeruginosa biofilms following 0.3 mg ml⁻¹ RNase A treatment, which was not enough to break down the eDNA network (Supplementary Figure 10(iv)).”

Line 471:

“Without mild pre-DNase I treatment of 0.3 mg ml⁻¹ of RNase A and RNase H, there was a slight increase in tan δ , and 18 and 51% reductions in eDNA fibre fluorescence intensity respectively, demonstrating that their viscoelastic networks were preserved (Figure 6D and E).”

(iii) There is a reduction in molecular size of nucleic acids after RNase A treatment (Supplementary Figure 8A, lane 4).

This has been amended to:

Line 187:

“RNase A treatment also reduced their size to a similar degree (Supplementary Figure 8A(i), lane 4).”

(iv) The extracellular nucleic acids in the gel isolate from *P. aeruginosa* biofilms were sensitive to all enzymes to different extents and highly susceptible to combined enzymes compared with single enzymes (Supplementary Figure 8A).

This has been amended to:

Line 191:

“Thus, the extracellular nucleic acid gel isolate from P. aeruginosa biofilms was additionally sensitised to RNase A treatment after DNase I treatment.”

(v) These results do not suggest that RNA was protected from enzymatic hydrolysis.

We have removed reference to the effect of DNA binding protecting RNA from RNase A.

3. Thank you for providing a result showing degradation of gamma globulin (Figure 1 in the response letter). However, the band intensity of a protein larger than 170 kDa seems to be slightly reduced in the lane 3. In addition, no band corresponding to DNase I is visible. How much concentration of gamma globulin was treated with 0.4 mg/ml DNase I and how long? The catalog numbers of the other enzymes are indicated in the methods section but that of DNase I is missing. Furthermore, gamma globulin is a tightly folded protein and thus some how resistant to proteases. Unfolded proteins (e.g., casein) is generally used as a substrate to examine activities of proteases in biochemical studies.

We have performed SDS-PAGE on protein standard casein (Sigma Aldrich, product no:7078) as suggested by the reviewer. The gel image (Figure 4 in response letter) showed no change in band molecular weight of casein (24 kDa) before and after DNase I treatment for 1 h at 37°C suggesting no proteolytic activity of DNase I. The DNase I band could not be detected in the gel. We have provided the SDS-PAGE gel image below for the reviewer's consideration (Figure 5 below).

The treatments conditions are included to the caption of Figure 1 of response letter. The catalog number is included in the methods section.

Lane 1- ladder, 2- 0.4 mg ml⁻¹ untreated casein, lane 3- 0.4 mg ml⁻¹ DNase I treated casein.

Rejoinder Figure E: SDS-PAGE of caesein protein standard (0.4 mg ml⁻¹) before (second lane) and after 0.4 mg/ml DNase I treatment at 37°C for 1 hour (Third lane) and prestained Nu-PAGE protein ladder (Thermofisher) (First lane).

4. It is strange that not only vertical error bars but also horizontal error bars are shown (Figure 2D, day 1 white square)

We thank the reviewer for picking up the error with this data set and have corrected it to horizontal error bar as for all other data sets in the corresponding image.

5. Supplementary Figure 15C: Obviously, this is also minor, but some *lasB*-positive signals are visible in the region close to the scale bar.

The red spots near the scale bar are background noise. To avoid confusion, we have described this in the corresponding figure legend.

Reviewer #3 (Remarks to the Author):

Authors have successfully addressed my previous comments.

Reviewer #4 (Remarks to the Author):

The revised manuscript by Mugunthan et al has much improved with the additional (clarifying) experiments and the more detailed explanations.

However, I still have a few concerns

1. The majority of the data appears to be shown in the supplement but there are several results that should be shown in the main manuscript. For example, RNase H only treated biofilms should be shown in Figure 5. I strongly encourage the authors to carefully review their data and move pertinent information including controls to the main manuscript

We thank the reviewer for the suggestion. As per reviewer's suggestion, we have moved material from the Supplementary Information into the main text.

These include,

- 1) Addition of 3- dimensional confocal micrograph of RNase H only treated biofilm to Figure 6B along with eDNA fibre reduction quantification data for enzymatic treated biofilms (Figure 6D).
- 2) Transfer of rheology data of both untreated and alkalinised wildtype *P. aeruginosa* biofilm to Figure 1E from Supplementary Figure 5C.
- 3) Inclusion of fluorescent quantification of cell viability across different days of biofilm growth in Figure 2F.
- 4) Previous Supplementary Figure 15 showing DNase I treated clinical sputum sample, along with negative and positive control have been shifted to the main Figure 5.
- 5) Sections 5 and 6 in the main text, have now been swapped as per reviewer's suggestion.

2. None of the data including the next gen sequencing data include statistics. Please include throughout. Also, please indicate statistical methods in the Methods section.

We have addressed this comment by adding the required information to the methods section.

This is clarified with following text.

Line 669

"Statistics

In this study, differential expression analysis was performed using the DESeq2 package in R. DESeq2 employs statistical methods tailored for RNA-Seq data analysis. The negative binomial model was fitted to estimate size factors and dispersion. Significance was determined using two statistical tests: the false discovery rate correction for adjusted p-values and a threshold for log 2-fold change and unadjusted p-values ($P < 0.01$). The volcano plot, created with ggplot2, visualised the results by plotting log 2-fold change against negative log₁₀ p-values. This analysis identified differentially expressed genes and provided insights into gene expression changes between conditions.

The standard deviation bars indicated in bar graphs are generated based on the mean values calculated from three biological replicates. For rheogram data, the biological triplicates are averaged for each condition and plotted against frequency."

Also the details on the number of replicates for each experiment has been included under each figure caption.

3. Data only show averages but should really show individual data points as well

We include the figure with the individual data points here for the reviewer's reference only, however, we believe that the averaged results with error bars and statistical analysis conveys the same information in a more succinct, and hence clearer, manner.

Rejoinder figure F: Standardised gene counts (%) of highly abundant mRNA transcripts (B) *lasB*, (C) *bkDA-2*, (D) *IpDV*, (E) *antB*, (F) *ssrA* and (G) *crcZ* in extracellular biofilm matrix and intracellular biofilm cells across different days of biofilm growth (days 1-5) with individual data points included in the above figure.

4. Figure 3

- Fig. 3E, indicate how biofilms were treated prior to image acquisition.

Biofilms were processed in a similar way as Figure 3A-D. The detailed methodology on biofilm sample preparation and processing for smiFISH is explained under the sample and probe preparation section in the materials and methods.

Line 743:

“Five-day P. aeruginosa biofilms and clinical sputum samples were collected by gently aspirating with a shortened 1 ml pipette tip. These were subsequently transferred to 2 ml Eppendorf tubes and rinsed with double-distilled water. The biofilms were neither fixed nor permeabilised to avoid intracellular contamination. The primary probe and secondary FLAP were prehybridised using conditions described in Supplementary Tables 6 and 7. The washed biofilms were incubated in 40% formamide freshly prepared in 1X SSC buffer for 15 min at 22°C. During incubation, Mix 1 and Mix 2 were prepared on ice (Supplementary Table 8). Mix 2 was vortexed for 30 s. Mix 1 and Mix 2 were combined and vortexed for 30 s. One hundred microlitres of the hybridisation mix were added to slides with deposited biofilms and covered with a Petri dish, wrapped in parafilm and incubated at 37°C overnight⁷. After hybridisation, biofilms were gently pipetted onto a poly-L-lysine-coated microscopic slide and covered with a coverslip.”

- Fig. 3G, Mander’s coefficient is only based on 4 images. Are these images from biological or technical replicates, please specify. If the images are from technical replicates, additional biological replicates will be required-

Figure 3G was acquired and quantified from three biological replicates. Confocal imaging and quantification on *in vitro* biofilms samples were also performed on biological replicates, as stated below:

Line 721:

“All microscopy imaging and quantification were performed across three biological replicates.”

- Fig. 3H-I, same as for Figure 3G; also, what is the meaning of (A) in legend for Fig 3I?

The motive of DNase I pretreatment was to understand if access to the eRNA by the smiFISH probe was impaired by eDNA binding. We assessed and quantified the effect of DNase I pretreatment only for *lasB* mRNA visualisation (Figure 3G) as it was the second most highly enriched extracellular mRNA (Supplementary table 2A) and focused on *lasB* as the biomarker for eRNA in *P. aeruginosa* biofilm matrices. With the information on DNase I pretreatment concentration (0.1 mg ml⁻¹) required for mRNA visualisation, other highly enriched extracellular mRNA such as *bkdA-2*, *IpDV* and *antB* were also identified in *P. aeruginosa* biofilm matrix (Figure 3H-I).

Line 44:

“Demonstrating its suitability as a biomarker for eRNA in Pseudomonas biofilm matrices”.

The letter (A) was a typographical error and rectified at the appropriate place.

5. Inconsistency, e.g. 5 days vs 5 d-

This has been addressed wherever necessary.

6. Awkward wording or excess of commas/periods, see l. 305, l. 369, l. 488-489,

These sentences have been edited to improve clarity.

7. Section on *lasB* and contaminating eRNAs is not very clear/confusing, with the section on contaminating eRNA and the role of *lasB* (*lasB* mutant data) not being very supportive of the major findings reported here. It actually brings the overall findings into question. Suggest revising

We thank the reviewer for their suggestion to improve the clarity of these sections. We now provide data showing the increase in the percentage of dead cells over time to account for contamination of extracellular matrix RNA with intracellular RNA, as described by the following:

Line 230:

*“gene counts standardised against total reads per sample were higher in the biofilm matrix than in biofilm cells and planktonic cells throughout the biofilm growth assay. These extracellular mRNA are therefore enriched in the biofilm matrix compared to its abundance in both planktonic and biofilm cells. *ssrA* and *crcZ* transcripts, on the other hand, were higher in the biofilm cells than in the biofilm matrix, yet present in similar abundances in the biofilm cells and planktonic cells (Figure 2F, G). The difference between eRNA levels in the biofilm matrix and its cells became increasingly less pronounced from days 2 to 5. This correlates with increased cell death and lysis with time (Supplementary Figure 7) and is likely a consequence of increased iRNA solubility in the matrix.”*

We highlight the importance of our microscopy data as validation of the sequencing, in terms of assigning eRNA and iRNA, where the *lasB* expression levels in the matrix correlate with *lasB* signal intensity, as determined using smiFISH:

Line 363:

*“This reduction in *lasB* from days 4 to 5 was also observed in the mRNA count for the biofilm (Figure 2B).”*

The *lasB* mutant was used to validate our smiFISH results, where we showed that the oligoribonucleotide probes did not bind to the fibres in the mutant. Thus, our observation of *lasB* co-localising with eDNA fibres in the wild type is validated, and we have clarified this with the following text:

Line 277:

*“These results suggest that the observed *lasB* signal resulted from the primary probe binding to *lasB* mRNA colocalising with eDNA fibres in the *P. aeruginosa* biofilm.”*

Finally, the *lasB* mutant was additionally used to assess whether *lasB* was a fundamental component of eDNA fibres, and we found that even after knocking the *lasB* gene out, *P. aeruginosa* produced eDNA fibres albeit with a different eRNA expression profile (Supplementary Figure 14A). Thus, we accept that the focus on *lasB* could be misconstrued as a statement that it is a fundamental component of the eDNA fibres, which the time-series and *lasB* knockout biofilm growth assays show to not be the case. Our observations of eRNA in the eDNA fibres in model and clinical biofilms using validated and novel microscopic methods (smiFISH), is the first time any defined nucleic acid sequence has been detected in

the extracellular matrix of any biofilm, as illustrated most clearly by *lasB*. It represents a significant advance from non-specific eDNA staining methods. Furthermore, it validates our method for determining iRNA and eRNA, provides direct evidence of the presence of eRNA in eDNA fibres, and also shows that *lasB* is a suitable biomarker for detecting the presence of eRNA in model and clinical *P. aeruginosa* biofilms. This is reflected with the following text:

Line 531:

*“While *lasB* appears not to be essential for *P. aeruginosa* biofilms, it is nonetheless a suitable biomarker for observing eRNA in viscoelastic eDNA networks of clinical and model *P. aeruginosa* biofilms.”*

8. suggest moving section I. 387-401 after section I. 419-445

This has been addressed as per reviewer’s suggestion. We have swapped the section and 5 and 6 in the main text and have included necessary controls and information into the main text from supplementary information.

This is clarified with the following text,

Line 411:

“*lasB* mRNA colocalises with eDNA fibres in airway specimens colonised with *P. aeruginosa*

*In a clinical bronchiectasis airway sputum specimen with a 96% abundance of *P. aeruginosa* relative to total microbial reads, colocalisation of *lasB* mRNA (Figure 5A, merged image of eDNA and *lasB*) was clearly observed along eDNA fibres following pre-treatment with 0.1 mg ml⁻¹ DNase I, as per the in vitro *P. aeruginosa* biofilm model.”*

Line 459:

“Extracellular RNA is key to viscoelasticity of *Pseudomonas* biofilms

*“There was a reduction in DNA length by approximately 8000 bp following alkaline transesterification of the NA gel isolate (Supplementary Figure 14B), indicating some loss of mass or secondary structure. To assess directly whether eRNA contributes to *P. aeruginosa* biofilm viscoelasticity, 0.3 mg ml⁻¹ of RNase A digestion of the *P. aeruginosa* biofilm was coupled with mild pre-DNase I treatment (0.1 mg ml⁻¹), as per the smiFISH microscopy protocol. This increased the $\tan \delta$ for the biofilm, relative to the untreated biofilm, in the rheology frequency sweep and reduced eDNA fibre fluorescence intensity by 47% (Figure 6A-E); although $\tan \delta$ remained below 1. This is consistent with the observation that the viscoelastic network was preserved. However, when RNase A was replaced with 0.3 mg ml⁻¹ of RNase H, targeting the RNA strand of RNA:DNA hybrids, the model *P. aeruginosa* biofilm dissolved and lost its viscoelasticity, as indicated by the increase in $\tan \delta$ to greater than 1 and 73% reduction in eDNA fibre fluorescence intensity (Figure 6D and E). Without mild pre-DNase I treatment and 0.3 mg ml⁻¹ of RNase A and RNase H, there was a slight increase in $\tan \delta$, and 18, and 51% reductions in eDNA fibre fluorescence intensity respectively. This demonstrates that the viscoelastic networks were preserved (Figure 6D and E). Mild DNase I treatment (0.1 mg ml⁻¹) with RNase H buffer alone additionally did not dissolve the biofilm and resulted in less than 4% loss of eDNA fibre fluorescence intensity (Figure 6D and E)....”*

1. Xu S, *et al.* Transcriptomic Analysis Reveals the Role of tmRNA on Biofilm Formation in *Bacillus subtilis*. *Microorganisms* **10**, 1338 (2022).
2. Prescott AM, Fricker CR. Use of PNA oligonucleotides for the in situ detection of *Escherichia coli* in water. *Molecular and Cellular Probes* **13**, 261-268 (1999).
3. Tang YZ, Gin KY, Lim TH. High-temperature fluorescent in situ hybridization for detecting *Escherichia coli* in seawater samples, using rRNA-targeted oligonucleotide probes and flow cytometry. *Appl Environ Microbiol* **71**, 8157-8164 (2005).
4. Fuchs BM, Wallner G, Beisker W, Schwippl I, Ludwig W, Amann R. Flow Cytometric Analysis of the In Situ Accessibility of *Escherichia coli* 16S rRNA for Fluorescently Labeled Oligonucleotide Probes. *Appl Environ Microbiol* **64**, 4973-4982 (1998).
5. Chew SC, *et al.* Dynamic remodeling of microbial biofilms by functionally distinct exopolysaccharides. *mBio* **5**, e01536-01514 (2014).
6. Seviour T, *et al.* The biofilm matrix scaffold of *Pseudomonas aeruginosa* contains G-quadruplex extracellular DNA structures. *NPJ Biofilms Microbiomes* **7**, 27 (2021).
7. Tsanov N, *et al.* smiFISH and FISH-quant - a flexible single RNA detection approach with super-resolution capability. *Nucleic Acids Res* **44**, e165-e165 (2016).

Reviewer #3 (Remarks to the Author):

Authors have successfully addressed all reviewers comments.

Reviewer #4 (Remarks to the Author):

The authors addressed the reviewer comments adequately. It is apparent that responses to reviewer comments have significantly evolved the manuscript. I have no further comments/suggestions.